# What Is Missing In Homophily? Disentangling Graph Homophily For Graph Neural Networks

**Yilun Zheng[1], Sitao Luan[2], Lihui Chen[1] †**

yilun001@e.ntu.edu.sg, sitao.luan@mail.mcgill.ca, elhchen@ntu.edu.sg
[1]Nanyang Technological University, Centre for Info. Sciences and Systems;
[2]Mila - Quebec Artificial Intelligence Institute; † Corresponding Author

## Abstract

Graph homophily refers to the phenomenon that connected nodes tend to share similar characteristics. Understanding this concept and its related metrics is crucial for designing effective Graph Neural Networks (GNNs). The most widely used homophily metrics, such as edge or node homophily, quantify such "similarity" as label consistency across the graph topology. These metrics are believed to be able to reflect the performance of GNNs, especially on node-level tasks. However, many recent studies have empirically demonstrated that the performance of GNNs does not always align with homophily metrics, and how homophily influences GNNs still remains unclear and controversial. Then, a crucial question arises: What is missing in our current understanding of homophily? To figure out the missing part, in this paper, we disentangle the graph homophily into three aspects: label, structural, and feature homophily, which are derived from the three basic elements of graph data. We argue that the synergy of the three homophily can provide a more comprehensive understanding of GNN performance. Our new proposed structural and feature homophily consider the neighborhood consistency and feature dependencies among nodes, addressing the previously overlooked structural and feature aspects in graph homophily. To investigate their synergy, we propose a Contextual Stochastic Block Model with three types of Homophily (CSBM-3H), where the topology and feature generation are controlled by the three metrics. Based on the theoretical analysis of CSBM-3H, we derive a new composite metric, named Tri-Hom, that considers all three aspects and overcomes the limitations of conventional homophily metrics. The theoretical conclusions and the effectiveness of Tri-Hom have been verified through synthetic experiments on CSBM-3H. In addition, we conduct experiments on 31 real-world benchmark datasets and calculate the correlations between homophily metrics and model performance. Tri-Hom has significantly higher correlation values than 17 existing metrics that only focus on a single homophily aspect, demonstrating its superiority and the importance of homophily synergy. Our code is available at `https://github.com/zylMozart/Disentangle_GraphHom`.

## 1 Introduction

Graph Neural Networks (GNNs) have been widely used in processing non-Euclidean data due to their superiority in extracting topological relations [10, 13, 26, 41, 35]. They have achieved great success on numerous real-world applications, *e.g.,* recommendation [49, 58], bio-informatics [21, 20] and telecommunication [34]. It is found that their success, especially on node-level tasks, is closely related to the homophily assumption [47, 72, 38, 43, 36, 71], *i.e.,* similar nodes tend to be connected [17]. On the other hand, when dissimilar nodes are more likely to be connected, which is known as the non-homophily/heterophily scenario, GNNs fail to capture the useful neighbor information and even underperform Multilayer perceptrons (MLPs) [37]. Several homophily metrics, such as edge homophily [72, 1] and node homophily [50] were proposed, which were believed to be able to recognize the difficult datasets [43] and measure the performance of GNNs [39].

However, recent studies [46, 39, 51, 40] show that the conventional homophily metrics [72, 1, 50] are insufficient to measure the performance of GNNs: Luan *et al.* [38] show that the homophily metrics cannot tell if GNNs work well under heterophily. Ma *et al.* [46] reveals that homophily is not a necessary assumption for effective GNNs and they propose to identify "good" and "bad" heterophily to explain why GNNs still work well under heterophily. Luan *et al.* [39] discovers a mid-homophily pitfall, showing the performance of GNNs reaches the worst in a medium level of homophily instead of the lowest. Then, a crucial question arises based on the above studies: What is missing in our current understanding of homophily?

In this paper, we fill the missing parts by investigating different perspectives of the "node similarity". Conventional homophily metrics quantify the "similarity" as an indicator function of whether connected nodes share the same label while ignoring the co-existence of three basic elements in graph data: label, structural, and node feature information. The ignorance of structural and feature information leads to insufficient understanding and unsatisfactory alignment between homophily metrics and GNN performance.

A complete understanding of graph homophily should include all the above three basic elements. To this end, we disentangle graph homophily into three corresponding aspects: label, structural, and feature homophily. Specifically, our new proposed structural homophily quantifies the "similarity" by considering the neighborhood structure consistency, and feature homophily measures the dependencies of node features across the topology. To investigate how their synergy affects GNN performance, we propose a Contextual Stochastic Block Model controlled with three types of Homophily (CSBM-3H). The node feature generation process in CSBM-3H breaks the *i.i.d.* assumption in previous studies [46, 39], which is closer to real-world scenarios [14, 59, 61]. With the three metrics, CSBM-3H enables a more comprehensive study on the impact of graph homophily than previous analysis [46, 39, 29, 60].

From the theoretical study of CSBM-3H, we derive a new composite metric named Tri-Hom to measure the synergy, which includes all three homophily aspects. Through CSBM-3H, our theoretical analysis and simulation results both show that the performance of GNNs is highly influenced by Tri-Hom. It can help explain how the three types of homophily influence GNN behavior individually or collectively. In addition, our theoretical findings can explain some interesting phenomena observed in previous literature, such as "good" or "bad" heterophily [46, 38] and the impact of feature shuffling on GNNs [29]. To verify the effectiveness of Tri-Hom, we conduct experiments on 31 real-world datasets. The results show that GNN performance is significantly better aligned with Tri-Hom than the other 17 existing metrics that focus only on a single homophily aspect. This implies that Tri-Hom can complete the absent parts in existing homophily metrics.

## 2 Preliminary

We denote $\mathcal{G} = (\mathcal{V}, \mathcal{E})$ as an undirected graph, where $\mathcal{V}$ is the node set and $\mathcal{E}$ is the edge set. The graph has $N$ nodes with $C$ classes. The adjacency matrix of the graph is denoted as $\boldsymbol{A} \in \mathbb{R}^{N \times N}$. We use $\boldsymbol{A}_{uv} = 1$ or $e_{uv} \in \mathcal{E}$ to denote the existence of an edge between node $u$ and $v$, otherwise $\boldsymbol{A}_{uv} = 0$ or $e_{uv} \notin \mathcal{E}$. Node degree vector is denoted as $\boldsymbol{D} \in \mathbb{R}^N$ where $\boldsymbol{D}_u$ is the degree of node $u$. Node label vector is denoted as $\boldsymbol{Y} \in \mathbb{R}^N$ and its one-hot encoding matrix is $\boldsymbol{Z} \in \mathbb{R}^{N \times C}$. The number of nodes in class $c$ is denoted as $N_c = |\{u|\boldsymbol{Y}_u = c, u \in \mathcal{V}\}|$. The neighbor set of node $u$ is denoted as $\mathcal{N}_u = \{v|e_{uv} \in \mathcal{E}\}$. The features of all the nodes is denoted as $\boldsymbol{X} \in \mathbb{R}^{N \times M}$, where $\boldsymbol{X}_{v,:}$ are the features of node $v$ with $M$ dimensions. We use $\boldsymbol{I}_E \in \mathbb{R}^{E \times E}$ and $\boldsymbol{1}_E \in \mathbb{R}^{E \times E}$ to denote identify matrix and all-ones matrix with size $E$, respectively.

**Graph homophily metrics** are used to measure the similarity between connected nodes. Edge [1, 72] and node homophily [50] are 2 most commonly used metrics and are defined as follows,

$$h_{\text{edge}}(\mathcal{G}, \boldsymbol{Y}) = \frac{\left|\{e_{uv}|e_{uv} \in \mathcal{E}, Y_u = Y_v\}\right|}{|\mathcal{E}|}, \; h_{\text{node}}(\mathcal{G}, \boldsymbol{Y}) = \frac{1}{|\mathcal{V}|} \sum_{v \in \mathcal{V}} \frac{\left|\{u|u \in \mathcal{N}_v, Y_u = Y_v\}\right|}{\left|\mathcal{N}_v\right|} \quad (1)$$

These metrics qualify the ratio of whether the labels of two connected nodes are the same in a graph. However, this definition of graph homophily only considers a label aspect and neglects structural and feature aspects, resulting in a partial understanding of graph homophily. Therefore, we propose to disentangle the graph homophily as label, structural, and feature homophily in the next section.

**Graph-aware models** $\mathcal{M}^{\mathcal{G}}$ **and graph-agnostic models** $\mathcal{M}^{\neg \mathcal{G}}$ refer to the models that either utilize structure information or do not, respectively. For example, baseline graph-aware models

$\mathcal{M}^{\mathcal{G}}$, such as Graph Convolutional Network (GCN) [26], Graph Attention Network (GAT) [57] and GraphSage [16], encode both graph structure and node feature information in each layer; the corresponding graph-agnostic models $\mathcal{M}^{\neg\mathcal{G}}$ are the Multilayer Perceptrons (MLPs), which only encode node features [39].

**Structural-agnostic features** refer to the node features $X$ that are conditionally independent of graph topology $A$ given $Y$, *i.e.,*$(X \perp\!\!\!\perp A|Y)$; **structural-aware features** indicate $(X \not\perp\!\!\!\perp A|Y)$.

## 3 Disentangled Graph Homophily

In this section, we first introduce the definition of disentangled graph homophily from label, structural, and feature aspects to complete the missing part of the graph homophily. Then, in the next section, we will introduce how they collectively impact the performance of GNNs.

### 3.1 Label Homophily

**Definition 1.** *Label homophily is defined as the consistency of node labels across the topology.*

Label homophily is the most widely used conventional metric of graph homophily and it qualifies the similarity between connected nodes $u$ and $v$ using an indicator function $\mathbb{1}(Y_u = Y_v)$. Most of the conventional homophily metrics focus on label homophily, including edge homophily [1, 72], node homophily [50], class homophily [33], adjusted homophily [51], density-aware homophily [30], 2-hop neighbor class homophily [5], and neighbor homophily [15].

However, label homophily only focuses on the consistency of label information for connected nodes while neglecting structural and feature information, which are two indispensable components of graph data. Hence, it offers only a partial understanding of graph homophily, which cannot always align well with the performance of GNNs [46, 38, 39]. To capture the missing structural and feature information and better understand graph homophily, we give the definitions of structural and feature homophily in the following 2 subsections.

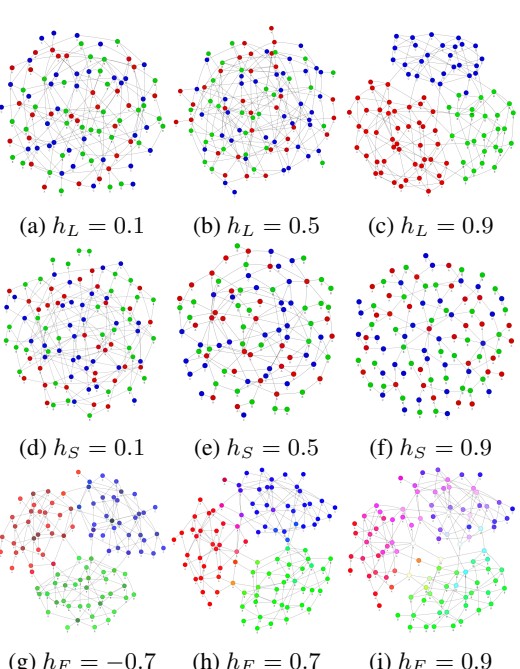

(a) $h_L = 0.1$    (b) $h_L = 0.5$    (c) $h_L = 0.9$

(d) $h_S = 0.1$    (e) $h_S = 0.5$    (f) $h_S = 0.9$

(g) $h_F = -0.7$    (h) $h_F = 0.7$    (i) $h_F = 0.9$

Figure 1: Visualization of synthetic graphs generated by CSBM-3H with varying levels of label homophily, structural homophily, and feature homophily. The node colors denote node classes in sub-figure (a-f) and node features in sub-figure (g-i)

### 3.2 Structural homophily

For structural homophily, the "atom" information of a node is structural information instead of the label. It is meaningless to define the structural homophily using the consistency across the graph topology as in the label homophily because the structural information already contains the information from the graph topology. Therefore, we define the structural information as the consistency of structural information among the nodes from the same classes[1], which better disentangles itself from the label homophily.

**Definition 2.** *Structural homophily is defined as the consistency of structural information of nodes within the same class. The structural homophily in a graph is defined as:*

$$h_S(\mathcal{G}, \mathcal{S}, \boldsymbol{Y}) = \frac{1}{C} \sum_{c=1}^{C} h_{S,c}, \ \ where \ h_{S,c}(\mathcal{G}, \mathcal{S}, \boldsymbol{Y}) = 1 - \frac{\sigma(\{\mathcal{S}(u)|u \in \mathcal{V}, Y_u = c\})}{\sigma_{max}} \quad (2)$$

---

[1]There we do not consider the inter-class structural information because the structural homophily represents a property of a graph instead of the node distinguishability[39]. The detailed discussion of the node distinguishability is shown in Section 4.2.

*where $h_{S,c}$ is the class-wise structural homophily for class $c$, function $\mathcal{S}(\cdot)$ measures structural information, $\sigma$ denotes standard deviation of structural information, and $\sigma_{max}$ denotes the maximum value of $\sigma$.*

In this paper, we quantify the structural information for node $u$ through neighbor distribution (the class distribution of local neighbors) $\boldsymbol{D}_u^{\mathcal{N}} = [p_{u,1}, p_{u,2}, \ldots, p_{u,c}]$, where $p_{u,k} = \frac{|\{Y_v=k|v\in\mathcal{N}_u\}|}{|\mathcal{N}_u|}$ is the proportion of neighbors of node $u$ that belong to class $c$. A high structural homophily indicates that the graph-aware models leveraging structural encoding will have similar embeddings for intra-class nodes after aggregation, which are expected to outperform graph-agnostic models, irrespective of a low label homophily. There are also some homophily metrics that focus on the structural aspect in previous studies, including label informativeness [51], neighborhood similarity [46], and aggregation homophily [38], which is similar as the structural homophily defined there.

### 3.3 Feature Homophily

Previous feature-based graph homophily metrics, such as generalized edge homophily [23], local similarity [7], attribute homophily [67], and class-controlled feature homophily [29], mainly focus on the consistency of node features across the graph topology, which is similar as the definition of Dirichlet energy in graphs. However, these homophily metrics on feature consistency cannot fully disentangle itself from label homophily: Since the features of nodes in a graph are supposed to depend on their classes, when the graph shows a high/low label homophily, the connected nodes are more likely to share the same/different labels, resulting in a high/low feature similarity. Therefore, these feature-based homophily metrics are dependent on label homophily. Such dependency contains redundancy, which decreases the useful information inside feature-based homophily and impedes our understanding of the relationship between node features and GNN performance.

To disentangle the feature effect from label and graph structure, we define the feature homophily as the dependencies of node features across the graph topology, thereby dissociating it from label homophily and structural homophily. Inspired by graph diffusion [6] and interactive particle systems [55, 64], we have the structural-agnostic unobserved feature $\boldsymbol{X}(0)$ and the observed structural-aware feature $\boldsymbol{X}$ that satisfy the following relation

$$\boldsymbol{X} = \Big[ \sum_{t=0}^{\infty} (\omega \boldsymbol{A})^t \Big] \boldsymbol{X}(0) = (\boldsymbol{I} - \omega \boldsymbol{A})^{-1} \boldsymbol{X}(0) \tag{3}$$

The detailed process of this relation is given in Appendix B. Here $\omega \in \left(-\frac{1}{\rho(\boldsymbol{A})}, \frac{1}{\rho(\boldsymbol{A})}\right)$ is a parameter that controls the feature dependencies, where a positive, negative, or zero value corresponds to an attractive relation, repulsive relation, or independence of the nodes with their neighbors in graphs [55, 64]. The feature dependencies $(\omega \boldsymbol{A})^t$ of $t$-order neighbors are introduced to structural-agnostic features $\boldsymbol{X}(0)$. Finally, the state of all the nodes will converge to an equilibrium with structure-aware feature $\boldsymbol{X}$. The $\omega$ in Eq. (3) is independent of the graph topology because no matter how the label homophily or structural homophily changes, $\omega$ will remain unaffected. To disentangle feature homophily from label homophily and structural homophily, we define the feature homophily based on $\omega$ as follows.

**Definition 3.** *Feature homophily is defined as the degree of feature dependencies of nodes across the topology. For the linear case of the graph diffusion process with feature dependencies, the feature homophily for feature $m$ satisfies*

$$\boldsymbol{X}_{:,m} = \left( \boldsymbol{I} - \frac{h_{F,m}}{\rho(\boldsymbol{A})} \boldsymbol{A} \right)^{-1} \boldsymbol{X}_{:,m}(0) \tag{4}$$

*where $\rho(\boldsymbol{A})$ is the spectral radius of $\boldsymbol{A}$, $\boldsymbol{X}(0) \sim p(\boldsymbol{X}|\boldsymbol{Y})$ are the unseen structural-agnostic node features, and $\boldsymbol{X} \sim p(\boldsymbol{X}|\boldsymbol{Y}, \boldsymbol{A})$ is the observed structural-aware node features. The feature homophily for the whole graph is the averaged feature homophily for all the features*

$$h_F(\mathcal{G}, \boldsymbol{X}, \boldsymbol{Y}) = \frac{1}{M} \sum_{m=1}^{M} h_{F,m} \tag{5}$$

Note that it is easy to control the feature homophily in the generation of synthetic graphs. However, since both $h_{F,m}$ and $\boldsymbol{X}_{:,m}(0)$ are unknown in Eq. (4), one more condition is required to estimate feature homophily in real-world datasets. To address this issue, we consider the case where the intra-class distances of $\boldsymbol{X}_{:,m}(0)$ are small. This case holds in lots of real-world scenarios [38, 39]

and we can utilize this property to estimate $h_{F,m}$ without solving $\boldsymbol{X}_{:,m}(0)$. Specifically, the feature homophily $h_{F,m}$ for feature $m$ can be estimated with the following optimization process

$$h_{F,m}^*(\mathcal{G}, \boldsymbol{X}_{:,m}, \boldsymbol{Y}) = \arg\min_{h_{F,m}} \sum_{\substack{u,v \in \mathcal{V}, \\ Y_u = Y_v}} \|X_{u,m}(0) - X_{v,m}(0)\|^2, \text{ where } \boldsymbol{X}_{:,m}(0) = \left(\boldsymbol{I} - \frac{h_{F,m}}{\rho(\boldsymbol{A})}\boldsymbol{A}\right)\boldsymbol{X}_{:,m} \quad (6)$$

The estimation of feature homophily is invariant to the operations of feature shifts, scaling, or changing variance, where the proof is shown in Appendix C. This estimation process will be used in Section 5.2 for calculation.

**Remark** To better understand the definitions of three types of graph homophily, Figure 1 visualizes examples of graphs under varying $h_L$, $h_S$, and $h_F$: 1) **Label homophily.** As label homophily $h_L$ increases, as shown in Figures 1(a), (b), and (c), nodes are more likely to connect with others that share the same label. Particularly, a high $h_L$ (Figure 1(c)) results in several clusters with distinct class boundaries, while a low $h_L$ causes nodes to more likely connect to nodes with different classes. 2) **Structural homophily.** As structural homophily $h_S$ increases, as shown in Figures 1(d), (e), and (f), the neighbor distributions of intra-class nodes become more consistent. Therefore, a high $h_S$ is expected to capture effective structural information with message aggregation. Interestingly, we also find that a higher $h_S$ makes a graph resemble planar graphs [3] and periodic graphs [9]. We hypothesize this phenomenon occurs because stable structural information leads to more regular and meaningful patterns, which would be interesting to explore the connection between $h_S$ and these geometric properties of graphs in the future. 3) **Feature homophily.** Figures 1(g), (h), and (i) illustrate different levels of feature homophily ($h_F$) within the same graph topology. Figure 1 (h) demonstrates that under a medium positive $h_F$, features of some boundary nodes exhibit characteristics of both neighboring classes. For instance, a node on the boundary of the red class and the blue class appears purple, a mixture of these classes. A higher $h_F$ (Figure 1 (i)) increases feature dependencies, particularly affecting more nodes closer to class boundaries. In social networks, a positive $h_F$ indicates that people's opinions are influenced by their friends, resulting in similar characteristics. Conversely, a negative $h_F$ causes nodes to become more dissimilar from their neighbors. As shown in Figure 1 (g), a negative $h_F$ creates a distinct boundary between classes. Additionally, for the intra-class nodes in Figure 1 (g), the node colors differ in shades from their neighbors. This occurs because node features become more dissimilar due to the "repulsive force" rather than the "attractive force" induced by a negative $h_F$. For example, in online social media, people are likely to argue with those holding different opinions on certain topics. After such interactions, individuals may reinforce their original opinions, a phenomenon resulting from the "repulsive force" associated with a negative $h_F$.

## 4 Impact of Disentangled Graph Homophily

To study the model performance in a graph, the Contextual Stochastic Block Model (CSBM) has been widely used to study the performance of GNNs with controlled graph topology and node features. Previous studies [51, 29, 39, 46] on graph homophily generally adopt a modified CSBM to control the label homophily through assigning nodes with different probabilities that connect to the nodes from other classes. Then the node features are sampled solely based on the classes. However, this graph modeling, which only considers label homophily, has two drawbacks: First, the probabilities of nodes from the same class connecting to the nodes with different classes are uniform, which lacks diversity. Second, the sampled node features are independent with their structures *i.e.,*$(\boldsymbol{X} \perp\!\!\!\perp \boldsymbol{A}|\boldsymbol{Y})$, which is uncommon in real-world scenarios where interactions influence the attributes of connected nodes [61, 14, 59]. Therefore, we propose a Contextual Stochastic Block Model with three types of Homophily (CSBM-3H), a random graph generative model that integrates the three types of homophily (Section 4.1), where the newly proposed structural homophily $h_S$ and feature homophily $h_F$ can well address the aforementioned two drawbacks and fills the missing part of graph homophily. Then, Based on CSBM-3H, we theoretically study how the graph-agnostic and graph-aware models are affected by label, structural, and feature homophily metrics to explore their relationship and verify the effectiveness of proposed metrics (Section 4.2).

### 4.1 CSBM-3H

**Graph Topology Generation** Following the topology generation process in existing studies [39, 60], we assume all the nodes are class-balanced and share the same node degree $d$. We use node homophily $h_L$ to control the label consistency across the graph topology and $h_S$ to control the consistency of neighbor distribution $\boldsymbol{D}^\mathcal{N}$ of nodes within the same classes. Then, the neighbor distribution can be expressed as:

$$\boldsymbol{D}^{\mathcal{N}} = \mathbb{E}_{\epsilon}[\boldsymbol{Z}\boldsymbol{S}], \text{ where } \boldsymbol{S} = \frac{1-h_L}{c-1}\mathbf{1}_C + (h_L - \frac{1-h_L}{c-1})\boldsymbol{I}_C + \epsilon \tag{7}$$

where $\boldsymbol{D}^{\mathcal{N}} \in \mathbb{R}^{N \times C}$ is the neighbor distribution for all the nodes, $\boldsymbol{S} \in \mathbb{R}^{C \times C}$ is a class-sampling matrix, and $\epsilon \in \mathbb{R}^{C \times C}$ is a noise matrix. Each entry of $\epsilon$ is a noise of neighbor sampling that follows a Gaussian distribution $N(0, \frac{(1-h_S)^2}{c-1})$. The class-sampling matrix $\boldsymbol{S}$ should be legal in practice [60] *i.e.,* $\boldsymbol{S}_{u,v} > 0$ and $\sum_v \boldsymbol{S}_{u,v} = 1$. Then an adjacency matrix $\boldsymbol{A}$ can be sampled from a neighborhood sampling matrix $\boldsymbol{A}_p = \frac{Cd}{N}\boldsymbol{D}^{\mathcal{N}}\boldsymbol{Z}^T$, where $\mathbb{E}[A_{uv} = 1] = (\boldsymbol{A}_p)_{uv}$ for each pair of nodes $u, v$. In this way, we control the label homophily $h_L$ and structural homophily $h_S$ in a graph.

**Node Feature Generation**. For any node $u$ in a graph, we first sample its structural-agnostic features $\boldsymbol{X}_u(0) \in \mathbb{R}^F$ from a class-wised Gaussian distribution $\boldsymbol{X}_u(0) \sim \boldsymbol{N}_{Y_u}(\boldsymbol{\mu}_{Y_u}, \boldsymbol{\Sigma}_{Y_u})$ with $\boldsymbol{\mu}_{Y_u} \in \mathbb{R}^F$ and $\boldsymbol{\Sigma}_{Y_u} \in \mathbb{R}^{F \times F}$. We also assume each dimension of the feature vector is independent of each other, thereby $\boldsymbol{\Sigma}_{Y_u} \in \mathbb{R}^{F \times F}$ is a diagonal matrix. Then the observed structural-aware features can be generated by the unseen structure-agnostic feature as described in Eq. (4).

### 4.2 Node distinguishability

Suppose we have the representations of node $u$ as $\boldsymbol{H}_{u:} = \frac{1}{d}\sum_{v \in \mathcal{N}_u}\boldsymbol{X}_{v:}$ for graph-aware models $\mathcal{M}^{\mathcal{G}}$ and $\boldsymbol{H}_{u:} = \boldsymbol{X}_{u:}$ for graph-agnostic models $\mathcal{M}^{\neg\mathcal{G}}$. Inspired by the principles of neural collapse [28, 27] and node distinguishability [39], we quantify the impact of the aforementioned homophily metrics on both the graph-aware models $\mathcal{M}^{\mathcal{G}}$ and graph-agnostic models $\mathcal{M}^{\neg\mathcal{G}}$ by measuring the ratio of intra-class node distance to inter-class distance. To ideally distinguish nodes from different classes, a smaller intra-class distance $D_{\text{intra}}(\boldsymbol{H})$ and larger inter-class distance $D_{\text{intra}}(\boldsymbol{H})$ is preferred because this will reduce boundary nodes and increase the margins among classes. The metric is defined as follows,

$$\mathcal{J} = \frac{D_{\text{intra}}(\boldsymbol{H})}{D_{\text{inter}}(\boldsymbol{H})} = \frac{\mathbb{E}_{y_u=y_v,\epsilon}\left[\|\boldsymbol{H}_u - \boldsymbol{H}_v\|^2\right]}{\mathbb{E}_{y_u \neq y_v,\epsilon}\left[\|\boldsymbol{H}_u - \boldsymbol{H}_v\|^2\right]} \tag{8}$$

A smaller $\mathcal{J}$ indicates better node embeddings for the model performance and vice versa, which has been proved in [39]. With the proposed CSBM-3H, we can analyze the impacts of $h_L$, $h_S$, and $h_F$ on $\mathcal{M}^{\neg\mathcal{G}}$ and $\mathcal{M}^{\mathcal{G}}$ by studying their relations with $\mathcal{J}$, which will be derived in the following theorems.

**Theorem 1.** In CSBM-3H, the ratio of the expectation of intra-class distance to the expectation of inter-class distance of node representations for graph-agnostic models $\mathcal{M}^{\neg\mathcal{G}}$ and graph-aware models $\mathcal{M}^{\mathcal{G}}$ is:

$$\mathcal{J}^{\neg\mathcal{G}} = (1 + \mathcal{J}_{\boldsymbol{N}}\mathcal{J}_h^{\neg\mathcal{G}})^{-1} \text{ and } \mathcal{J}^{\mathcal{G}} = (1 + \mathcal{J}_{\boldsymbol{N}}\mathcal{J}_h^{\mathcal{G}})^{-1} \tag{9}$$

where $\mathcal{J}_{\boldsymbol{N}} = \frac{\sum_{Y_u \neq Y_v}[2C(C-1)]^{-1}\|\boldsymbol{\mu}_{Y_u} - \boldsymbol{\mu}_{Y_v}\|^2}{C^{-1}|\boldsymbol{\sigma}^2|}$, $\mathcal{J}_h^{\neg\mathcal{G}} = \frac{1 - (\frac{h_F}{\rho(\boldsymbol{A})})^2(C(\frac{1-h_L}{C-1})^2 + C\frac{(1-h_S)^2}{C-1} + (\frac{h_L C-1}{C-1})^2)}{\left[1 - (\frac{h_F}{\rho(\boldsymbol{A})})(\frac{h_L C-1}{C-1})\right]^2}$,

and $\mathcal{J}_h^{\mathcal{G}} = \frac{(\frac{h_L C-1}{C-1})^2}{C(\frac{1-h_L}{C-1})^2 + C\frac{(1-h_S)^2}{C-1} + (\frac{h_L C-1}{C-1})^2}\mathcal{J}_h^{\neg\mathcal{G}}$. (See the proof in Appendix G.1.)

From Theorem 1 we can see that, $\mathcal{J}_{\boldsymbol{N}}$ is a normalized distance term which is a constant given the distribution of structural-agnostic features and is irrelevant with graph information; $\mathcal{J}_h^{\neg\mathcal{G}}$ and $\mathcal{J}_h^{\mathcal{G}}$ are controlled by the three types of homophily, which can reflect the influence of graph homophily on model performance. We name $\mathcal{J}_h^{\neg\mathcal{G}}$ and $\mathcal{J}_h^{\mathcal{G}}$ as **Tri-Hom** for $\mathcal{M}^{\neg\mathcal{G}}$ and $\mathcal{M}^{\mathcal{G}}$. To study the effect of Tri-Hom in more detail, we take the partial derivative of $\mathcal{J}_h^{\mathcal{G}}$ with respect to $h_S$, $h_F$, and $h_L$ to show the analytical results of their influences[2]. (See calculation in Appendix G.5, G.6, and G.7.)

**Theorem 2.1.** The partial derivative of $\mathcal{J}_h^{\mathcal{G}}$ with respect to label homophily $h_L$ satisfies,

$$\begin{cases} \frac{\partial \mathcal{J}_h^{\mathcal{G}}}{\partial h_L} < 0, & \text{if } h_L \in [0, \frac{1}{C}) \\ \frac{\partial \mathcal{J}_h^{\mathcal{G}}}{\partial h_L} \geq 0, & \text{if } h_L \in [\frac{1}{C}, 1] \end{cases} \tag{10}$$

From Theorem 2.1 we can see that, the worst performance of $\mathcal{M}^{\mathcal{G}}$ is reached when $h_L = \frac{1}{C}$, which corresponds to the scenario with the highest number of unpredictable boundary nodes among classes. This finding explains the misalignment between label homophily and GNN performance mentioned in previous studies [46, 38, 39].

---

[2]We also calculate the partial derivative of $\mathcal{J}_h^{\neg\mathcal{G}}$ and discuss the impact of three types of homophily on $\mathcal{M}^{\neg\mathcal{G}}$ in Appendix D.

**Theorem 2.2.** The partial derivative of $\mathcal{J}_h^{\mathcal{G}}$ with respect to structural homophily $h_S$ satisfies,

$$\frac{\partial \mathcal{J}_h^{\mathcal{G}}}{\partial h_S} \geq 0 \tag{11}$$

From Theorem 2.2 we can see that, a larger $h_S$ consistently improves the performance of $\mathcal{M}^{\mathcal{G}}$. This is intuitive for $\mathcal{M}^{\mathcal{G}}$ because more consistent intra-class neighbor distributions will lead to closer intra-class node representations after feature aggregation. This conclusion is also shown in Wang *et al.* [60], where the topological noise (which is inversely proportional to $h_S$) has a detrimental impact on node separability.

**Theorem 2.3.** The partial derivative of $\mathcal{J}_h^{\mathcal{G}}$ with respect to feature homophily $h_F$ satisfies,

$$\begin{cases} \frac{\partial \mathcal{J}_h^{\mathcal{G}}}{\partial h_F} < 0, & \text{if } h_L \in (0, h_L^-);\ h_L \in (h_L^-, h_L^+)\ \text{and}\ h_F \in (\hat{h}_F, 1) \\ \frac{\partial \mathcal{J}_h^{\mathcal{G}}}{\partial h_F} > 0, & \text{if } h_L \in (h_L^+, 1];\ h_L \in (h_L^-, h_L^+)\ \text{and}\ h_F \in (-1, \hat{h}_F) \\ \frac{\partial \mathcal{J}_h^{\mathcal{G}}}{\partial h_F} = 0, & \text{if } h_L = \frac{1}{C};\ h_L \in (h_L^-, h_L^+)\ \text{and}\ h_F = \hat{h}_F \end{cases} \tag{12}$$

where $0 < h_L^- < h_L^+ < 1$ and $-1 < \hat{h}_F < 1$. The expressions and detailed calculation of $h_L^-$, $h_L^+$, and $\hat{h}_F$ are shown in Appendix G.7.

From Theorem 2.3 we can see that, when $h_L$ is high in a graph, nodes with the same labels tend to be connected, thereby a larger $h_F$ makes the intra-class nodes share more similar representations and positively affect $\mathcal{M}^{\mathcal{G}}$; when $h_L$ is low in a graph, nodes with the different labels tend to be connected, thereby a larger $h_F$ makes the inter-class nodes share more similar representations and negatively affect $\mathcal{M}^{\mathcal{G}}$; when $h_L$ is on a medium level *i.e.*, $h_L \in (h_L^-, h_L^+)$, an increase of $h_F$ will first improve and then reduces the performance of $\mathcal{M}^{\mathcal{G}}$ with the cut-off point at $h_F = \hat{h}_F$. There the $\hat{h}_F$ is influenced by $h_L, h_S, C$, and $\rho(\boldsymbol{A})$, and the $h_L^-$ and $h_L^+$ are influenced by $h_S, C$, and $\rho(\boldsymbol{A})$.

**Remark**   Apart from the new findings mentioned above, Tri-Hom can help explain other interesting but under-explored phenomena of the graph homophily in previous studies, *e.g.,* 1) "good" or "bad" heterophily [46], which states that GNN can still perform well in some heterophily cases; and 2) feature shuffling [29], which states that shuffling the node features randomly within the same class can improves the performance of GNNs on node classification. Our explanations with Tri-Hom are: 1) The occurrence of "good" or "bad" heterophily is due to the fact that the model performance is influenced by a combination of $h_L, h_F$, and $h_S$, instead of $h_L$ alone. When the $h_L$ is low, the graph-aware models can still achieve good performance with a high $h_S$ or a low $h_F$; 2) feature shuffling is due to the existence of the structural-aware features of nodes. When $h_F > 0$, nodes are positively dependent on their neighbors. In this case, the nodes at the class boundaries or class centers are the hardest or easiest ones to predict because of the feature dependencies. If we randomly shuffle the nodes inside their classes, the nodes at the class boundaries will be easier to be classified because their features are more likely to be replaced by the nodes from the center, which are more distinguishable. For the nodes close to the class centers, it will be compensated by their neighbors.

## 5   Experimental Results

In Section 5.1, we conduct experiments on synthetic data generated by CSBM-3H to verify the conclusions from Theorem 2.1, 2.2, and 2.3, demonstrate the synergy of label $h_L$, structural $h_S$, and feature homophily $h_F$ and test whether Tri-Hom $\mathcal{J}_h^{\mathcal{G}}$ can reflect GNN performance. In Section 5.2, we evaluate the effectiveness of Tri-Hom on real-world benchmark datasets to test how well it can predict the model performance in real-world scenarios. In addition, we calculate the correlation between Tri-Hom value and model performance and compare them with the results of other 17 existing metrics. The results show that Tri-Hom has a significantly higher correlation with the model performance, demonstrating the necessity of filling the missing part by disentangling graph homophily from three aspects.

### 5.1   Experiments on Synthetic Datasets

To verify our theoretical results in a more general case, we measure the performance of GCN on synthetic datasets, where we can easily control $h_L, h_S$, and $h_F$. Specifically,

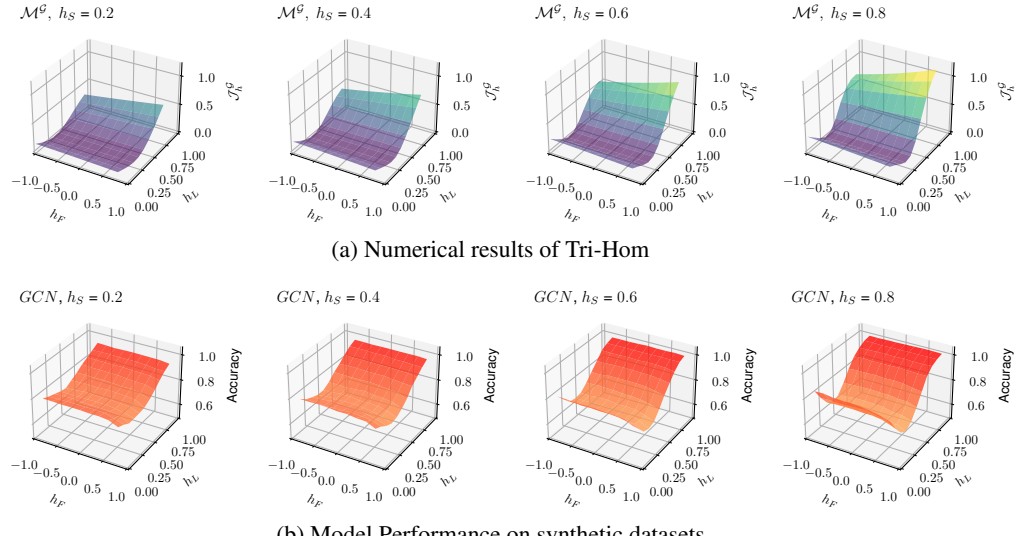

(a) Numerical results of Tri-Hom

(b) Model Performance on synthetic datasets

Figure 2: We measure the impact of label homophily $h_L$, feature homophily $h_F$, and structural homophily $h_S$ through numerical results of Tri-Hom $\mathcal{J}_h^{\mathcal{G}}$ and simulation results of the node classification accuracy with GCN on synthetic datasets.

**Synthetic Data Generation** We generate the synthetic graphs using CSBM-3H with a given tuple $(h_L, h_S, h_F)$, where $h_L \in \{0, 0.1, \ldots, 0.9, 1\}$, $h_S \in \{0, 0.1, \ldots, 0.9, 1\}$ and $h_F \in \{-0.8, 0.6, \ldots, 0.6, 0.8\}$. For each $(h_L, h_S, h_F)$, we generate 1000 nodes with three balanced classes and the node degrees are sampled from a uniform distribution $[1, 10]$. For node features, we first sample the structural-agnostic features from class-specific Gaussian distributions, then we use Eq. (4) to propagate these features across the graph topology with feature homophily $h_F$ to generate the observed structural-aware features as the final node features. Then, we evaluate the node classification performance of GCN [26] on these synthetic graphs. To get a robust evaluation and mitigate the numerical instability, we generate 10 graphs with 10 random seeds for each $(h_L, h_S, h_F)$ and report the average and standard deviation of classification accuracy on validation sets. The detailed process of topology and feature generation is shown in Appendix E.1.1.

**Numerical Results of Tri-Hom** To verify whether Tri-Hom $\mathcal{J}_h^{\mathcal{G}}$ can reflect the behavior of GCN, we calculate its numerical results with the same setting as the synthetic graphs and make a comparison. Specifically, we set $C = 3$ and $\rho(\boldsymbol{A}) = 10$ to mitigate the influences of varying numbers of classes and spectral radius.

**Comparison and Analysis** The results are shown in Figure 2. For a better demonstration, we only show the results for $h_S = \{0.2, 0.4, 0.6, 0.8\}$ and each subfigure is a slice of $h_S$ that visualizes the impact of $h_L$ and $h_F$ on GCN and Tri-Hom. From the comparison of Figure 2a and 2b, we have two main observations: 1) Overall, the impact of $h_L$, $h_S$, and $h_F$ on synthetic datasets aligns well with the numerical results of Tri-Hom $\mathcal{J}_h^{\mathcal{G}}$. The only difference is that $h_F$ seems to have less impact on GCN. We speculate that this is because the parameters in the graph filter in GCN are optimized during the training process, while these parameters are simplified as a fixed value in the theoretical analysis, leading to the difference[3]. 2) Our theoretical analysis of the impact of $h_L$, $h_F$ and $h_S$ on Tri-Hom is consistent with GCN's behavior in Figure 2b: Theorem 2.1 shows the worst performance of $\mathcal{M}^{\mathcal{G}}$ is reached when $h_L = \frac{1}{C}$, corresponding to the ravine in Figure 2b for $h_L$; Theorem 2.2 shows an increase of $h_S$ consistently improves the performance of $\mathcal{M}^{\mathcal{G}}$, corresponding to the overall increases of the accuracy in Figure 2b from the left to the right; Theorem 2.3 shows the influences of $h_F$ to $\mathcal{M}^{\mathcal{G}}$ is determined by the $h_L$. Even if this is not obvious in Figure 2b, we show more detailed figures of the individual impact of $h_L$, $h_S$, and $h_F$ in Appendix E.6, confirming the impact of $h_F$ in Theorem 2.3.

---

[3]We also show the detailed influences of $h_L$, $h_S$, and $h_F$ individually to MLP or GCN in Appendix E.6, which correlates well with our theoretical results of the homophily impact.

## 5.2 Experiments on Real-world Datasets

In this subsection, we show the superiority of our proposed Tri-Hom over the existing metrics by studying the correlation between the estimated metric values and model performance on real-world graph data.

**Experimental Settings**  To verify the effectiveness of our proposed Tri-Hom and compare with existing metrics, we train baseline models, MLP[4], GCN [26], GraphSage [16], GAT [57] and estimate metrics, $\mathcal{J}_h^{\neg\mathcal{G}}$, $\mathcal{J}_h^{\mathcal{G}}$, $h_F$, $h_S$, and $h_L$(we use $h_{node}$ in our experiments), on 31 real-world heterophilic and homophilic datasets. These datasets include *Roman-Empire, Amazon-Ratings, Mineweeper, Tolokers*, and *Questions* from [52]; *Squirrel, Chameleon, Actor, Texas, Cornell, Wisconsin* originally from [50, 53] and refined by [52]; *Cora, PubMed*, and *CiteSeer* from [69]; *CoraFull, Amazon-Photo, Amazon-Computer, Coauthor-CS*, and *Coauther-Physics* from [54]; *Flickr* from [70]; *WikiCS* from [48]; *Blog-Catalog* from [68]; *Ogbn-Arxiv* from [19]; *Genius, Twitch-DE, Twitch-ENGB, Twitch-ES, Twitch-FR, Twitch-PTBR, Twitch-RU*, and *Twitch-TW* from [32] [5]. Besides, we also calculate other graph homophily and performance metrics on the benchmark datasets, the metrics include label-based homophily: $h_{edge}$ [72], $h_{node}$ [50], $h_{class}$ [33], $h_{adj}$ [51], $h_{den}$ [30], $h_{2hop}$ [5], $h_{nei}$ [15]; structural-based homophily: $LI$ [51], $h_{NS}$ [46], $h_{agg}$ [38]; feature-based homophily: $h_{GE}$ [23], $h_{LS\text{-}cos}$ [7], $h_{LS\text{-}euc}$ [7], $h_{attr}$ [67], $h_{CF}$ [29]; and classifier-based homophily metrics [39]: $h_{KR}$, $h_{GNB}$, $h_{SVM}$ on these datasets. The detailed definitions of these metrics are summarized in Appendix A and the details of all estimations are shown in Appendix E.1.2.

| Metric | MLP | | GCN | | GraphSage | | GAT | | Rank |
|---|---|---|---|---|---|---|---|---|---|
| | Cor. | p-value | Cor. | p-value | Cor. | p-value | Cor. | p-value | |
| $h_{edge}$ | 0.4441 | 0.0123 | 0.5663 | 0.0009 | 0.4737 | 0.0071 | 0.5344 | 0.0020 | 6.25 |
| $h_{node}$ | 0.4232 | 0.0177 | 0.5457 | 0.0015 | 0.4524 | 0.0106 | 0.5257 | 0.0024 | 8.00 |
| $h_{class}$ | 0.6078 | 0.0003 | 0.6120 | 0.0003 | 0.5790 | 0.0006 | 0.6169 | 0.0002 | 2.50 |
| $h_{adj}$ | 0.4972 | 0.0044 | 0.5486 | 0.0014 | 0.4932 | 0.0048 | 0.5396 | 0.0017 | 5.25 |
| $h_{den}$ | 0.0038 | 0.9839 | 0.1483 | 0.4258 | 0.0525 | 0.7791 | 0.1258 | 0.5001 | 19.00 |
| $h_{2hop}$ | 0.4517 | 0.0107 | 0.5182 | 0.0028 | 0.4692 | 0.0078 | 0.4870 | 0.0055 | 7.50 |
| $h_{nei}$ | 0.3961 | 0.0274 | 0.4793 | 0.0064 | 0.4473 | 0.0116 | 0.4535 | 0.0104 | 10.75 |
| $LI$ | 0.4502 | 0.0110 | 0.4992 | 0.0043 | 0.4270 | 0.0166 | 0.4731 | 0.0072 | 9.75 |
| $h_{NS}$ | 0.2898 | 0.1139 | 0.3603 | 0.0465 | 0.3452 | 0.0572 | 0.3671 | 0.0422 | 14.00 |
| $h_{agg}$ | 0.5201 | 0.0027 | 0.5617 | 0.0010 | 0.6040 | 0.0003 | 0.5832 | 0.0006 | 3.75 |
| $h_S$ | 0.0981 | 0.5994 | 0.2345 | 0.2042 | 0.1981 | 0.2854 | 0.2886 | 0.1153 | 17.50 |
| $h_{GE}$ | 0.3641 | 0.0440 | 0.4501 | 0.0111 | 0.4347 | 0.0145 | 0.4094 | 0.0222 | 11.75 |
| $h_{LS\text{-}cos}$ | 0.3511 | 0.0528 | 0.4389 | 0.0135 | 0.4254 | 0.0170 | 0.4061 | 0.0234 | 13.00 |
| $h_{LS\text{-}euc}$ | 0.1272 | 0.4953 | 0.1101 | 0.5555 | 0.1117 | 0.5498 | 0.1168 | 0.5313 | 18.50 |
| $h_{attr}$ | 0.2022 | 0.2754 | 0.0990 | 0.5963 | 0.0735 | 0.6945 | 0.1121 | 0.5482 | 19.00 |
| $h_{CF}$ | 0.2549 | 0.1664 | 0.2890 | 0.1149 | 0.3154 | 0.0840 | 0.3167 | 0.0825 | 15.25 |
| $h_F$ | 0.4035 | 0.0244 | 0.4994 | 0.0042 | 0.4814 | 0.0061 | 0.4767 | 0.0067 | 8.50 |
| $h_{KR}$ | -0.5318 | 0.0021 | -0.3536 | 0.0510 | -0.3854 | 0.0323 | -0.3599 | 0.0468 | 22.00 |
| $h_{GNB}$ | -0.3796 | 0.0352 | -0.2440 | 0.1858 | -0.2828 | 0.1232 | -0.2421 | 0.1894 | 21.00 |
| $h_{SVM}$ | 0.2430 | 0.1878 | 0.2741 | 0.1356 | 0.3320 | 0.0681 | 0.2961 | 0.1058 | 15.75 |
| $\mathcal{J}_h^{\neg\mathcal{G}}$ | 0.5800 | 0.0006 | 0.6286 | 0.0002 | 0.5978 | 0.0004 | 0.6136 | 0.0002 | 2.50 |
| $\mathcal{J}_h^{\mathcal{G}}$ | 0.5471 | 0.0014 | 0.6650 | 0.0000 | 0.6223 | 0.0002 | 0.6731 | 0.0000 | 1.50 |

Table 1: Pearson correlation with p-value of all the metrics with model performance of node classification on 31 real-world datasets.

**Correlations with Model Performance**  To find out which metric can better align with the performance of GNNs on graphs with different properties, in Table 1, we show Pearson correlation between all the metrics and model performance on the 31 real-world datasets[6]. For example, to show

---

[4]We measure the performance of MLP to verify the effectiveness our proposed $\mathcal{J}_h^{\neg\mathcal{G}}$ and compare the performance gap between GNNs and MLP.

[5]See the dataset statistics, training details and model performance on node classification in Appendix E.1.2, E.2 and E.3, respectively.

[6]In addition to the Pearson correlation, we show Kendall's Tau rank correlation [4] in Appendix E.10. Besides, we show the correlations between the metrics with performance gaps between GNNs and MLP in Appendix E.9.

how well $h_{edge}$ align with the performance of GCN, we measure the values of $h_{edge}$ (in Table 3) and GCN performance (in Table 4) on node classification tasks across 31 datasets then calculate their correlation. For each model, the homophily metrics with the best, second, and third highest correlation values are highlighted in red, blue and purple, respectively. To get more robust comparison results, we rank the metrics for each model and report the average rank in the last column.

**Comparison and Analysis**    The results show that: 1) $\mathcal{J}_h^{\mathcal{G}}$ achieves the highest correlation values with all models, much better than the other metrics that only consider a single aspect of graph data. This confirms the effectiveness of Tri-Hom in filling the missing part of graph homophily by taking all the three types of homophily $h_L, h_S, h_F$ into account, providing a comprehensive understanding of graph homophily; 2) $\mathcal{J}_h^{\neg\mathcal{G}}$, the Tri-Hom for $\mathcal{M}^{\neg\mathcal{G}}$, also shows high correlation values. This indicates the existence of structural-aware features of nodes in graphs, which justifies our modeling of feature homophily in CSBM-3H. Compared with other variants of CSBM [46, 39], which assume node features are conditionally independent of graph topology given node labels, the setting of CSBM-3H is closer to real-world scenarios. This is one of the important reasons that the metrics derived from CSBM-3H are better than the existing metrics.

## 6    Conclusions

In this paper, we study the missing components of graph homophily by disentangling it from label, structural, and feature perspectives. Compared with previous homophily metrics, the combination of the three homophily metrics provides a unique and comprehensive understanding of graph homophily. Notably, our proposed feature homophily can measure the feature dependencies among nodes, fully disentangling itself from the label and structural homophily, which helps us analyze the disentangled impact. The theoretical study on CSBM-3H leads us to Tri-Hom, a combination of the three types of homophily. By investigating how each type influences Tri-Hom, we gain deeper insights into the effect of graph homophily on model performance and elucidate intriguing phenomena observed in previous studies [29, 60]. The synthetic experiments on CSBM-3H verify the theoretical results and the effectiveness of Tri-Hom. The high correlation with GNN behaviors on 31 real-world benchmark datasets confirms the superiority of Tri-Hom over 17 existing metrics.

## 7    Future Directions

In the future, it will be interesting to investigate the disentangled graph homophily in more general CSBM settings without some assumptions, *e.g.,* uniform node degrees, balanced class, and linear feature dependencies. Additionally, the estimation of three types of homophily in unsupervised or weak-supervised scenarios would be important for label-scarcity cases. Since the theoretical results in the paper reveal a nuanced understanding of graph homophily, we briefly introduce future studies on model designs or applications as follows.

For the model designs, previous studies [38, 50, 42] of heterophily-oriented GNNs mostly focus on label homophily, neglecting structural homophily $h_S$ and feature homophily $h_F$, that also influence GNN performance. For $h_S$, the proposed structural information function $\mathcal{S}(\cdot)$ could be replaced by any of the measurements regarding the task. Besides, the consistency of structural information could be different in the different classes, leading to the class-specific designs for each class. For $h_F$, it will be interesting to propose new approaches to identify how node features are influenced by their neighbors from both the global and local perspectives and then design new methods to treat the graphs or nodes differently according to $h_F$. Please refer to Appendix F.1 for more detailed explanations of the future directions on model designs

For the applications, such as social networks, recommendation systems, or urban computing, our proposed disentangled graph homophily could provide new insights. For example, in recommendation systems, the preferences of each user could be influenced by their neighbors. When people recognize or reject others' suggestions, it leads to a high $h_F$ or low $h_F$, causing them more likely or less likely to buy specific items. Please refer to Appendix F.2 for more details on the applications of the disentangled graph homophily.

## 8    Acknowledgements

Thanks to all the reviewers for their insightful comments. We also thank the organizers of NeurIPS for honoring us with the Scholar Award.

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

# A  Related Work on Homophily Measurements

In general, graph homophily metrics can be categorized as either statistic-based metrics or classifier-based metrics. The former can be further classified into the homophily on label, structural, or feature aspect. The definitions of these metrics are introduced as follows.

## A.1  Homophily on Label Aspect

**Edge homophily** [72] measures the graph homophily at the edge level, which is defined as the fraction of edges in a graph that connects nodes with the same labels:

$$h_{edge}(\mathcal{G}, \boldsymbol{Y}) = \frac{\left|\{e_{uv} \mid e_{uv} \in \mathcal{E}, Y_u = Y_v\}\right|}{|\mathcal{E}|} \tag{13}$$

**Node homophily** [50] measures the graph homophily at the node level, where the homophily degree for each node is computed as the proportion of the neighbors sharing the same class. Then the node homophily for the whole graph is defined as the average homophily degree for all the nodes:

$$h_{node}(\mathcal{G}, \boldsymbol{Y}) = \frac{1}{|\mathcal{V}|} \sum_{u \in \mathcal{V}} \frac{\left|\{v \mid v \in \mathcal{N}_u, Y_u = Y_v\}\right|}{d_u}, \tag{14}$$

**Class homophily** [33] addresses class imbalance by treating all classes equally. This metric mitigates the sensitivity of edge homophily and node homophily to the number of classes and nodes in each class. The definition of class homophily is given by:

$$h_{class}(\mathcal{G}, \boldsymbol{Y}) = \frac{1}{C-1} \sum_{c=1}^{C} \left[ \frac{\sum_{u \in \mathcal{V}, Y_u = c} \left|\{v \mid v \in \mathcal{N}_u, Y_u = Y_v\}\right|}{\sum_{u \in \{u \mid Y_u = c\}} d_u} - \frac{N_c}{N} \right]_+ \tag{15}$$

**Adjusted homophily** [51] considers the probability of an edge endpoint connecting to a node with a particular class and adjusts the edge homophily using node degrees, which is defined as:

$$h_{adj}(\mathcal{G}, \boldsymbol{Y}) = \frac{h_{edge}(\mathcal{G}, \boldsymbol{Y}) - \sum_{c=1}^{C} \frac{D_c^2}{(2|\mathcal{E}|)^2}}{1 - \sum_{c=1}^{C} \frac{D_c^2}{(2|\mathcal{E}|)^2}} \tag{16}$$

where $D_c$ represents the total degree of class $c$, *i.e.*, $D_k = \left|\{e_{uv} | Y_u = c \text{ or } Y_v = c, e_{uv} \in \mathcal{E}\}\right|$.

**Density-aware homophily** [30] is introduced as an improvement over class homophily, which only captures relative edge proportions and disregards graph connectivity. This limitation results in inflated homophily scores for highly disconnected graphs [51]. The proposed density-aware homophily aims to provide a more accurate measurement of edge density and is defined as:

$$h_{den}(\mathcal{G}, \boldsymbol{Y}) = \frac{1 + \min\left\{\zeta_c - \hat{\zeta}_c\right\}_{c=1}^{C}}{2} \tag{17}$$

where $\zeta_k$ is the edge density of the subgraph formed by intra-class edges of class $k$ and $\hat{\zeta}_k$ is the maximum intra-class edge density of class $k$.

**2-hop Neighbor Class Similarity** [5]. Since the information of 1-hop neighbors might be less representative or even misleading [5], 2-hop Neighbor Class Similarity extends the concept of "neighbors" from 1-hop to 2-hop, which is defined as:

$$h_{2hop}(\mathcal{G}, \boldsymbol{Y}) = \frac{1}{|\mathcal{V}|} \sum_{u \in \mathcal{V}} \frac{\left|\{u \mid u \in \mathcal{N}_u^{(2)}, Y_u = Y_v\}\right|}{d_u}, \tag{18}$$

where $\mathcal{N}_u^{(2)} = \{\bigcup_{v \in \mathcal{N}_u} \mathcal{N}_v\} \backslash \{u\}$ represents the two-hop neighbors of node $u$.

**Neighbor Homophily** [15] is proposed to address the "good" and "bad" heterophily issue by considering the dominant neighbors. The homophily score for any given node $u$ is based on the number

of nodes in which its class holds dominance among its k-hop neighbors. The definition of neighbor homophily in a graph is defined as:

$$h_{DN}(\mathcal{G}, \boldsymbol{Y}, k) = \frac{1}{|\mathcal{V}|} \sum_{u \in \mathcal{V}} \frac{\max \left\{ \left| v | v \in \mathcal{N}_u^{(k)} \right| \right\}_{c=1}^C}{\left| \mathcal{N}_u^{(k)} \right|} \tag{19}$$

where $\mathcal{N}_v^{(k)}$ represents the k-hop neighbors of node $u$.

## A.2 Homophily on Structure Aspect

**Label informativeness** [51] measures the informativeness of a neighbor's label for a node's label using condition entropy:

$$LI(\mathcal{G}, \boldsymbol{Y}) = -\frac{\sum_{c_1,c_2} p(c_1,c_2) \log \frac{p(c_1,c_2)}{\bar{p}(c_1),\bar{p}(c_2)}}{\sum_c \bar{p}(c) \log \bar{p}(c)} = 2 - \frac{\sum_{c_1,c_2} p(c_1,c_2) \log p(c_1,c_2)}{\sum_c \bar{p}(c) \log \bar{p}(c)} \tag{20}$$

where $p(c_1,c_2) = \sum_{(u,v) \in \mathcal{E}} \frac{\mathbb{1}\{Y_u=c_1, Y_v=c_2\}}{2|\mathcal{E}|}$ represents mutual distribution for a randomly sampled edge from class $c_1$ to class $c_2$ and $\bar{p}_c = \frac{D_c}{2|\mathcal{E}|}$ represents the distribution of the node degree for class $c$.

**Neighborhood Similarity** [46] measures the similarity of the neighbor distributions between two classes. A high similarity of intra-class neighbor distributions and a low similarity of inter-class neighbor distributions ensure the neighborhood patterns for nodes with different labels are distinguishable [46]. Therefore, the ratio of inter-class to intra-class neighborhood similarity could reflect the performance of GNNs[7]

$$h_{NS}(\mathcal{G}, \boldsymbol{D}^{\mathcal{N}}, \boldsymbol{Y}) = \frac{\mathbb{E}_{Y_u=Y_v}[D_u^{\mathcal{N}}(D_v^{\mathcal{N}})^T]}{\mathbb{E}_{Y_u \neq Y_v}[D_u^{\mathcal{N}}(D_v^{\mathcal{N}})^T]} \tag{21}$$

where $\mathcal{N}_v^{(k)}$ is the k-hop neighbors of node $u$.

**Aggregation homophily** [38] measures the ratio of nodes in a graph that has a higher intra-class aggregation similarity than inter-class aggregation similarity. The aggregation similarity of node $u$ and $u$ is the multiplication of their neighbor distribution *i.e.*, $D_u^{\mathcal{N}} D_v^{\mathcal{N}}$. The definition of aggregation homophily is given as:

$$h_{agg}(\mathcal{G}, \boldsymbol{D}^{\mathcal{N}}, \boldsymbol{Y}) = \frac{1}{|\mathcal{V}|} \left| \left\{ u \middle| \frac{\sum_{Y_u=Y_v} D_u^{\mathcal{N}}(D_v^{\mathcal{N}})^T}{|\{v|Y_u=Y_v\}|} \geq \frac{\sum_{Y_u \neq Y_v} D_u^{\mathcal{N}}(D_v^{\mathcal{N}})^T}{|\{v|Y_u \neq Y_v\}|}, v \in \mathcal{V}, u \in \mathcal{V} \right\} \right| \tag{22}$$

## A.3 Homophily on Feature Aspect

**Generalized edge homophily** [23] defines the feature homophily in graphs as the feature consistency across the graph topology:

$$h_{GE}(\mathcal{G}, \boldsymbol{X}) = \frac{1}{|\mathcal{E}|} \sum_{e_{uv} \in \mathcal{E}} \frac{\boldsymbol{X}_u \boldsymbol{X}_v^T}{\|\boldsymbol{X}_u\| \|\boldsymbol{X}_v\|} \tag{23}$$

The difference between this homophily with edge homophily is that generalized edge homophily replaced the indicator function of two connected nodes in edge homophily, *i.e.*, $\mathbb{1}\{Y_u=Y_v\}$, to a similarity measurement of node features, *i.e.*, $\text{sim}(\boldsymbol{X}_u, \boldsymbol{X}_v)$.

**Local Similarity** [7] measures feature homophily at the node level based on the hypothesis that nodes with similar features are likely to belong to the same class. The definition is given based on either cosine similarity

$$h_{LS-cos}(\mathcal{G}, \boldsymbol{X}) = \frac{1}{|\mathcal{V}|} \sum_{u \in \mathcal{V}} \frac{1}{d_u} \sum_{v \in \mathcal{N}_u} \frac{\boldsymbol{X}_u \boldsymbol{X}_v^T}{\|\boldsymbol{X}_u\| \|\boldsymbol{X}_v\|} \tag{24}$$

---

[7]This definition is not directly given in the original paper [46]. We define this ratio based on the proposition [46] that a high intra-class neighborhood similarity and a low inter-class neighborhood similarity improves GNN performance.

or Euclidean similarity.

$$h_{LS-euc}(\mathcal{G}, \boldsymbol{X}) = \frac{1}{|\mathcal{V}|} \sum_{u \in \mathcal{V}} \frac{1}{d_u} \sum_{v \in \mathcal{N}_u} \left( - \|\boldsymbol{X}_u - \boldsymbol{X}_v\|_2 \right) \tag{25}$$

**Attribute homophily** [67] considers the homophily with respect to each feature, which is defined as:

$$h_{attr,m}(\mathcal{G}, \boldsymbol{X}_{:,m}) = \frac{1}{\sum_{u \in \mathcal{V}} X_{u,m}} \sum_{u \in \mathcal{V}} \left( X_{u,m} \frac{\sum_{v \in \mathcal{N}_u} X_{v,m}}{d_u} \right)$$

$$h_{attr}(\mathcal{G}, \boldsymbol{X}) = \frac{1}{M} \sum_{m=1}^{M} h_{attr,m}(\mathcal{G}, \boldsymbol{X}_{:,m}), \tag{26}$$

where $h_{attr}$ represents the attribute homophily for the whole graph and $h_{attr,m}$ represents the attribute homophily for feature $m$.

**Class-controlled feature homophily** [29] considers the interplay between graph topology and feature dependence through the disparity of nodes' expected distances to random nodes with their neighbors, which is defined as:

$$h_{CF}(\mathcal{G}, \boldsymbol{X}, \boldsymbol{Y}) = \frac{1}{|\mathcal{V}|} \sum_{u \in \mathcal{V}} \frac{1}{d_u} \sum_{v \in \mathcal{N}_u} \left( \boldsymbol{d}(v, \mathcal{V} \backslash \{u\}) - \boldsymbol{d}(v, \{u\}) \right)$$

$$\boldsymbol{d}(u, \mathcal{V}') = \frac{1}{|\mathcal{V}'|} \sum_{v \in \mathcal{V}'} \|(\boldsymbol{X}_u | \boldsymbol{Y}) - (\boldsymbol{X}_v | \boldsymbol{Y})\| \tag{27}$$

$$\boldsymbol{X}_u | \boldsymbol{Y} = \boldsymbol{X}_u - \left( \frac{\sum_{Y_u = Y_v} \boldsymbol{X}_v}{|\{v | Y_u = Y_v, v \in \mathcal{V}\}|} \right)$$

where $\boldsymbol{X}_u | \boldsymbol{Y}$ represents class-controlled features and $\boldsymbol{d}(\cdot)$ denotes a distance function.

### A.4 Classifier-based homophily

**Classifier-based homophily** [39] uses a classifier to capture the feature-based linear or non-linear information without iterative training. To determine when graph-aware models perform better than graph-agnostic models, a hypothesis test is conducted on the original feature $\boldsymbol{X}$ and the aggregated features $\boldsymbol{H}$, as shown below:

$\text{H}_0 : \text{Prop(G-aware model)} \geq \text{Prop(G-agnostic model)}; \; \text{H}_1 : \text{Prop(G-aware model)} < \text{Prop(G-agnostic model)}$

The resulting p-value from this hypothesis test can indicate whether the performance of $\boldsymbol{H}$ is superior to that of $\boldsymbol{X}$. Three types of classifier, Gaussian Naive Bayes, Kernel Regression, and Support Vector Machine are used in [39], which correspond to the metrics $h_{GNB}$, $h_{KR}$, and $h_{SVM}$ in this paper.

## B Structural-Aware Node Features

This section explores the impact of graph topology on node features. Unlike the data in Euclidean space, where samples are *i.i.d.*, the data sampled from a graph for each node are structural-aware *i.e.,* $(\boldsymbol{X} \not\perp \boldsymbol{A} | \boldsymbol{Y})$. To model the feature dependencies of nodes in graphs, we follow [55] and adopt a graph diffusion process [6], which is shown as follows

$$\boldsymbol{X}(t) = \boldsymbol{X}(0) + \int_0^T \frac{\partial \boldsymbol{X}(t)}{\partial t} \, dt,$$

$$\text{where } \frac{\partial \boldsymbol{X}(t)}{\partial t} = (\mathcal{F}(\boldsymbol{A}) - \boldsymbol{I})\boldsymbol{X}(t-1) + \boldsymbol{X}(0) \tag{28}$$

Here the structural-agnostic node features $\boldsymbol{X}(0) \sim p(\boldsymbol{X} | \boldsymbol{Y})$ are sampled from the distributions with respect to each class, while the structural-aware node features $\boldsymbol{X}(t > 0) \sim p(\boldsymbol{X} | \boldsymbol{Y}, \boldsymbol{A})$ are generated through the diffusion process. In $\frac{\partial \boldsymbol{X}(t)}{\partial t}$, the first term describes how the dependencies are introduced with a feature dependency function $\mathcal{F} : (\boldsymbol{A}) \mapsto \mathbb{R}^{N \times N}$ and the second term $\boldsymbol{X}(0)$ preserves the node distinguishability. Then, based on the Eq. (28), we have

$$\boldsymbol{X}(t) = \boldsymbol{X}(0) + \mathcal{F}(\boldsymbol{A})\boldsymbol{X}(t-1) = \boldsymbol{X}(0) + (\mathcal{F}(\boldsymbol{A}))^1 \boldsymbol{X}(0) + \cdots + (\mathcal{F}(\boldsymbol{A}))^t \boldsymbol{X}(0) \tag{29}$$

We can also interpret the process as an interactive particle system [55] where all node features collapse into an equilibrium eventually. The equilibrium requires Eq. (29) to have a closed form when $t \to \infty$, implying the spectral radius of the adjacency matrix $\rho(\mathcal{F}(\boldsymbol{A})) < 1$. This constraint also implies that $|\mathcal{F}(\boldsymbol{A})|^k \geq |\mathcal{F}(\boldsymbol{A})|^{k-1}$ for $k > 0$, which aligns with the common relations in graphs that nodes are likely to have higher dependencies with their closer neighbors than with farther neighbors.

For simplicity and a better understanding of feature dependencies, we consider a linear case as in [6, 55] and use a parameter $\omega$ to control the feature dependencies *i.e.*,$\mathcal{F}(\boldsymbol{A}) = \omega \boldsymbol{A}$ with a range of $(-\frac{1}{\rho(\boldsymbol{A})}, \frac{1}{\rho(\boldsymbol{A})})$. Then we can represent the structural-aware features as

$$\boldsymbol{X} = \Big[\sum_{t=0}^{\infty}(\omega\boldsymbol{A})^t\Big]\boldsymbol{X} = (\boldsymbol{I} - \omega\boldsymbol{A})^{-1}\boldsymbol{X}(0) \tag{30}$$

## C  Properties of Feature Homophily

We further investigate properties of the estimation of feature homophily under specific feature transformations, including shifts, scaling, and changes in variance. The problem is defined as follows: In a graph $\mathcal{G} = \{\mathcal{V}, \mathcal{E}\}$ with $N$ nodes, there are node labels $\boldsymbol{Y}$ and node features $\boldsymbol{X}_{:,m}$ in dimension $m$. We can estimate the feature homophily $h_{F,m}^*$ for feature $\boldsymbol{X}_{:,m}$ in dimension $m$ as:

$$h_{F,m}^*(\mathcal{G}, \boldsymbol{X}_{:,m}, \boldsymbol{Y}) = \arg\min_{h_{F,m}} \sum_{\substack{u,v\in\mathcal{V}, \\ Y_u=Y_v}} [X_{u,m}(0) - X_{v,m}(0)]^2, \quad \text{where } \boldsymbol{X}_{:,m}(0) = \Big(\boldsymbol{I} - \frac{h_{F,m}}{\rho(\boldsymbol{A})}\boldsymbol{A}\Big)\boldsymbol{X}_{:,m}$$

$$\tag{31}$$

Then we prove the estimation of feature homophily is invariant to the operations of feature shifts, scaling, or changing variance.

### A. Shifts

Let's consider a shift of node features $\boldsymbol{X}_{:,m}$ by a constant vector $\boldsymbol{C}$:

$$\boldsymbol{X}'_{:,m} = \boldsymbol{X}_{:,m} + \boldsymbol{C} \tag{32}$$

Then we have structural-agnostic features as

$$
\begin{aligned}
\boldsymbol{X}'_{:,m}(0) &= \Big(\boldsymbol{I} - \frac{h_{F,m}}{\rho(\boldsymbol{A})}\boldsymbol{A}\Big)\boldsymbol{X}'_{:,m} \\
&= \Big(\boldsymbol{I} - \frac{h_{F,m}}{\rho(\boldsymbol{A})}\boldsymbol{A}\Big)(\boldsymbol{X}_{:,m} + \boldsymbol{C}) \\
&= \boldsymbol{X}_{:,m}(0) + \Big(\boldsymbol{I} - \frac{h_{F,m}}{\rho(\boldsymbol{A})}\boldsymbol{A}\Big)\boldsymbol{C}
\end{aligned}
\tag{33}
$$

Then a new estimation of $h'_{F,m}$ under this feature shift can be expressed as

$$
\begin{aligned}
h'_{F,m}(\mathcal{G}, \boldsymbol{X}'_{:,m}, \boldsymbol{Y}) &= \arg\min_{h_{F,m}} \sum_{\substack{u,v\in\mathcal{V}, \\ Y_u=Y_v}} \big[X'_{u,m}(0) - X'_{v,m}(0)\big]^2 \\
&= \arg\min_{h_{F,m}} \sum_{\substack{u,v\in\mathcal{V}, \\ Y_u=Y_v}} \Big[\Big(X_{u,m}(0) + \Big(\boldsymbol{I} - \frac{h_{F,m}}{\rho(\boldsymbol{A})}\boldsymbol{A}\Big)C_u\Big) \\
&\qquad\qquad - \Big(X_{v,m}(0) + \Big(\boldsymbol{I} - \frac{h_{F,m}}{\rho(\boldsymbol{A})}\boldsymbol{A}\Big)C_v\Big)\Big]^2 \\
&= \arg\min_{h_{F,m}} \sum_{\substack{u,v\in\mathcal{V}, \\ Y_u=Y_v}} \Big[(X_{u,m}(0) - X_{v,m}(0)) + \Big(\boldsymbol{I} - \frac{h_{F,m}}{\rho(\boldsymbol{A})}\boldsymbol{A}\Big)(C_u - C_v)\Big]^2
\end{aligned}
$$

$$\tag{34}$$

Since $\boldsymbol{C}$ is a constant vector, we have $C_u - C_v = 0$. Next, we have

$$
\begin{aligned}
h'_{F,m}(\mathcal{G}, \boldsymbol{X}'_{:,\boldsymbol{m}}, \boldsymbol{Y}) &= \arg\min_{h_{F,m}} \sum_{\substack{u,v \in \mathcal{V}, \\ Y_u = Y_v}} [(X_{u,m}(0) - X_{v,m}(0))]^2 \\
&= h^*_{F,m}(\mathcal{G}, \boldsymbol{X}_{:,\boldsymbol{m}}, \boldsymbol{Y})
\end{aligned}
\tag{35}
$$

Therefore, we proved the estimation of feature homophily is invariant to the operation of feature shifts.

**B. Scaling**

Let's consider scaling node features $\boldsymbol{X}_{:,\boldsymbol{m}}$ by a constant $\alpha$:

$$
\boldsymbol{X}'_{:,\boldsymbol{m}} = \alpha \boldsymbol{X}_{:,\boldsymbol{m}}
\tag{36}
$$

Then we have structural-agnostic features as

$$
\begin{aligned}
\boldsymbol{X}'_{:,\boldsymbol{m}}(0) &= \left( \boldsymbol{I} - \frac{h_{F,m}}{\rho(\boldsymbol{A})} \boldsymbol{A} \right) (\alpha \boldsymbol{X}'_{:,\boldsymbol{m}}) \\
&= \alpha \left( \boldsymbol{I} - \frac{h_{F,m}}{\rho(\boldsymbol{A})} \boldsymbol{A} \right) \boldsymbol{X}'_{:,\boldsymbol{m}} \\
&= \alpha \boldsymbol{X}_{:,\boldsymbol{m}}(0)
\end{aligned}
\tag{37}
$$

Then a new estimation of $h'_{F,m}$ under this feature shift can be expressed as

$$
\begin{aligned}
h'_{F,m}(\mathcal{G}, \boldsymbol{X}'_{:,\boldsymbol{m}}, \boldsymbol{Y}) &= \arg\min_{h_{F,m}} \sum_{\substack{u,v \in \mathcal{V}, \\ Y_u = Y_v}} \left[ X'_{u,m}(0) - X'_{v,m}(0) \right]^2 \\
&= \arg\min_{h_{F,m}} \sum_{\substack{u,v \in \mathcal{V}, \\ Y_u = Y_v}} \alpha^2 \left[ X_{u,m}(0) - X_{v,m}(0) \right]^2 \\
&= \arg\min_{h_{F,m}} \alpha^2 \sum_{\substack{u,v \in \mathcal{V}, \\ Y_u = Y_v}} \left[ X_{u,m}(0) - X_{v,m}(0) \right]^2
\end{aligned}
\tag{38}
$$

Since $\arg\min_x(\cdot)$ is invariant to the scaling, e.g. $\arg\min_x(cf(x)) = \arg\min_x(f(x))$, we have

$$
\begin{aligned}
h'_{F,m}(\mathcal{G}, \boldsymbol{X}'_{:,\boldsymbol{m}}, \boldsymbol{Y}) &= \arg\min_{h_{F,m}} \sum_{\substack{u,v \in \mathcal{V}, \\ Y_u = Y_v}} [(X_{u,m}(0) - X_{v,m}(0))]^2 \\
&= h^*_{F,m}(\mathcal{G}, \boldsymbol{X}_{:,\boldsymbol{m}}, \boldsymbol{Y})
\end{aligned}
\tag{39}
$$

Therefore, we proved the estimation of feature homophily is invariant to the operation of feature scaling.

**C. Variance Changing**

Changing the variance of $\boldsymbol{X}_{:,\boldsymbol{m}}$ can be seen as the combination of scaling and shifts. Assume node features follow a Gaussian distribution $N(\mu, \sigma^2)$, after the operation of changing the variance from $\sigma^2$ to $\beta\sigma^2$, we have new node features as

$$
\boldsymbol{X}'_{:,\boldsymbol{m}} = \sqrt{\beta}(\boldsymbol{X}_{:,\boldsymbol{m}} - \mu) + \mu
\tag{40}
$$

where $\boldsymbol{X}'_{:,\boldsymbol{m}}$ is calculated by deducing $\mu$, multiplying $\sqrt{\beta}$, and adding $\mu$. We already show the estimation of feature homophily is invariant to the operations of feature shifts and scaling. Since the operation of variance changing is a combination of scaling and shifts, we can conclude that the estimation of feature homophily is invariant to the variance changing.

# D  Impact of Graph Homophily on Graph-agnostic Models

In addition to the impact of label homophily $h_L$, structural homophily $h_S$, and feature homophily $h_F$ on Graph-aware models as discussed in Section 4.2, we further discuss the impact on Graph-agnostic models through $\mathcal{J}_h^{\neg\mathcal{G}}$ in this section. Specifically, we compute the partial derivative of $\mathcal{J}_h^{\neg\mathcal{G}}$ with respect to $h_S$, $h_F$, and $h_L$ to reveal the analytical influences. (See calculation in Appendix G.2, G.3, and G.4.)

**Theorem 3.1.** The partial derivative of $\mathcal{J}_h^{\neg\mathcal{G}}$ with respect to label homophily $h_L$ satisfies,

$$\frac{\partial \mathcal{J}_h^{\neg\mathcal{G}}}{\partial h_L} \begin{cases} < 0, & \text{if } h_F \in (-1, 0) \\ \geq 0, & \text{if } h_F \in [0, 1) \end{cases} \tag{41}$$

From Theorem 3.1 we can see that, under a positive $h_F$ *i.e.,*the features of connected nodes become similar, the increase of $h_L$ makes the features of intra-class nodes more distinguishable, thereby improving the performance of $\mathcal{M}^{\neg\mathcal{G}}$. Conversely, under a negative $h_F$ *i.e.,*the features of connected nodes become dissimilar, the increase of $h_L$ makes the features of intra-class nodes more indistinguishable, resulting in a degradation of the performance of $\mathcal{M}^{\neg\mathcal{G}}$.

**Theorem 3.2.** The partial derivative of $\mathcal{J}_h^{\neg\mathcal{G}}$ with respect to structural homophily $h_S$ satisfies,

$$\frac{\partial \mathcal{J}_h^{\neg\mathcal{G}}}{\partial h_S} \geq 0 \tag{42}$$

From Theorem 3.2 we can see that, since the existence of feature dependencies, a larger $h_S$ makes the structural-aware features more distinguishable among different classes, improving the performance of $\mathcal{M}^{\neg\mathcal{G}}$.

**Theorem 3.3.** The partial derivative of $\mathcal{J}_h^{\neg\mathcal{G}}$ with respect to feature homophily $h_F$ satisfies,

$$\frac{\partial \mathcal{J}_h^{\neg\mathcal{G}}}{\partial h_F} \begin{cases} < 0, & \text{if } h_L \in (0, h_L^-); h_L \in (h_L^-, h_L^+) \text{ and } h_F \in (\hat{h}_F, 1) \\ > 0, & \text{if } h_L \in (h_L^+, 1]; h_L \in (h_L^-, h_L^+) \text{ and } h_F \in (-1, \hat{h}_F) \\ = 0, & \text{if } h_L \in (h_L^-, h_L^+) \text{ and } h_F = \hat{h}_F \end{cases} \tag{43}$$

where $0 < h_L^{*,-} < h_L^{*,+} < 1$ and $-1 < h_F^* < 1$.

From Theorem 3.3 we can see, how the impact of $h_F$ on $\mathcal{M}^{\neg\mathcal{G}}$ is determined by $h_L$. This result is similar to the Theorem 2.3 in the case of $\mathcal{M}^{\mathcal{G}}$. The increase of $h_F$ makes node features more similar to their neighbors. As a result, under a high $h_L$, features of intra-class nodes become more similar, thereby improving the performance of $\mathcal{M}^{\neg\mathcal{G}}$; under a low $h_L$, features of intra-class nodes becomes more similar, thereby reducing the performance of $\mathcal{M}^{\neg\mathcal{G}}$.

# E  Experimental Details

## E.1  Datasets

### E.1.1  Synthetic Datasets

We show the detailed process of constructing CSBM-3H in Algorithm 1. First, graph topology is constructed with label homophily $h_L$ and structural homophily $h_S$. Then, structural-aware node features are constructed with feature homophily $h_F$. To investigate how three types of homophily influence the model performance, we generate graphs with $h_L \in [0, 0.1, \ldots, 1]$, $h_S \in [0, 0.1, \ldots, 1]$, and $h_F \in [-0.8, -0.6, \ldots, 0.8]$. For each given $h_L$, $h_S$, and $h_F$, we generate the graphs with 10 seeds from $[0, 1, \ldots, 9]$ to mitigate random deviations. Each graph contains 1000 nodes distributed across 3 classes, with node degrees sampled from a uniform distribution in the range

For the random graph generative models with homophily, current studies [51, 29, 39, 46] generally adopt a Contextual Stochastic Block Model with Homophily(CSBM-H) to control the label homophily $h_L$ through assigning nodes with different probabilities that connect to the nodes from other classes.

**Algorithm 1** Stochastic Block Model controlled with 3 types of Homophily(CSBM-3H)

---

**Input:** Label homophily $h_L$, structural homophily $h_S$, and feature homophily $h_F$,
  number of nodes $N$, number of classes $C$, node degree $\boldsymbol{D}$,
  dimension of node features $M$, random seed $\phi$,
  class-wised Gaussian mean $\boldsymbol{\mu} = [\boldsymbol{\mu_1}, \ldots, \boldsymbol{\mu_C}]^T$, covariance $\boldsymbol{\Sigma} = [\boldsymbol{\Sigma_1}, \ldots, \boldsymbol{\Sigma_C}]^T$ with $\boldsymbol{\mu_c} \in \mathcal{R}^M$
  and $\boldsymbol{\Sigma_c} \in \mathcal{R}^{M \times M}$ for class $c$.

**Output:** Adjacency matrix $\boldsymbol{A}$, node features $\boldsymbol{X}$ and node labels $\boldsymbol{Y}$ of a synthetic graph.
  Initialize node features $\boldsymbol{X}$, node labels $\boldsymbol{Y}$, with $\boldsymbol{X} \in \mathbb{R}^{N \times M}$, $\boldsymbol{Y} \in \mathbb{R}^N$. Set random seed $\phi$.
  **for** $n = 0$ to $N$ **do**                                            $\triangleright$ Sample node labels
    $k \sim \lfloor \text{Uniform}(0, C) \rfloor$
    $Y_n \leftarrow k$
  **end for**
  $\boldsymbol{Z} \leftarrow \text{One-hot}(\boldsymbol{Y})$                        $\triangleright$ One-hot encoding of node labels
  $\boldsymbol{S} \leftarrow h_L \boldsymbol{I}_C + \frac{1-h_L}{C-1}(\mathbf{1}_C - \boldsymbol{I}_C)$           $\triangleright$ Introduce label homophily $h_L$
  $\boldsymbol{D}^{\mathcal{N}} \leftarrow \boldsymbol{Z}\boldsymbol{S}$                      $\triangleright$ Construct neighbor distribution
  **for** $D_{u,c}^{\mathcal{N}} \in \boldsymbol{D}^{\mathcal{N}}$ **do**
    $\epsilon \sim \text{Normal}(0, \frac{(1-h_S)^2}{C-1})$           $\triangleright$ Introduce structural homophily $h_S$
    $D_{u,c}^{\mathcal{N}} \leftarrow D_{u,c}^{\mathcal{N}} + \epsilon$
  **end for**
  $\hat{\boldsymbol{D}} \leftarrow \text{diag}(\boldsymbol{D})$
  $\boldsymbol{A_p} \sim \frac{C}{N} \hat{\boldsymbol{D}}^{-\frac{1}{2}} \boldsymbol{D}^{\mathcal{N}} \hat{\boldsymbol{D}}^{-\frac{1}{2}} \boldsymbol{Z}^T$       $\triangleright$ Construct neighbor sampling matrix with node degrees
  $\boldsymbol{A_p} \leftarrow \max(0, \min(1, \boldsymbol{A_p}))$                   $\triangleright$ Bound with $[0, 1]$ before sampling
  $\boldsymbol{A} \leftarrow <\text{Sym} \circ \text{Binarize}> (\boldsymbol{A_p})$    $\triangleright$ Binarization sampling and symmetrize adjacency matrix
  **for** $n = 0$ to $N$ **do**                            $\triangleright$ Sample Structural-agnostic node features
    $\boldsymbol{X_{n,:}} \sim \text{Normal}(\boldsymbol{\mu}_{Y_n}, \boldsymbol{\Sigma}_{Y_n})$
  **end for**
  $\boldsymbol{X} \leftarrow (\boldsymbol{I}_C - \frac{h_F}{\sigma(\boldsymbol{A})}\boldsymbol{A})^{-2}\boldsymbol{X}$                  $\triangleright$ Introduce feature homophily $h_F$
  **return** $\boldsymbol{A}, \boldsymbol{X}, \boldsymbol{Y}$

---

Then the node features are sampled solely based on the classes. However, these random graph generative models have two drawbacks: First, the probabilities of nodes connecting to the nodes with different classes are uniform, which lacks diversity. Second, the sampled node features are independent with their structures *i.e.,* $(\boldsymbol{X} \perp\!\!\!\perp \boldsymbol{A}|\boldsymbol{Y})$, which is uncommon in real-world scenarios where interactions influence the attributes of connected nodes [61, 14, 59]. Our proposed CSBM-3H well address these drawbacks by considering $h_S$ and $h_F$, thereby providing a more comprehensive and realistic model.

### E.1.2 Real-World Datasets

We conduct our experiments on 31 real-world datasets: Roman-Empire, Amazon-Ratings, Mineweeper, Tolokers, and Questions from [52]; Squirrel, Chameleon, Actor, Texas, Cornell, Wisconsin originally from [50] and refined by [52]; Cora, PubMed, and CiteSeer from [69]; CoraFull, Amazon-Photo, Amazon-Computer, Coauthor-CS, and Coauther-Physics from [54]; Flickr from [70]; WikiCS from [48]; Blog-Catalog from [68]; Ogbn-Arxiv from [19]; Genius, Twitch-DE, Twitch-ENGB, Twitch-ES, Twitch-FR, Twitch-PTBR, Twitch-RU, and Twitch-TW from [32]. These datasets contain both the homophilic and heterophilic graphs that come from citation networks, webpage networks, purchase networks, image description networks, coauthor networks, actor networks, and social networks. The diversity of these datasets enables us to evaluate the model performance in a general case. For these datasets, we show basic statistics in Table 2 and the graph homophily metrics or model performance metrics in Table 3.

### E.2 Training Detail

For all the datasets, we randomly split the train, validation, and test set as 50%:25%:25% for 10 runs. We use the Adam optimizer [25] with a learning rate of 0.001. The maximum training epoch is set to 1000 with a patience of 40 for early stopping. To enhance performance, we incorporated

| Dataset | #Nodes | #Edges | #Features | #Classes | Average Degrees | Spectral Radius |
|---|---|---|---|---|---|---|
| Roman-Empire | 22,662 | 32,927 | 300 | 18 | 1.45 | 4.58 |
| Amazon-Ratings | 24,492 | 93,050 | 300 | 5 | 3.80 | 20.39 |
| Minesweeper | 10,000 | 39,402 | 7 | 2 | 3.94 | 7.99 |
| Tolokers | 11,758 | 519,000 | 10 | 2 | 44.14 | 392.36 |
| Questions | 48,921 | 153,540 | 301 | 2 | 3.14 | 95.31 |
| Squirrel | 2,223 | 46,998 | 2,089 | 5 | 21.14 | 206.02 |
| Chameleon | 890 | 8,854 | 2,325 | 5 | 9.95 | 78.05 |
| Actor | 7,600 | 26,659 | 932 | 5 | 3.51 | 37.37 |
| Texas | 183 | 279 | 1,703 | 5 | 1.52 | 10.98 |
| Cornell | 183 | 277 | 1,703 | 5 | 1.51 | 10.08 |
| Wisconsin | 251 | 450 | 1,703 | 5 | 1.79 | 11.88 |
| Cora | 2,708 | 10,556 | 1,433 | 7 | 3.90 | 14.39 |
| CoraFull | 19,793 | 126,842 | 8,710 | 70 | 6.41 | 25.63 |
| CiteSeer | 3,327 | 9,228 | 3,703 | 6 | 2.77 | 13.74 |
| PubMed | 19,717 | 88,651 | 500 | 3 | 4.50 | 23.24 |
| Flickr | 89,250 | 899,756 | 500 | 7 | 10.08 | 83.06 |
| Amazon-Photo | 7,650 | 238,162 | 745 | 8 | 31.13 | 122.54 |
| Amazon-Computer | 13,752 | 491,722 | 767 | 10 | 35.76 | 169.71 |
| Coauthor-CS | 18,333 | 163,788 | 6,805 | 15 | 8.93 | 24.60 |
| Coauthor-Physics | 34,493 | 495,924 | 8,415 | 5 | 14.38 | 51.18 |
| WikiCS | 11,701 | 431,726 | 300 | 10 | 36.90 | 149.77 |
| Blog-Catalog | 5,196 | 343,486 | 8,189 | 6 | 66.11 | 114.01 |
| Ogbn-Arxiv | 169,343 | 1,166,243 | 128 | 40 | 6.89 | 180.27 |
| Genius | 421,961 | 984,979 | 12 | 2 | 2.33 | 212.82 |
| Twitch-DE | 9,498 | 153,138 | 2,514 | 2 | 16.12 | 149.92 |
| Twitch-ENGB | 7,126 | 35,324 | 2,545 | 2 | 4.96 | 43.41 |
| Twitch-ES | 4,648 | 59,382 | 2,148 | 2 | 12.78 | 89.82 |
| Twitch-FR | 6,549 | 112,666 | 2,275 | 2 | 17.20 | 130.24 |
| Twitch-PTBR | 1,912 | 31,299 | 1,449 | 2 | 16.37 | 99.09 |
| Twitch-RU | 4,385 | 37,304 | 2,224 | 2 | 8.51 | 76.26 |
| Twitch-TW | 2,772 | 63,462 | 1,288 | 2 | 22.89 | 143.43 |

Table 2: Statistics on real-world datasets

skip connections [18] and layer normalization [2] in each layer. All models are trained on a single NVIDIA RTX A5000 GPU with 24GB memory. For hyperparameter tuning, we perform a grid search on the validation set. The search space included the following hyperparameters:

- Number of layers: {1, 2},
- Hidden dimension: {64, 128, 256},
- Dropout rate: {0.2, 0.4, 0.6, 0.8},
- Weight decay: {1e-3, 1e-4, 1e-5}.

### E.3 Node Classification Performance

Table 4 shows the averaged mean and stand deviation accuracy of node classification performance for MLP, GCN, GraphSage, and GAT across 10 runs. Notably, graph-aware models $\mathcal{M}^{\mathcal{G}}$ outperform graph-agnostic models $\mathcal{M}^{\neg\mathcal{G}}$ in most datasets. This phenomenon holds true for both homophilic datasets, such as Cora, Citeseer, and PubMed, and heterophilic datasets, such as Roman-empire, Chameleon-filtered, and Flickr. Consequently, relying solely on label homophily is insufficient to determine the performance of $\mathcal{M}^{\mathcal{G}}$, which aligns with previous studies [46, 39]

### E.4 Tri-Hom for Graph-agnostic Models

We show the numerical results of Tri-Hom $\mathcal{J}_h^{\neg\mathcal{G}}$ for graph-agnostic models $\mathcal{M}^{\neg\mathcal{G}}$ in Figure 3a, where each subfigure is a slicer of $h_S$ that visualizes the influences of $h_L$ and $h_F$ on $\mathcal{J}_h^{\mathcal{G}}$. For the impact of $h_L$, $h_F$, and $h_S$ individually, we can get the same conclusions as in Theorem 3.1, 3.2, and 3.3. To validate our theoretical results of $\mathcal{M}^{\neg\mathcal{G}}$ in a more general case, we further show the impact of three types of homophily with MLP on synthetic datasets in Figure 3b. The results correlate well with our

| Dataset | $h_{edge}$ | $h_{node}$ | $h_{class}$ | $h_{adj}$ | $h_{den}$ | $h_{2hop}$ | $h_{nei}$ | $LI$ | $h_{NS}$ | $h_{agg}$ | $h_S$ |
|---|---|---|---|---|---|---|---|---|---|---|---|
| Roman-Empire | 0.0469 | 0.0460 | 0.0208 | -0.0497 | 0.4994 | 0.0750 | 0.2925 | -0.6554 | 3.2151 | 0.5874 | 0.5271 |
| Amazon-Ratings | 0.3804 | 0.3757 | 0.1266 | 0.1386 | 0.5000 | 0.3686 | 0.5132 | -0.1462 | 1.7253 | 0.4191 | 0.5256 |
| Minesweeper | 0.6828 | 0.6829 | 0.0094 | 0.0095 | 0.5000 | 0.6809 | 0.7999 | 0.0200 | 2.1402 | 0.3937 | 0.7070 |
| Tolokers | 0.5945 | 0.6344 | 0.1801 | 0.0944 | 0.4984 | 0.6510 | 0.7390 | 0.0047 | 2.1232 | -0.3217 | 0.5451 |
| Questions | 0.8616 | 0.8980 | 0.0790 | 0.1552 | 0.4999 | 0.9097 | 0.9342 | 0.0007 | 23.2975 | -0.5480 | 0.5576 |
| Squirrel | 0.2072 | 0.1905 | 0.0398 | 0.0087 | 0.4907 | 0.2239 | 0.3191 | -0.0316 | 1.2125 | -0.1498 | 0.6030 |
| Chameleon | 0.2361 | 0.2441 | 0.0444 | 0.0276 | 0.4917 | 0.2739 | 0.4141 | -0.0620 | 1.1318 | 0.1685 | 0.5367 |
| Actor | 0.2167 | 0.2199 | 0.0064 | 0.0024 | 0.4999 | 0.2135 | 0.3091 | -0.2189 | 1.0833 | -0.1476 | 0.3841 |
| Texas | 0.0609 | 0.0567 | 0.0000 | -0.2822 | 0.4444 | 0.5429 | 0.6946 | -0.2338 | 4.3899 | 0.4863 | 0.5158 |
| Cornell | 0.1227 | 0.1110 | 0.0383 | -0.2526 | 0.4828 | 0.3989 | 0.5819 | -0.4002 | 2.5518 | 0.4098 | 0.3676 |
| Wisconsin | 0.1778 | 0.1552 | 0.0461 | -0.1909 | 0.4838 | 0.4239 | 0.5959 | -0.2842 | 3.0825 | 0.6574 | 0.4687 |
| Cora | 0.8100 | 0.8252 | 0.7657 | 0.7717 | 0.5015 | 0.7407 | 0.8118 | 0.3240 | 14.9687 | 0.8789 | 0.6164 |
| CoraFull | 0.5670 | 0.5861 | 0.4959 | 0.5552 | 0.5012 | 0.4383 | 0.5661 | 0.2895 | 44.7329 | 0.8626 | 0.6980 |
| CiteSeer | 0.7391 | 0.7166 | 0.6267 | 0.6673 | 0.5010 | 0.6849 | 0.8389 | -0.0571 | 6.4521 | 0.6495 | 0.3909 |
| PubMed | 0.8024 | 0.7924 | 0.6641 | 0.6836 | 0.5002 | 0.7435 | 0.8179 | 0.2525 | 3.9050 | 0.6707 | 0.3792 |
| Flickr | 0.3809 | 0.3221 | 0.0698 | 0.1758 | 0.5000 | 0.3362 | 0.4657 | 0.0130 | 2.9056 | 0.1328 | 0.6086 |
| Amazon-Photo | 0.8272 | 0.8493 | 0.7722 | 0.7845 | 0.5036 | 0.6576 | 0.7112 | 0.6373 | 22.0358 | 0.9312 | 0.7559 |
| Amazon-Computer | 0.7772 | 0.8017 | 0.7002 | 0.6809 | 0.5015 | 0.5656 | 0.6439 | 0.4962 | 19.7670 | 0.9236 | 0.7628 |
| Coauthor-CS | 0.8081 | 0.8320 | 0.7547 | 0.7846 | 0.5008 | 0.6862 | 0.7274 | 0.5292 | 43.5786 | 0.9474 | 0.7213 |
| Coauthor-Physics | 0.9314 | 0.9153 | 0.8474 | 0.8692 | 0.5005 | 0.8369 | 0.8655 | 0.6675 | 26.9190 | 0.9424 | 0.7330 |
| WikiCS | 0.6547 | 0.6774 | 0.5675 | 0.5786 | 0.5022 | 0.3710 | 0.4306 | 0.3340 | 9.3530 | 0.7431 | 0.6366 |
| Blog-Catalog | 0.4011 | 0.3914 | 0.2680 | 0.2722 | 0.5029 | 0.2061 | 0.2479 | 0.0704 | 1.5237 | 0.5277 | 0.6987 |
| Ogbn-Arxiv | 0.6778 | 0.6353 | 0.4211 | 0.6158 | 0.5001 | 0.5013 | 0.6251 | 0.4535 | 51.7619 | 0.7367 | 0.5939 |
| Genius | 0.6689 | 0.5087 | 0.0229 | 0.1432 | 0.5000 | 0.7216 | 0.8078 | 0.0025 | 2.9629 | 0.0000 | 0.0931 |
| Twitch-DE | 0.6322 | 0.5958 | 0.1394 | 0.1351 | 0.5001 | 0.5431 | 0.6568 | -0.0029 | 1.1644 | 0.2426 | 0.5364 |
| Twitch-ENGB | 0.5560 | 0.5452 | 0.0852 | 0.0823 | 0.5000 | 0.5235 | 0.6060 | -0.0489 | 1.0328 | 0.1485 | 0.3853 |
| Twitch-ES | 0.5800 | 0.6186 | 0.1468 | 0.1067 | 0.4995 | 0.5794 | 0.6712 | -0.0051 | 1.5146 | -0.4148 | 0.5700 |
| Twitch-FR | 0.5595 | 0.5739 | 0.0855 | 0.0856 | 0.5000 | 0.5349 | 0.6105 | -0.0081 | 1.1885 | -0.2628 | 0.6009 |
| Twitch-PTBR | 0.5708 | 0.5949 | 0.1196 | 0.1082 | 0.4996 | 0.5418 | 0.6213 | -0.0047 | 1.2712 | -0.3086 | 0.5801 |
| Twitch-RU | 0.6176 | 0.6383 | 0.0424 | 0.0296 | 0.4998 | 0.6276 | 0.7470 | -0.0086 | 1.7610 | -0.5097 | 0.5698 |
| Twitch-TW | 0.5332 | 0.5500 | 0.0339 | 0.0331 | 0.5000 | 0.5212 | 0.5884 | -0.0101 | 1.1133 | -0.2150 | 0.6248 |

| Dataset | $h_{GE}$ | $h_{LS-cos}$ | $h_{LS-euc}$ | $h_{attr}$ | $h_{CF}$ | $h_F$ | $h_{KR1}$ | $h_{GNB}$ | $h_{SVM}$ | $\mathcal{J}^{-\mathcal{G}}$ | $\mathcal{J}^{\mathcal{G}}$ |
|---|---|---|---|---|---|---|---|---|---|---|---|
| Roman-Empire | 0.0257 | 0.0255 | -8.0595 | 0.0585 | 0.7781 | 0.1300 | 0.0000 | 0.0000 | 0.0000 | 0.9887 | 0.1519 |
| Amazon-Ratings | 0.1146 | 0.1192 | -24.3967 | 0.6159 | 0.5976 | 1.0000 | 0.9992 | 0.0253 | 1.0000 | 1.0079 | 0.0080 |
| Minesweeper | 0.3330 | 0.3332 | -0.9430 | 0.1856 | 0.0930 | 0.5000 | 1.0000 | 1.0000 | 1.0000 | 1.0533 | 0.6042 |
| Tolokers | 0.8093 | 0.7705 | -0.2957 | 0.1099 | 0.2228 | 1.0000 | 0.9982 | 0.0000 | 0.0000 | 1.0768 | 0.2814 |
| Questions | 0.4267 | 0.5544 | -0.1423 | 0.0050 | 0.9060 | 0.7100 | 0.0000 | 0.6304 | 1.0000 | 1.1122 | 0.7210 |
| Squirrel | 0.0138 | 0.0219 | -0.5155 | 0.0007 | -0.4113 | 0.0000 | 1.0000 | 1.0000 | 0.0000 | 1.0000 | 0.0481 |
| Chameleon | 0.0141 | 0.0135 | -0.5673 | 0.0006 | 0.0819 | 0.0000 | 0.9893 | 1.0000 | 0.0000 | 1.0000 | 0.0272 |
| Actor | 0.1594 | 0.1566 | -0.6630 | 0.0011 | 0.0229 | 0.0000 | 0.0000 | 0.0000 | 0.0000 | 1.0000 | 0.0289 |
| Texas | 0.3487 | 0.3448 | -0.1400 | 0.0008 | -0.0266 | 0.0000 | 0.0000 | 0.0000 | 1.0000 | 1.0000 | 0.1415 |
| Cornell | 0.3322 | 0.3391 | -0.1455 | 0.0009 | 0.1393 | 0.0000 | 0.0000 | 0.0000 | 1.0000 | 1.0000 | 0.0782 |
| Wisconsin | 0.3414 | 0.3373 | -0.1444 | 0.0009 | 0.2975 | 0.0000 | 0.0000 | 0.0000 | 1.0000 | 1.0000 | 0.0553 |
| Cora | 0.1677 | 0.1780 | -0.3338 | 0.0066 | 0.1410 | 0.1500 | 1.0000 | 1.0000 | 1.0000 | 1.0216 | 0.6881 |
| CoraFull | 0.1447 | 0.1604 | -0.2263 | 0.0037 | 0.0052 | 0.0000 | 1.0000 | 1.0000 | 0.0000 | 1.0000 | 0.3069 |
| CiteSeer | 0.1906 | 0.2229 | -0.2208 | 0.0067 | 0.1487 | 0.7700 | 0.0000 | 1.0000 | 0.0000 | 1.0935 | 0.4090 |
| PubMed | 0.2719 | 0.2658 | -0.2384 | 0.0099 | 0.2326 | 1.0000 | 0.0000 | 0.0000 | 0.0000 | 1.1443 | 0.5139 |
| Flickr | 0.3815 | 0.4084 | -0.0950 | 0.0027 | 0.0924 | 0.3000 | 0.0000 | 0.0000 | 0.0000 | 0.9982 | 0.0007 |
| Amazon-Photo | 0.4878 | 0.4450 | -0.0843 | 0.0016 | 0.1062 | 0.9100 | 0.0000 | 0.0000 | 0.0034 | 1.1435 | 0.9500 |
| Amazon-Computer | 0.4897 | 0.4402 | -0.0829 | 0.0015 | 0.0337 | 0.0000 | 0.0000 | 0.0000 | 1.0000 | 1.1413 | 0.8955 |
| Coauthor-CS | 0.2944 | 0.3194 | -0.1763 | 0.0056 | 0.2355 | 0.4400 | 0.0000 | 0.0000 | 1.0000 | 1.0644 | 0.8288 |
| Coauthor-Physics | 0.3513 | 0.3723 | -0.2160 | 0.0069 | 0.3741 | 0.8800 | 0.0000 | 0.0000 | 1.0000 | 1.1704 | 1.0292 |
| WikiCS | 0.3342 | 0.3916 | -277.1482 | -0.0209 | 0.9998 | 1.0000 | 0.0000 | 0.0000 | 0.0000 | 1.0980 | 0.4899 |
| Blog-Catalog | 0.1223 | 0.1228 | -0.1818 | 0.0001 | -0.0430 | 0.0000 | 0.0000 | 0.0000 | 0.0000 | 1.0000 | 0.0249 |
| Ogbn-Arxiv | 0.8389 | 0.8649 | -0.0546 | 0.0028 | 0.2352 | 1.0000 | 0.0057 | 0.0000 | 0.0000 | 1.0974 | 0.4470 |
| Genius | 0.6656 | 0.5909 | -0.2162 | 0.1585 | 0.1898 | 0.7500 | 0.0001 | 0.0000 | 0.0000 | 1.0521 | 0.1059 |
| Twitch-DE | 0.1961 | 0.1919 | -0.2860 | 0.0006 | 0.0782 | 0.0000 | 0.0390 | 0.9831 | 0.0000 | 1.0000 | 0.3217 |
| Twitch-ENGB | 0.2101 | 0.2001 | -0.2836 | 0.0007 | 0.0232 | 0.0000 | 0.0571 | 0.9191 | 0.0231 | 1.0000 | 0.1350 |
| Twitch-ES | 0.2039 | 0.1976 | -0.2939 | 0.0007 | 0.0278 | 0.0000 | 0.3051 | 0.9999 | 0.0001 | 1.0000 | 0.2505 |
| Twitch-FR | 0.2251 | 0.2117 | -0.2850 | 0.0007 | 0.0716 | 0.0000 | 0.9968 | 0.2459 | 0.0000 | 1.0000 | 0.2304 |
| Twitch-PTBR | 0.2135 | 0.2030 | -0.2860 | 0.0009 | 0.0911 | 0.0000 | 0.3399 | 0.9971 | 0.0009 | 1.0000 | 0.2396 |
| Twitch-RU | 0.1925 | 0.1902 | -0.2829 | 0.0006 | 0.0256 | 0.0000 | 0.9293 | 0.0307 | 0.4450 | 1.0000 | 0.3195 |
| Twitch-TW | 0.1893 | 0.1914 | -0.3037 | 0.0010 | 0.0271 | 0.0000 | 0.3110 | 0.0705 | 0.0001 | 1.0000 | 0.1935 |

Table 3: Graph homophily metrics and graph performance metrics on real-world datasets

| Dataset | MLP | GCN | GraphSage | GAT |
|---|---|---|---|---|
| | Acc± Std | Acc± Std | Acc± Std | Acc± Std |
| Roman-Empire | 65.47±0.66 | 78.76±0.66 | 83.74±0.52 | 83.79±0.65 |
| Amazon-Ratings | 47.58±0.30 | 50.67±0.75 | 53.21±0.66 | 51.72±0.59 |
| Minesweeper | 51.06±0.92 | 90.16±0.59 | 90.74±0.57 | 90.55±0.67 |
| Tolokers | 73.52±0.90 | 84.55±0.65 | 83.21±0.49 | 84.28±0.58 |
| Questions | 71.36±0.96 | 76.35±0.93 | 76.84±0.97 | 77.83±0.75 |
| Squirrel | 40.12±1.55 | 40.24±1.58 | 40.37±1.83 | 40.06±3.12 |
| Chameleon | 41.47±3.38 | 43.10±2.73 | 41.88±4.32 | 41.70±5.99 |
| Actor | 35.97±1.08 | 35.10±1.05 | 35.52±0.92 | 35.51±0.88 |
| texas | 75.26±4.61 | 72.28±7.41 | 79.60±5.47 | 74.72±4.27 |
| Cornell | 72.43±4.90 | 67.84±5.90 | 73.24±4.31 | 69.19±5.73 |
| Wisconsin | 80.98±4.81 | 78.24±5.43 | 83.53±3.94 | 79.41±6.49 |
| Cora | 71.57±1.56 | 86.36±0.93 | 87.10±1.43 | 87.26±1.51 |
| CoraFull | 59.18±0.68 | 68.74±0.76 | 69.06±0.71 | 70.12±0.69 |
| CiteSeer | 72.36±1.30 | 76.44±0.79 | 77.07±1.09 | 76.96±1.12 |
| PubMed | 87.26±0.24 | 89.12±0.38 | 89.30±0.28 | 89.26±0.44 |
| Flickr | 46.83±0.33 | 52.80±0.43 | 52.41±0.35 | 53.52±0.37 |
| Amazon-Photo | 91.28±0.51 | 95.16±0.54 | 95.72±0.35 | 95.59±0.24 |
| Amazon-Computer | 83.99±0.59 | 91.58±0.61 | 91.17±0.56 | 91.75±0.54 |
| Coauthor-CS | 94.54±0.32 | 95.68±0.23 | 95.53±0.29 | 95.58±0.24 |
| Coauthor-Physics | 95.18±0.27 | 96.93±0.25 | 96.83±0.30 | 96.73±0.28 |
| WikiCS | 81.38±0.51 | 85.20±0.28 | 85.71±0.47 | 85.89±0.48 |
| Blog-Catalog | 93.95±0.70 | 96.07±0.57 | 96.49±0.64 | 96.02±0.54 |
| Ogbn-Arxiv | 58.02±0.30 | 73.91±0.21 | 73.39±0.25 | 73.82±0.14 |
| Genius | 86.53±0.07 | 90.59±0.22 | 90.99±0.17 | 81.92±4.75 |
| Twitch-DE | 67.65±0.93 | 72.26±1.00 | 70.14±0.85 | 72.46±1.14 |
| Twitch-ENGB | 61.47±1.11 | 62.89±0.95 | 61.96±1.55 | 62.50±1.08 |
| Twitch-ES | 61.59±1.84 | 65.58±1.64 | 62.80±1.58 | 66.71±2.19 |
| Twitch-FR | 60.93±1.64 | 64.84±1.79 | 61.82±1.94 | 64.77±2.29 |
| Twitch-PTBR | 63.61±2.74 | 66.90±1.58 | 65.05±1.98 | 67.60±1.77 |
| Twitch-RU | 50.34±1.94 | 53.74±3.21 | 50.88±1.51 | 53.00±2.50 |
| Twitch-TW | 59.84±2.16 | 61.89±2.25 | 60.68±1.79 | 63.73±1.84 |

Table 4: Node classification performance on real-world datasets.

numerical results, showing the effectiveness of $\mathcal{J}_h^{\neg\mathcal{G}}$ in measuring the performance of $\mathcal{M}^{\neg\mathcal{G}}$ under the influences of three types of homophily.

### E.5 Numerical Results of Tri-Hom with Three types of Homophily

We show how the label homophily $h_L$, structural homophily $h_S$, and feature homophily $h_F$ collectively influence the $\mathcal{J}_h^{\neg\mathcal{G}}$ and $\mathcal{J}_h^{\mathcal{G}}$ in Figure 4 in a more general case, where the x-axis, y-axis, and z-axis denotes $h_L$, $h_F$, and $h_N$. A lighter color (close to yellow) indicates a larger $\mathcal{J}_h^{\neg\mathcal{G}}$ or $\mathcal{J}_h^{\mathcal{G}}$, which also implies a better model performance. These numerical results align well with our theoretical results of the individual impact on $\mathcal{J}_h^{\neg\mathcal{G}}$ or $\mathcal{J}_h^{\mathcal{G}}$ with respect to $h_L$, $h_S$, or $h_F$. Furthermore, by observing the gradients in Figure 4, we can know how these three homophily metrics influence the model performance collectively.

### E.6 Influences of A Single Type of Homophily on Synthetic Datasets

To verify our theoretical results, we show how the performances of GCN and MLP are affected by three types of homophily individually in Figure 5, which shows the influences of label homophily $h_L$, structural homophily $h_S$, and feature homophily $h_F$ on the node classification accuracy of GCN and MLP on synthetic datasets. First, the top 3 subfigures show the influences of $h_L$ under different $h_F$ and $h_S$. With the increase of $h_L$, the accuracy of GCN first decreases and then increases with a pitfall on a medium-level of homophily, and the accuracy of MLP is dependent on the sign of $h_F$, leading to

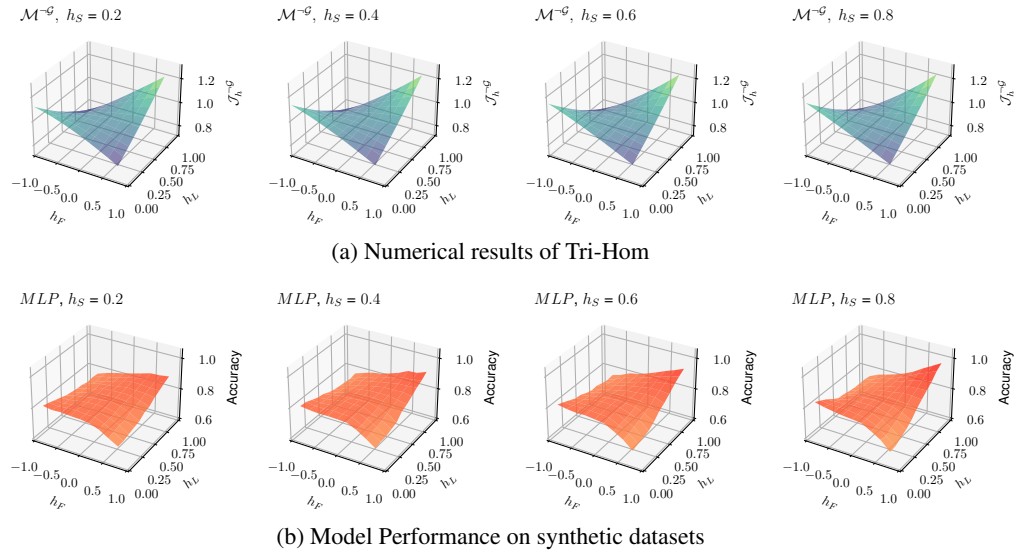

(a) Numerical results of Tri-Hom

(b) Model Performance on synthetic datasets

Figure 3: We measure the impact of label homophily $h_L$, feature homophily $h_F$, and structural homophily $h_S$ through numerical results of Tri-Hom $\mathcal{J}_h^{\neg\mathcal{G}}$ and simulation results of the node classification accuracy with MLP on synthetic datasets.

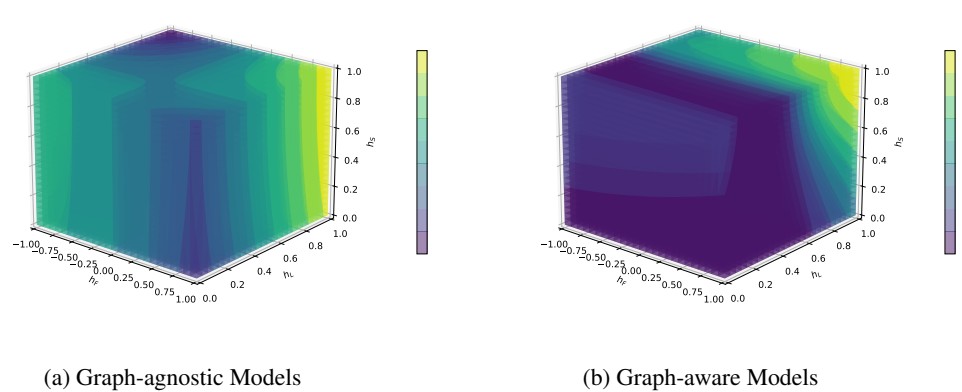

(a) Graph-agnostic Models          (b) Graph-aware Models

Figure 4: The Influences of label homophily $h_L$, structural homophily $h_S$, and feature homophily $h_F$ to graph-agnostic models and graph-aware models

the same results as in Theorem 2.1 and 3.1. Second, the medium 3 subfigures show the influences of $h_F$. With the increases of $h_F$, the accuracy of both of the GCN and MLP will decreases with a low $h_L$, increases with a high $h_L$, and increases first and then decreases with a medium $h_L$. The result also aligns well with Theorem 2.3 and 3.3. Last, the bottom 3 subfigures show the influences of $h_S$. Even if the influences under different $h_L$ or $h_F$ would slightly vary, an increase of $h_S$ generally improves the performance of both the GCN and MLP, aligning well with Theorem 2.2 and 3.2.

### E.7 Visualization of Real-world Dataset with Three Types of Homophily

To better visualize the impact of $h_L$, $h_S$, and $h_F$ to graph-aware models $\mathcal{M}^{\mathcal{G}}$ on real-world datasets, we show the performance of GCN on all the datasets with these homophily metrics in Figure 6, where $h_L$, $h_F$, and $h_S$ are shown as the x-axis, y-axis, and the size of the scatter respectively. Generally, the correlation of three homophily metrics with the performance of GCN is similar to our theoretical results in Figure 2a. With the increase of $h_L$, the performance of GCN decreases first and then

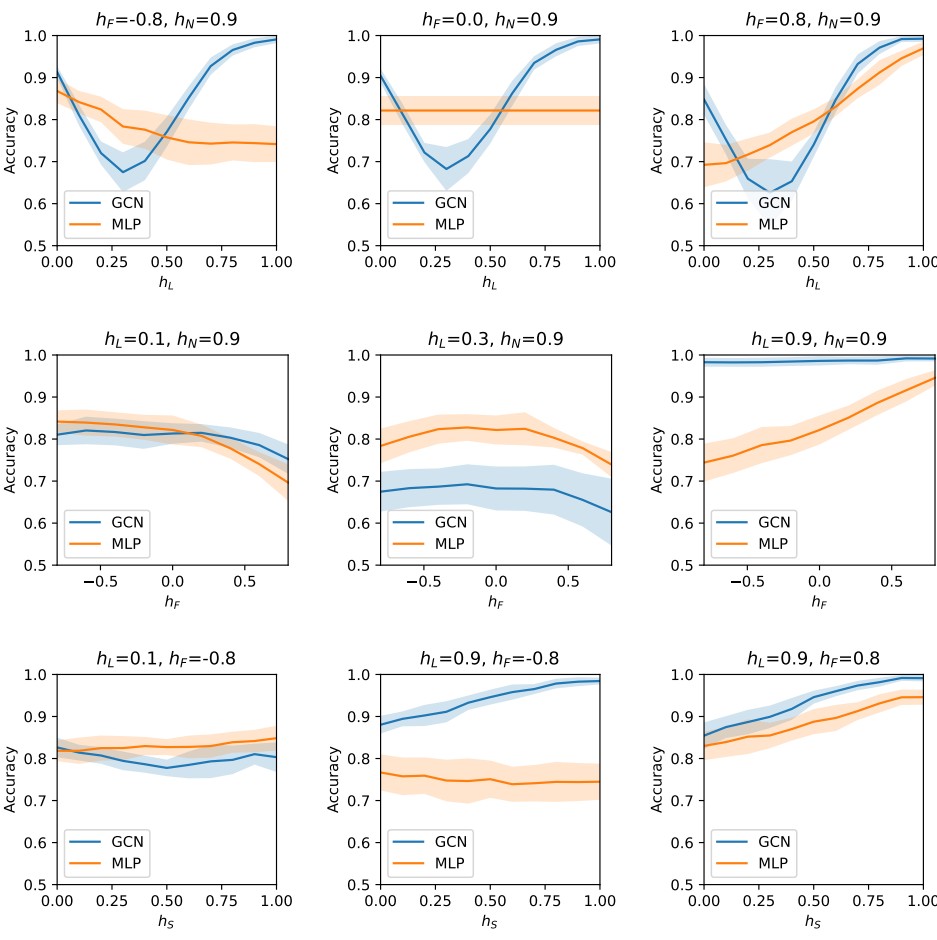

Figure 5: The impact of label homophily $h_L$, feature homophily $h_F$, and structural homophily $h_S$ on the accuracy of node classification using MLP and GCN.

increases, with a pitfall (as shown by datasets of Actor, Squirrel, and Chameleon). We also see that a higher $h_S$ leads to a better performance when we fix $h_F$ and $h_L$ (as shown by datasets of Question with Amazon-photo, and Cora with Coauthor-CS). As for the $h_F$, since its influence is not so obvious compared with $h_L$ and $h_S$ as shown in Figure 5 and the scarcity of real-world datasets, it is hard to see its influence in real-world datasets.

## E.8   Influences of Class-wised Structural Homophily on Real-world Datasets

As shown in Figure 7, we investigate more nuanced influences of structural homophily $h_S$ without the interference of $h_L$ and $h_F$ on real-world datasets with respect to each class. Specifically, we calculate node classification accuracy with respect to each class inside one dataset and show the alignment of the class-wised accuracy and class-wised structural homophily $h_S$. The results show for some of the datasets such as Citeseer, Amazon-ratings, and Coauthor-physics, we can clearly observe the accuracy increases with $h_S$ while for datasets such as Corafull and Coauthor-CS, we can only see the general tendency of alignment with noises. We speculate that for Corafull and Coauthor-CS, the $h_S$ is not as uniform as other datasets for the nodes in one class, leading to the noises.

## E.9   Correlation of Metrics with Performance Gap of GNNs and MLP

To investigate when GNNs are better than MLP, we show the differences in the performance between GNNs and MLP in Figure 5. The results indicate that, among the statistic-based homophily metrics,

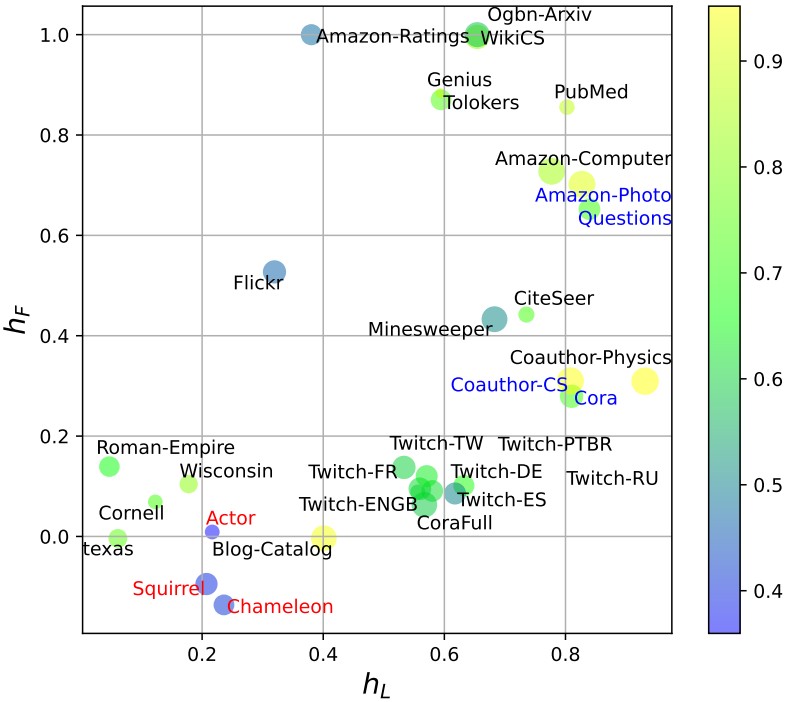

Figure 6: Label, feature, and structural homophily metrics on real-world datasets are shown as the x-axis, y-axis, and the size of the scatter respectively. The classification performance of GCN is denoted by the color of the scatters.

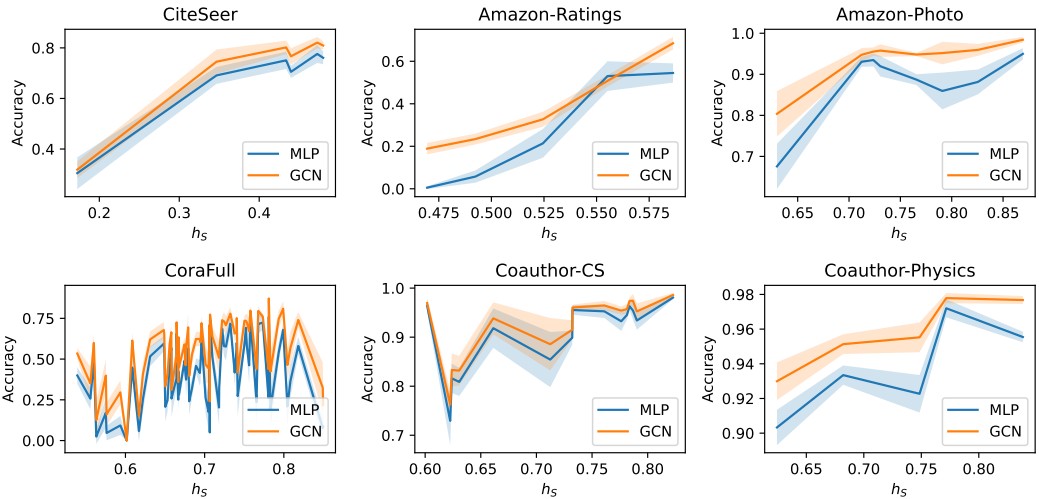

Figure 7: The influences of structural homophily to both the GCN and MLP for each class on real-world datasets.

| Metric | GCN-MLP | | GraphSage-MLP | | GAT-MLP | | Rank |
|--------|---------|---------|---------------|---------|---------|---------|------|
| | Cor. | p-value | Cor. | p-value | Cor. | p-value | |
| $h_{edge}$ | 0.2937 | 0.1087 | 0.1189 | 0.5242 | 0.2073 | 0.2632 | 8.00 |
| $h_{node}$ | 0.2930 | 0.1097 | 0.1155 | 0.5362 | 0.2316 | 0.2099 | 8.33 |
| $h_{class}$ | 0.0454 | 0.8084 | 0.0075 | 0.9678 | 0.0459 | 0.8064 | 19.67 |
| $h_{adj}$ | 0.1420 | 0.4461 | 0.0500 | 0.7896 | 0.1101 | 0.5555 | 16.00 |
| $h_{den}$ | 0.3164 | 0.0829 | 0.1096 | 0.5572 | 0.2537 | 0.1685 | 7.33 |
| $h_{2hop}$ | 0.1722 | 0.3543 | 0.0925 | 0.6208 | 0.0932 | 0.6178 | 15.67 |
| $h_{nei}$ | 0.2055 | 0.2675 | 0.1615 | 0.3854 | 0.1367 | 0.4632 | 11.33 |
| $LI$ | 0.1338 | 0.4729 | 0.0012 | 0.9948 | 0.0674 | 0.7186 | 18.33 |
| $h_{NS}$ | 0.1716 | 0.3561 | 0.1585 | 0.3945 | 0.1736 | 0.3503 | 11.00 |
| $h_{agg}$ | 0.1220 | 0.5132 | 0.2497 | 0.1755 | 0.1542 | 0.4076 | 9.67 |
| $h_S$ | 0.3040 | 0.0964 | 0.2356 | 0.2020 | 0.4001 | 0.0257 | 2.33 |
| $h_{GE}$ | 0.2096 | 0.2577 | 0.2011 | 0.2779 | 0.1102 | 0.5550 | 11.00 |
| $h_{LS-cos}$ | 0.2130 | 0.2500 | 0.2080 | 0.2614 | 0.1299 | 0.4862 | 9.67 |
| $h_{LS-euc}$ | -0.0299 | 0.8733 | -0.0197 | 0.9160 | -0.0159 | 0.9323 | 21.00 |
| $h_{attr}$ | -0.2137 | 0.2484 | -0.2644 | 0.1506 | -0.1781 | 0.3376 | 22.00 |
| $h_{CF}$ | 0.0897 | 0.6313 | 0.1657 | 0.3730 | 0.1398 | 0.4531 | 13.00 |
| $h_F$ | 0.2338 | 0.2056 | 0.2224 | 0.2292 | 0.1701 | 0.3604 | 7.67 |
| $h_{KR}$ | 0.3582 | 0.0478 | 0.2650 | 0.1497 | 0.3336 | 0.0667 | 1.33 |
| $h_{GNB}$ | 0.2740 | 0.1357 | 0.1721 | 0.3546 | 0.2689 | 0.1435 | 6.67 |
| $h_{SVM}$ | 0.0825 | 0.6590 | 0.2281 | 0.2171 | 0.1211 | 0.5162 | 12.67 |
| $\mathcal{J}^{\neg\mathcal{G}}$ | 0.1407 | 0.4503 | 0.1086 | 0.5610 | 0.0954 | 0.6096 | 16.00 |
| $\mathcal{J}^{\mathcal{G}}$ | 0.2903 | 0.1131 | 0.2333 | 0.2065 | 0.2860 | 0.1188 | 4.33 |

Table 5: Pearson correlation with p-value of all the metrics with the performance differences of GNNs with MLP of node classification on real-world datasets.

the structural homophily $h_S$ shows a strong correlation with the differences in the performance between GNNs and MLP. This is because $h_S$ measure the consistency of the structural information of nodes within the same class, thereby reflecting the differences in the performance between GNNs and MLP. We also observe that the $h_{KR}$ has the highest correlation with differences, confirming the effectiveness of the classifier-based metrics in measuring the differences in the performance between GNNs and MLP [39].

### E.10  Other Types of Correlation

In our experiments on real-world datasets, we explore the correlation between all the metrics and model performance. In addition to the widely used Pearson correlation, we employ Kendall's Tau rank correlation to assess these relationships. Let $\boldsymbol{x}$ and $\boldsymbol{y}$ be two observed variables with $\boldsymbol{x}, \boldsymbol{y} \in \mathbb{R}^N$, the Pearson correlation $\rho$ can be calculated as

$$\rho = \frac{\sum_{i=1}^{N}(x_i - \bar{x})(y_i - \bar{y})}{\sqrt{\sum_{i=1}^{N}(x_i - \bar{x})^2 \sum_{i=1}^{N}(y_i - \bar{y})^2}} \tag{44}$$

where $\bar{x}$ and $\bar{y}$ are the means of $\boldsymbol{x}$ and $\boldsymbol{y}$ respectively.

Kendall's Tau rank correlation [4] $\tau$ can be calculated as

$$\tau = \frac{|\text{concordant pairs}| - |\text{discordant pairs}|}{\frac{1}{2}N(N-1)} \tag{45}$$

where concordant pairs occur when the ranks of both variables agree, while discordant pairs occur when they disagree.

Pearson correlation measures linear correlation between two variables, which is widely used and easy to interpret. Kendall's Tau rank correlation measures the similarity ranking between two variables, which has no assumptions of the data distribution and are robust to outliers.

| Metric | MLP | | GCN | | GraphSage | | GAT | | GCN-MLP | | GraphSage-MLP | | GAT-MLP | |
|---|---|---|---|---|---|---|---|---|---|---|---|---|---|---|
| | Cor. | p-value | Cor. | p-value | Cor. | p-value | Cor. | p-value | Cor. | p-value | Cor. | p-value | Cor. | p-value |
| $h_{edge}$ | 0.3677 | 0.0033 | 0.4882 | 0.0001 | 0.4452 | 0.0003 | 0.4839 | 0.0001 | 0.3462 | 0.0058 | 0.2129 | 0.0960 | 0.2731 | 0.0314 |
| $h_{node}$ | 0.3204 | 0.0110 | 0.4409 | 0.0004 | 0.4065 | 0.0011 | 0.4624 | 0.0002 | 0.3161 | 0.0122 | 0.1828 | 0.1545 | 0.3118 | 0.0135 |
| $h_{class}$ | 0.3935 | 0.0016 | 0.4194 | 0.0007 | 0.4022 | 0.0012 | 0.4237 | 0.0006 | 0.1570 | 0.2231 | 0.1785 | 0.1647 | 0.2387 | 0.0611 |
| $h_{adj}$ | 0.3892 | 0.0018 | 0.4667 | 0.0001 | 0.4323 | 0.0005 | 0.4538 | 0.0002 | 0.2731 | 0.0314 | 0.2602 | 0.0407 | 0.2602 | 0.0407 |
| $h_{den}$ | 0.2430 | 0.0565 | 0.3634 | 0.0037 | 0.3548 | 0.0047 | 0.3677 | 0.0033 | 0.2559 | 0.0442 | 0.2258 | 0.0770 | 0.2430 | 0.0565 |
| $h_{2hop}$ | 0.3892 | 0.0018 | 0.4065 | 0.0011 | 0.3720 | 0.0029 | 0.3935 | 0.0016 | 0.1785 | 0.1647 | 0.1742 | 0.1754 | 0.0968 | 0.4578 |
| $h_{nei}$ | 0.3376 | 0.0073 | 0.3720 | 0.0029 | 0.3462 | 0.0058 | 0.3591 | 0.0042 | 0.1871 | 0.1448 | 0.2086 | 0.1031 | 0.1054 | 0.4178 |
| $LI$ | 0.3462 | 0.0058 | 0.4753 | 0.0001 | 0.4409 | 0.0004 | 0.5054 | 0.0000 | 0.3505 | 0.0052 | 0.2774 | 0.0287 | 0.3204 | 0.0110 |
| $h_{NS}$ | 0.3720 | 0.0029 | 0.4581 | 0.0002 | 0.4409 | 0.0004 | 0.4710 | 0.0001 | 0.2473 | 0.0521 | 0.4237 | 0.0006 | 0.2258 | 0.0770 |
| $h_{agg}$ | 0.3849 | 0.0020 | 0.4452 | 0.0003 | 0.4624 | 0.0002 | 0.4495 | 0.0003 | 0.0022 | 1.0000 | 0.2129 | 0.0960 | 0.0323 | 0.8135 |
| $h_S$ | 0.0667 | 0.6130 | 0.2387 | 0.0611 | 0.2129 | 0.0960 | 0.2774 | 0.0287 | 0.2172 | 0.0893 | 0.1613 | 0.2104 | 0.3075 | 0.0149 |
| $h_{GE}$ | 0.3376 | 0.0073 | 0.3376 | 0.0073 | 0.3290 | 0.0090 | 0.3247 | 0.0100 | 0.1441 | 0.2644 | 0.2860 | 0.0240 | 0.0624 | 0.6369 |
| $h_{LS-cos}$ | 0.3677 | 0.0033 | 0.3591 | 0.0042 | 0.3505 | 0.0052 | 0.3462 | 0.0058 | 0.1570 | 0.2231 | 0.3075 | 0.0149 | 0.0753 | 0.5664 |
| $h_{LS-euc}$ | -0.2989 | 0.0181 | -0.2387 | 0.0611 | -0.2559 | 0.0442 | -0.2172 | 0.0893 | 0.0409 | 0.7616 | -0.1527 | 0.2363 | 0.0968 | 0.4578 |
| $h_{attr}$ | -0.1183 | 0.3617 | -0.2903 | 0.0218 | -0.2817 | 0.0263 | -0.2688 | 0.0343 | -0.2688 | 0.0343 | -0.4366 | 0.0004 | -0.2387 | 0.0611 |
| $h_{CF}$ | 0.2860 | 0.0240 | 0.2946 | 0.0199 | 0.3032 | 0.0164 | 0.3075 | 0.0149 | 0.1699 | 0.1865 | 0.3118 | 0.0135 | 0.1742 | 0.1754 |
| $h_F$ | 0.2871 | 0.0365 | 0.3920 | 0.0043 | 0.3870 | 0.0048 | 0.3920 | 0.0043 | 0.3071 | 0.0253 | 0.4169 | 0.0024 | 0.2821 | 0.0399 |
| $h_{KR}$ | -0.4186 | 0.0010 | -0.3236 | 0.0107 | -0.3495 | 0.0059 | -0.3279 | 0.0097 | 0.2416 | 0.0568 | 0.0734 | 0.5631 | 0.2201 | 0.0828 |
| $h_{GNB}$ | -0.2929 | 0.0222 | -0.2317 | 0.0704 | -0.2580 | 0.0440 | -0.2361 | 0.0653 | 0.1093 | 0.3935 | -0.0874 | 0.4948 | 0.1487 | 0.2458 |
| $h_{SVM}$ | 0.1531 | 0.2364 | 0.1176 | 0.3631 | 0.1398 | 0.2797 | 0.1309 | 0.3114 | -0.1753 | 0.1752 | -0.0067 | 0.9589 | -0.0688 | 0.5948 |
| $\mathcal{J}^{-\mathcal{G}}$ | 0.4106 | 0.0024 | 0.4693 | 0.0005 | 0.4546 | 0.0008 | 0.4790 | 0.0004 | 0.1369 | 0.3108 | 0.2004 | 0.1378 | 0.1027 | 0.4472 |
| $\mathcal{J}^{\mathcal{G}}$ | 0.3290 | 0.0090 | 0.5097 | 0.0000 | 0.4753 | 0.0001 | 0.5398 | 0.0000 | 0.2473 | 0.0521 | 0.2344 | 0.0661 | 0.2602 | 0.0407 |

Table 6: Kendall's Tau rank correlation of all the metrics with model performance of node classification on real-world datasets

The correlation of all the metrics with model performance measured by Kendall's Tau rank correlation is shown in Table 6, which is similar as the results in Pearson correlation. Our Tri-Hom $\mathcal{J}_h^{\mathcal{G}}$ still shows the highest correlation with GNNs performance compared with other types of metrics, which confirms the necessity of disentangling graph homophily.

### E.11 Correlation of Homophily Metrics

To investigate the similarity of the information contained in graph homophily metrics and GNNs performance metrics, we show the Pearson correlation of these metrics on real-world datasets in Figure 8. We can see a high correlation among the homophily on label aspect (including $h_{edge}$, $h_{node}$, $h_{class}$, $h_{adj}$, $h_{den}$, $h_{2hop}$, and $h_{nei}$). These metrics measure the label consistency across the graph topology, sharing a similar characteristic. This also holds for the homophily on the structural aspect (including $LI$, $h_{NC}$, $h_{agg}$, and $h_S$), since these metrics measure how informative the neighbors are for the node labels. However, the Pearson correlation among the homophily on feature aspect ($h_{GE}$, $h_{LS-cos}$, $h_{LS-cos}$, $h_{attr}$, $h_{CF}$, and $h_F$) is low. We speculate that this is because different types of feature aspect ($h_{GE}$, $h_{CF}$, and $h_F$) or different similarity measurements ($h_{LS-cos}$ and $h_{LS-cos}$) could vary a lot for real-world datasets.

## F    Future Directions

Each of the disentangled graph homophily, including label homophily, structural homophily, or feature homophily, derives many interesting directions for future research. In this section, we introduce these potential future directions from the perspective of model design and applications.

### F.1    Model Design

**A. Label homophily**

We discussed how label homophily influences GCN and MLP, providing both theoretical proof and empirical experiments. Our results show that GCN performs better than MLP in conditions of extremely low homophily (good heterophily [46]), but significantly worse than MLP in medium levels of homophily (mid-homophily pitfall [39]). This suggests that GCNs sometimes fail to extract effective topological information. To mitigate this weakness, it is preferable to add a residual connection to GNNs or introduce a learnable parameter that allows the model to balance graph-aware and graph-agnostic information. The necessity of residual connections has been verified in previous studies [66, 45, 52].

In addition to model design at the graph level, we can also consider a fine-grained approach at the node level. Our results indicate that GCN does not always outperform MLP, suggesting that different

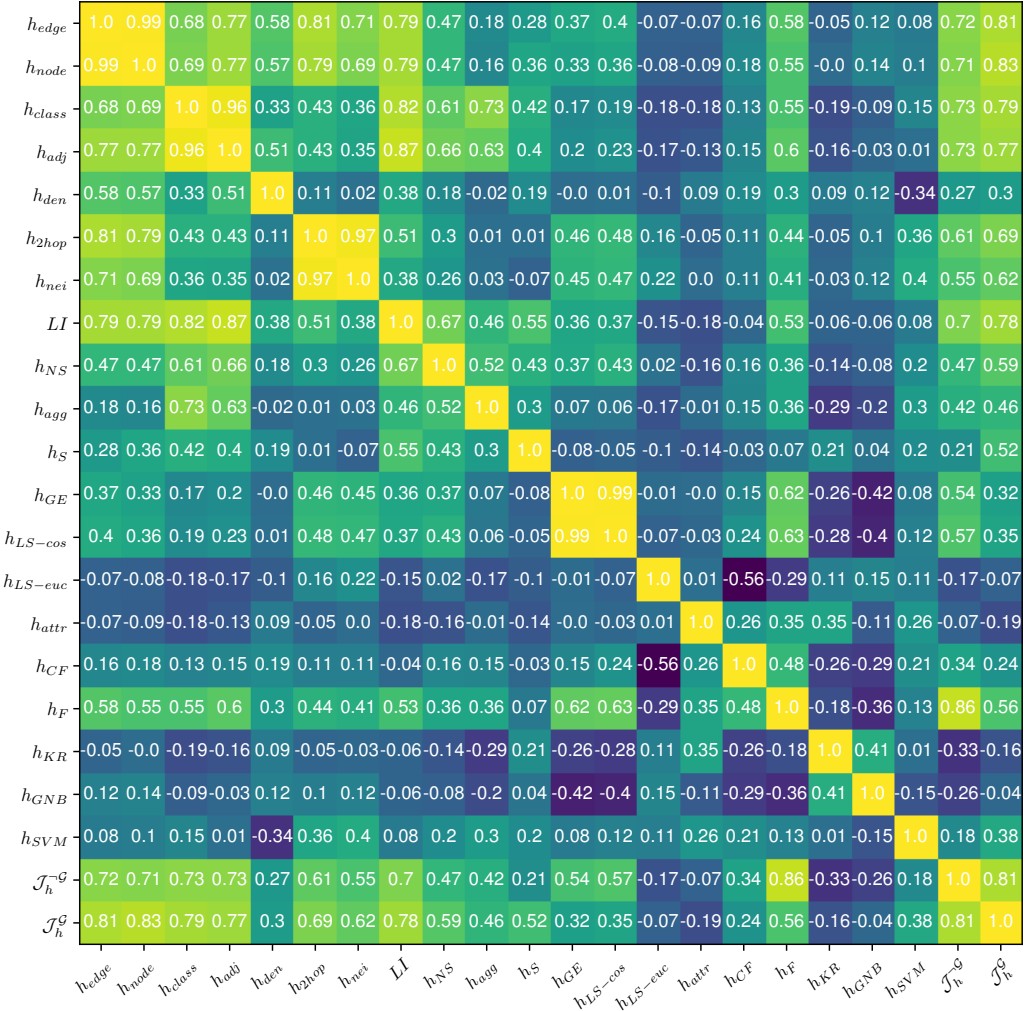

Figure 8: Pearson correlation between all the metrics. Each cell refers to the correlation between two metrics measured on 31 real-world datasets.

models could be applied to nodes with varying levels of label homophily. This type of personalized design has been explored in current GNN research on heterophilic graphs [62, 11], yet it holds significant potential for improving overall model performance.

**B. Structural homophily**

As mentioned in Theorem 2.2, the performance of graph-aware models improves with an increase in structural homophily. This leads to a crucial question: how can we deal with graphs exhibiting varying levels of structural homophily? To address this issue, we can enhance current GNNs using message-passing calibration and graph rewiring.

For the message passing calibration, several methods, such as GPRGNN [8], FB-GNNs [43], and ACM-GNNs [38], propose adding an additional high-pass filter to capture local variations and details in the graph structure. When structural homophily is low, the high-pass filter captures the diversification of individual nodes. Along with a low-pass filter, these methods perform well on graphs with varying levels of homophily. However, the high-pass filters used in these methods cannot capture more complicated structural information. Since $\mathcal{S}(\cdot)$ in structural homophily can be any measurement of structural homophily, this metric can evaluate more complex graph structures. In the

future, it is promising to design novel filters based on structural homophily to capture more intricate structural information and improve model performance.

For the graph rewiring, many methods (MVGCN [63], GloGNN [31], WRGNN [56]) propose deriving a new graph topology based on node features or embeddings. This operation improves the connectivity of nodes with similar semantic contexts, thereby enhancing model performance. However, this rewiring could connect nodes from different classes, which impedes GNN performance. To resolve this, we can measure class-wise structural homophily as shown in Eq. (2) and design adaptations for different classes, which will be particularly beneficial for class-imbalanced graphs, such as in bot detection and fraud detection. Furthermore, we can adapt the structural measurement function $\mathcal{S}(\cdot)$. Since most current rewiring methods do not evaluate the informativeness of their rewiring basis (node embeddings with structural information), the proposed structural homophily can serve as a metric to evaluate which types of rewiring basis to select. For example, Geom-GNN [50] uses structural node embeddings to construct new neighbors of nodes and empirically determines the best approach. Our structural homophily could identify the most effective embedding approach before training GNNs. Therefore, structural homophily provides a guideline for graph rewiring methods.

## C. Feature homophily

The feature homophily proposed in this paper measures how node features are influenced by their neighbors. To our knowledge, only a few GNNs [55] consider these feature dependencies in their design. There is significant potential to explore how feature dependencies function in graphs. For instance, in social networks, people's opinions are affected by those around them. Identifying different user types while filtering out the noise introduced by their neighbors remains an open question. Both our theoretical results (Theorem 2.3) and empirical results (Figure 4) demonstrate the synergy between feature homophily and label homophily in enhancing model performance. Based on these findings, future work could focus on designing various graph filters to optimize the objective in Eq. (8) by considering both label and feature homophily. Furthermore, feature homophily can explain the presence of node features, making it worthwhile to investigate how much features are influenced by their neighbors, particularly in temporal graphs.

## F.2 Applications

Current real-world applications of graph homophily only focus on label homophily, which cannot align well with GNNs performance as shown in this paper and other studies [46, 39]. To address this weakness, our findings provide a comprehensive view of 3 types of homophily, which could be applied in many real-world applications, such as social networks, recommendation, and urban computing.

## A. Social Networks

In social networks, homophily is defined as the tendency for people to seek out or be drawn to others who are similar to themselves [24]. This definition primarily explains the consistency of certain characteristics of people within the network topology. Our proposed concepts of structural homophily and feature homophily offer additional insights into social networks.

Structural homophily refers to the similarity of the local neighbors of individuals of the same type, which can be used to analyze the friend circles of specific user groups. For instance, in fraud detection on social media, fraudsters often target older individuals who are more vulnerable to scams, resulting in a high level of structural homophily. Therefore, we can identify potential fraudsters based on their structural connections. However, structural information can vary and may not always be informative. If fraudsters randomly select users to contact, identifying them through their neighbors becomes challenging, leading to a low level of structural homophily. Future research could focus on measuring the level of structural homophily in social networks to better understand user behaviors.

Feature homophily, on the other hand, describes how individuals are influenced by their neighbors. Different types of networks exhibit varying levels of feature homophily. When people discuss similar events online, their opinions may be influenced by those they follow, leading to a higher similarity in features with their neighbors, indicating a high level of feature homophily. Conversely, during online arguments, individuals connect with those holding different opinions. After such interactions, they may reinforce their original viewpoints, resulting in greater dissimilarity with their neighbors, indicating a low level of feature homophily. Investigating user behavior through feature homophily

can reveal underlying intentions and improve model performance. Furthermore, feature homophily provides valuable insights into the extent to which users are influenced by their neighbors.

### B. Recommendation

Previous studies [22, 12] on recommendation systems using graph homophily have primarily focused on label homophily, which may result in misalignment with model performance, similar to what occurs in purely homogeneous graphs. To address this issue, it would be beneficial to define structural homophily within recommendation systems. For instance, we can measure the structural information of users within the same community by assessing the consistency of the items they have purchased. This approach allows us to determine whether this topological information can effectively predict links between users and specific items.

### C. Urban Computing

A recent study [65] proposes a method to measure spatial graph homophily in urban computing using a spatial diversity score with direction-aware and distance-aware partitions. However, this metric focuses solely on label homophily, leaving a significant opportunity to explore structural homophily. Structural homophily involves measuring the consistency of geographic information among similar types of locations. For example, bookstores are often surrounded by coffee shops, where customers can enjoy coffee while reading books [44], indicating a high level of structural homophily. Conversely, convenience stores might be located near a high-end fashion boutique, a fast-food restaurant, or an office building. Since the geographic information does not reliably indicate the presence of a convenience store, it exhibits a low level of structural homophily. Future research could explore structural homophily in urban graphs to analyze the behaviors of various urban objects, aiding in better city planning.

## G   Proof of Theorems

We first specify the ranges of three types of homophily and spectral radius[8] of an adjacency matrix, which are used for the approximation in partial derivative process:

$$h_L \in [0,1], \ h_S \in [0,1], \ |h_F| \in (-1,1), \ \text{and} \ \rho(\boldsymbol{A}) \gg 1 \tag{46}$$

### G.1   Proof of Theorem 1

**Theorem 1.** In CSBM-3H, the ratio of the expectation of intra-class distance to the expectation of inter-class distance of node representations for graph-agnostic models $\mathcal{M}^{\neg\mathcal{G}}$ and graph-aware models $\mathcal{M}^{\mathcal{G}}$ is:

$$\mathcal{J}^{\neg\mathcal{G}} = (1 + \mathcal{J}_{\boldsymbol{N}}\mathcal{J}_h^{\neg\mathcal{G}})^{-1} \text{ and } \mathcal{J}^{\mathcal{G}} = (1 + \mathcal{J}_{\boldsymbol{N}}\mathcal{J}_h^{\mathcal{G}})^{-1} \text{ respectively,} \tag{47}$$

where $\mathcal{J}_{\boldsymbol{N}} = \frac{\sum_{Y_u \neq Y_v}[2C(C-1)]^{-1}\|\boldsymbol{\mu}_{Y_u} - \boldsymbol{\mu}_{Y_v}\|^2}{C^{-1}|\boldsymbol{\sigma}^2|}$, $\mathcal{J}_h^{\neg\mathcal{G}} = \frac{\left[1 - (\frac{h_F}{\rho(\boldsymbol{A})})^2(C(\frac{1-h_L}{C-1})^2 + C\frac{(1-h_S)^2}{C-1} + (\frac{h_L C-1}{C-1})^2)\right]}{\left[1 - (\frac{h_F}{\rho(\boldsymbol{A})})(\frac{h_L C-1}{C-1})\right]^2}$,

and $\mathcal{J}_h^{\mathcal{G}} = \frac{(\frac{h_L C-1}{C-1})^2}{(C(\frac{1-h_L}{C-1})^2 + \frac{(1-h_S)^2}{C-1} + (\frac{h_L C-1}{C-1})^2)}\mathcal{J}_h^{\neg\mathcal{G}}$.

*Proof.* We have the objective

$$\mathcal{J} = \frac{D_{intra}(\boldsymbol{H})}{D_{inter}(\boldsymbol{H})} = \frac{\mathbb{E}_{Y_u = Y_v, \epsilon}\left[\|\boldsymbol{H}_u - \boldsymbol{H}_v\|^2\right]}{\mathbb{E}_{Y_u \neq Y_v, \epsilon}\left[\|\boldsymbol{H}_u - \boldsymbol{H}_v\|^2\right]} \tag{48}$$

Then, we derive the objective for graph-agnostic models $\mathcal{M}^{\neg\mathcal{G}}$ and graph-aware models $\mathcal{M}^{\mathcal{G}}$ respectively as follows

---

[8]We consider the spectral radius from a general case in a graph. Here the specified range is further confirmed by the values of $\rho(\boldsymbol{A})$ in real-world datasets as shown in Table 2.

### G.1.1 Graph-agnostic Models

For the graph-agnostic models, we let $\boldsymbol{H}_u = \boldsymbol{X}_u$. The structural-agnostic feature $m$ in class $c$ is sampled from a Gaussian distribution $N_{c,m}(\mu_{c,m}, \sigma_{c,m}^2)$. We use $\boldsymbol{N} \in \mathbb{R}^{C \times M}$ to denote the distributions of structural-aware for all the classes with mean $\boldsymbol{\mu} \in \mathbb{R}^{C \times M}$ and standard deviation $\boldsymbol{\sigma^2} \in \mathbb{R}^{C \times M}$ in the matrix form.

For the structural-aware feature $m$ of node $u$, we have[9]

$$
\begin{aligned}
X_{u,m} &\sim \left[ (\boldsymbol{I}_C - \omega \boldsymbol{S})^{-1} \boldsymbol{N}_{:,m} \right]_{Y_u} \\
&= \left[ \sum_{k=0}^{\infty} (\omega \boldsymbol{S})^k \boldsymbol{N}_{:,m} \right]_{Y_u} \\
&= \left[ \sum_{k=0}^{\infty} \omega^k \left[ (\frac{1-h_L}{C-1} + \epsilon) \mathbf{1}_C + (h_L - \frac{1-h_L}{C-1}) \boldsymbol{I}_C \right]^k \boldsymbol{N}_{:,m} \right]_{Y_u}
\end{aligned}
\tag{49}
$$

Next, to simplify the expression, we let $p_1 = \frac{1-h_L}{C-1} + \epsilon$ and $p_0 = h_L - \frac{1-h_L}{C-1}$,

$$
\begin{aligned}
\mathbf{X}_{u,m} &\sim \left[ \sum_{k=0}^{\infty} \omega^k \left[ p_1 \mathbf{1}_C + p_0 \boldsymbol{I}_C \right]^k \boldsymbol{N}_{:,m} \right]_{Y_u} \\
&= \left[ \sum_{k=0}^{\infty} \omega^k \sum_{i=0}^{k} \binom{k}{i} (p_1 \mathbf{1}_C)^{k-i} (p_0 \boldsymbol{I}_C)^i \boldsymbol{N}_{:,m} \right]_{Y_u} \\
&= \left[ \sum_{k=0}^{\infty} \omega^k \sum_{i=0}^{k} \binom{k}{i} (p_1)^{k-i} (p_0)^i \mathbf{1}_C^{k-i} \boldsymbol{I}_C^i \boldsymbol{N}_{:,m} \right]_{Y_u} \\
&= \left[ \sum_{k=0}^{\infty} \omega^k \left[ p_0^k \boldsymbol{I}_C - p_0^k C^{-1} \mathbf{1}_C + \sum_{i=0}^{k} \binom{k}{i} (p_1)^{k-i} (p_0)^i C^{k-i-1} \mathbf{1}_C \right] \boldsymbol{N}_{:,m} \right]_{Y_u} \\
&= \left[ \sum_{k=0}^{\infty} \omega^k \left[ p_0^k \boldsymbol{I}_C - p_0^k C^{-1} \mathbf{1}_C + (Cp_1 + p_0)^k C^{-1} \mathbf{1}_C \right] \boldsymbol{N}_{:,m} \right]_{Y_u} \\
&= \left[ (1 - \omega p_0)^{-1} \boldsymbol{I}_C \boldsymbol{N}_{:,m} + \left[ (1 - \omega(Cp_1 + p_0))^{-1} - (1 - \omega p_0)^{-1} \right] C^{-1} \mathbf{1}_C \boldsymbol{N}_{:,m} \right]_{Y_u}
\end{aligned}
\tag{50}
$$

Then we have the feature distribution of node $u$

$$
X_{u,m} \sim N(\hat{\mu}_{u,m}, \hat{\sigma}_{u,m}^2)
\tag{51}
$$

where

$$
\begin{aligned}
\hat{\mu}_{Y_u,m} &= (1 - \omega p_0)^{-1} \mu_{Y_u,m} + C^{-1} \left[ (1 - \omega(Cp_1 + p_0))^{-1} - (1 - \omega p_0)^{-1} \right] |\boldsymbol{\mu}_{:,m}| \\
\hat{\sigma}_{Y_u,m}^2 &= (1 - \omega^2 p_0^2)^{-1} \sigma_{Y_u,m}^2 + C^{-1} \left[ (1 - \omega^2(Cp_1^2 + p_0^2))^{-1} - (1 - \omega^2 p_0^2)^{-1} \right] |\boldsymbol{\sigma^2}_{:,m}|
\end{aligned}
\tag{52}
$$

When two nodes $u$ and $v$ share same labels, $i.e., Y_u = Y_v$, we have

$$
X_{u,m} - X_{v,m} \sim N(0, \hat{\sigma}_{Y_u,m}^2 + \hat{\sigma}_{Y_v,m}^2)
\tag{53}
$$

Then the expectation of intra-class distance of $\boldsymbol{X}$ can be expressed as

$$
\begin{aligned}
\mathbb{E}_{Y_u = Y_v, \epsilon} \left[ \|\boldsymbol{X}_u - \boldsymbol{X}_v\|^2 \right] &= \mathbb{E}_{Y_u = Y_v, \epsilon} \left[ \sum_{m=0}^{M} (X_{u,m} - X_{v,m})^2 \right] \\
&= \sum_{m=0}^{M} \mathbb{E}_{Y_u = Y_v, \epsilon} \left[ (X_{u,m} - X_{v,m})^2 \right] \\
&= 2C^{-1} (1 - \omega^2 (C \mathbb{E}_\epsilon[p_1^2] + p_0^2))^{-1} |\boldsymbol{\sigma^2}|
\end{aligned}
\tag{54}
$$

---

[9]Instead of deriving structural-aware features from an adjacency matrix, we sample aggregated distributions of features following previous studies [39, 60]. The binarization and symmetrization operations result in compound probability distributions, which make the analysis complex and almost unsolvable.

When two nodes $u$ and $v$ share different labels, *i.e.*, $Y_u \neq Y_v$, we have

$$X_{u,m} - X_{v,m} \sim N(\hat{\mu}_{Y_u,m} - \hat{\mu}_{Y_v,m}, \hat{\sigma}^2_{Y_u,m} + \hat{\sigma}^2_{Y_v,m}) \tag{55}$$

Then the expectation of inter-class distance of $\boldsymbol{X}$ can be expressed as

$$
\begin{aligned}
\mathbb{E}_{Y_u \neq Y_v, \epsilon} \left[ \|\boldsymbol{X}_u - \boldsymbol{X}_v\|^2 \right] &= \mathbb{E}_{Y_u \neq Y_v, \epsilon} \left[ \sum_{m=0}^{M} (X_{u,m} - X_{v,m})^2 \right] \\
&= \sum_{m=0}^{M} \mathbb{E}_{Y_u \neq Y_v, \epsilon} \left[ (X_{u,m} - X_{v,m})^2 \right] \\
&= 2C^{-1}(1 - \omega^2(C\mathbb{E}_\epsilon[p_1^2] + p_0^2))^{-1} \left| \boldsymbol{\sigma^2} \right| \\
&\quad + [C(C-1)]^{-1}(1 - \omega p_0)^{-2} \sum_{Y_u \neq Y_v} \|\boldsymbol{\mu}_{Y_u,:} - \boldsymbol{\mu}_{Y_v,:}\|^2
\end{aligned}
\tag{56}
$$

Then, we come back to the objective by combining Eq. (54) and Eq. (56)

$$
\begin{aligned}
\mathcal{J}^{\neg \mathcal{G}} &= \frac{\mathbb{E}_{Y_u = Y_v, \epsilon} \left[ \|\boldsymbol{H}_u - \boldsymbol{H}_v\|^2 \right]}{\mathbb{E}_{Y_u \neq Y_v, \epsilon} \left[ \|\boldsymbol{H}_u - \boldsymbol{H}_v\|^2 \right]} \\
&= \frac{\mathbb{E}_{Y_u = Y_v, \epsilon} \left[ \|\boldsymbol{X}_u - \boldsymbol{X}_v\|^2 \right]}{\mathbb{E}_{Y_u \neq Y_v, \epsilon} \left[ \|\boldsymbol{X}_u - \boldsymbol{X}_v\|^2 \right]} \\
&= \left( 1 + \frac{\sum_{Y_u \neq Y_v}[2C(C-1)]^{-1} \|\boldsymbol{\mu}_{Y_u,:} - \boldsymbol{\mu}_{Y_u,:}\|^2}{C^{-1} |\boldsymbol{\sigma^2}|} \frac{(1 - \omega^2(C\mathbb{E}_\epsilon[p_1^2] + p_0^2))}{(1 - \omega p_0)^2} \right)^{-1}
\end{aligned}
\tag{57}
$$

Then we take back $p_0$, $p_1$, and $\omega$ with $h_L$, $h_S$, and $h_F$

$$
\begin{aligned}
p_0^2 &= \left( \frac{h_L C - 1}{C - 1} \right)^2 \\
\mathbb{E}_\epsilon[p_1^2] &= \left( \frac{1 - h_L}{C - 1} \right)^2 + \frac{(1 - h_S)^2}{C - 1} \\
\omega &= \frac{h_F}{\rho(\boldsymbol{A})}
\end{aligned}
\tag{58}
$$

So we have

$$
\frac{(1 - \omega^2(C\mathbb{E}_\epsilon[p_1^2] + p_0^2))}{(1 - \omega p_0)^2} = \frac{\left[ 1 - (\frac{h_F}{\rho(\boldsymbol{A})})^2(C(\frac{1-h_L}{C-1})^2 + C\frac{(1-h_S)^2}{C-1} + (\frac{h_L C - 1}{C-1})^2) \right]}{\left[ 1 - (\frac{h_F}{\rho(\boldsymbol{A})})(\frac{h_L C - 1}{C-1}) \right]^2}
\tag{59}
$$

Then, we can rewrite the objective as

$$\mathcal{J}^{\neg \mathcal{G}} = (1 + \mathcal{J}_{\boldsymbol{N}} \mathcal{J}_h^{\neg \mathcal{G}})^{-1} \tag{60}$$

where

$$
\begin{aligned}
\mathcal{J}_{\boldsymbol{N}} &= \frac{\sum_{Y_u \neq Y_v}[2C(C-1)]^{-1} \|\boldsymbol{\mu}_{Y_u,:} - \boldsymbol{\mu}_{Y_u,:}\|^2}{C^{-1} |\boldsymbol{\sigma^2}|}, \\
\mathcal{J}_h^{\neg \mathcal{G}} &= \frac{\left[ 1 - (\frac{h_F}{\rho(\boldsymbol{A})})^2(C(\frac{1-h_L}{C-1})^2 + C\frac{(1-h_S)^2}{C-1} + (\frac{h_L C - 1}{C-1})^2) \right]}{\left[ 1 - (\frac{h_F}{\rho(\boldsymbol{A})})(\frac{h_L C - 1}{C-1}) \right]^2}
\end{aligned}
\tag{61}
$$

### G.1.2 Graph-aware Models

For graph-aware models, we have $\boldsymbol{H}_u = \frac{1}{D_u} \sum_{v \in \mathcal{N}(u)} \boldsymbol{X}_v$, where $\boldsymbol{X}$ is the structural-aware features as in Section G.1.1. For the representation $m$ in node $u$, we have

$$
\begin{aligned}
\boldsymbol{H}_{u,m} &\sim \left[ \frac{\boldsymbol{S}}{\boldsymbol{I}_C - \omega \boldsymbol{S}} \boldsymbol{N}_{:,m} \right]_{Y_u} \\
&= \omega^{-1} \left[ \frac{\omega \boldsymbol{S} - \boldsymbol{I}_C + \boldsymbol{I}_C}{\boldsymbol{I}_C - \omega \boldsymbol{S}} \boldsymbol{N}_{:,m} \right]_{Y_u} \\
&= \omega^{-1} \left[ (\boldsymbol{I}_C - \omega \boldsymbol{S})^{-1} \boldsymbol{N}_{:,m} - \boldsymbol{N}_{:,m} \right]_{Y_u}
\end{aligned}
\tag{62}
$$

We take the result of $(\boldsymbol{I}_C - \omega \boldsymbol{S})^{-1} \boldsymbol{N}_{:,m}$ as shown in Eq. (50). Then we have

$$
\boldsymbol{H}_{u,m} \sim \left[ ((1 - \omega p_0)^{-1} - 1)\omega^{-1} \boldsymbol{I}_C \boldsymbol{N}_{:,m} + \left[ (1 - \omega(Cp_1 + p_0))^{-1} - (1 - \omega p_0)^{-1} \right] C^{-1} \omega^{-1} \mathbf{1}_C \boldsymbol{N}_{:,m} \right]_{Y_u}
\tag{63}
$$

Then we have the distribution of the representation of node $u$

$$
H_{u,m} \sim N(\hat{\mu}_{u,m}, \hat{\sigma}_{u,m}^2)
\tag{64}
$$

where

$$
\begin{aligned}
\hat{\mu}_{Y_u,m} &= \omega^{-1}(1 - \omega p_0)^{-1} - 1)\mu_{Y_u,m} + C^{-1}\omega^{-1} \left[ (1 - \omega(Cp_1 + p_0))^{-1} - (1 - \omega p_0)^{-1} \right] |\boldsymbol{\mu}_{:,m}| \\
\hat{\sigma}_{Y_u,m}^2 &= \omega^{-2}(1 - \omega^2 p_0^2)^{-1} - 1)\sigma_{Y_u,m}^2 + C^{-1}\omega^{-2} \left[ (1 - \omega^2(Cp_1^2 + p_0^2))^{-1} - (1 - \omega^2 p_0^2)^{-1} \right] |\boldsymbol{\sigma}_{:,m}^2|
\end{aligned}
\tag{65}
$$

When two nodes $u$ and $v$ share same labels, *i.e.*,$Y_u = Y_v$, we have

$$
H_{u,m} - H_{v,m} \sim N(0, \hat{\sigma}_{Y_u,m}^2 + \hat{\sigma}_{Y_v,m}^2)
\tag{66}
$$

Then the expectation of intra-class distance of $\boldsymbol{H}$ can be expressed as

$$
\begin{aligned}
\mathbb{E}_{Y_u = Y_v, \epsilon} \left[ \|\boldsymbol{H}_u - \boldsymbol{H}_v\|^2 \right] &= \mathbb{E}_{Y_u = Y_v, \epsilon} \left[ \sum_{m=0}^M \|H_{u,m} - H_{v,m}\|^2 \right] \\
&= 2\omega^{-2} C^{-1}[(1 - \omega^2(C\mathbb{E}[p_1^2] + p_0^2))^{-1} - 1] |\boldsymbol{\sigma}_{:,m}^2|
\end{aligned}
\tag{67}
$$

When two nodes $u$ and $v$ share different labels, *i.e.*,$Y_u \neq Y_v$, we have

$$
H_{u,m} - H_{v,m} \sim N(\hat{\mu}_{Y_u,m} - \hat{\mu}_{Y_v,m}, \hat{\sigma}_{Y_u,m}^2 + \hat{\sigma}_{Y_v,m}^2)
\tag{68}
$$

Then the expectation of inter-class distance of $\boldsymbol{H}$ can be expressed as

$$
\begin{aligned}
\mathbb{E}_{Y_u \neq Y_v, \epsilon} \left[ \|\boldsymbol{H}_u - \boldsymbol{H}_v\|^2 \right] &= \mathbb{E}_{Y_u \neq Y_v, \epsilon} \left[ \sum_{m=0}^M \|H_{u,m} - H_{v,m}\|^2 \right] \\
&= 2\omega^{-2} C^{-1}[(1 - \omega^2(C\mathbb{E}[p_1^2] + p_0^2))^{-1} - 1] |\boldsymbol{\sigma}_{:,m}^2| \\
&\quad + \omega^{-2}[C(C-1)]^{-1}[(1 - \omega p_0)^{-1} - 1]^2 \sum_{Y_u \neq Y_v} \|\boldsymbol{\mu}_{Y_u,:} - \boldsymbol{\mu}_{Y_v,:}\|^2
\end{aligned}
\tag{69}
$$

Then, we come back to the objective

$$
\begin{aligned}
\mathcal{J} &= \frac{\mathbb{E}_{Y_u = Y_v, \epsilon} \left[ \|\boldsymbol{H}_u - \boldsymbol{H}_v\|^2 \right]}{\mathbb{E}_{Y_u \neq Y_v, \epsilon} \left[ \|\boldsymbol{H}_u - \boldsymbol{H}_v\|^2 \right]} \\
&= \left( 1 + \frac{\sum_{Y_u \neq Y_v} [2C(C-1)]^{-1} \|\boldsymbol{\mu}_{Y_u,:} - \boldsymbol{\mu}_{Y_u,:}\|^2}{C^{-1} |\boldsymbol{\sigma}^2|} \frac{[(1 - \omega p_0)^{-1} - 1]^2}{[(1 - \omega^2(C\mathbb{E}[p_1^2] + p_0^2))^{-1} - 1]} \right)^{-1}
\end{aligned}
\tag{70}
$$

Let's consider the term

$$\frac{[(1-\omega p_0)^{-1}-1]^2}{[(1-\omega^2(C\mathbb{E}[p_1^2]+p_0^2))^{-1}-1]} = \frac{p_0^2}{(C\mathbb{E}[p_1^2]+p_0^2)}\frac{(1-\omega^2(C\mathbb{E}[p_1^2]+p_0^2))}{(1-\omega p_0)^2} \tag{71}$$

We notice the second term $\frac{(1-\omega^2(C\mathbb{E}[p_1^2]+p_0^2))}{(1-\omega p_0)^2} = \mathcal{J}_h^{\neg\mathcal{G}}$, thereby we simplify the expression as

$$\frac{[(1-\omega p_0)^{-1}-1]^2}{[(1-\omega^2(C\mathbb{E}[p_1^2]+p_0^2))^{-1}-1]} = \frac{p_0^2}{(C\mathbb{E}[p_1^2]+p_0^2)}\mathcal{J}_h^{\neg\mathcal{G}} \tag{72}$$

Then we take back $p_0$, $p_1$, and $\omega$ with $h_L$, $h_S$, and $h_F$ as in Eq. (58)

$$\frac{[(1-\omega p_0)^{-1}-1]^2}{[(1-\omega^2(C\mathbb{E}[p_1^2]+p_0^2))^{-1}-1]} = \frac{(\frac{h_L C-1}{C-1})^2}{(C(\frac{1-h_L}{C-1})^2+C\frac{(1-h_S)^2}{C-1}+(\frac{h_L C-1}{C-1})^2)}\mathcal{J}_h^{\neg\mathcal{G}} \tag{73}$$

Then, we can rewrite the objective as

$$\mathcal{J}^{\mathcal{G}} = (1+\mathcal{J}_N\mathcal{J}_h^{\mathcal{G}})^{-1} \tag{74}$$

where

$$\begin{aligned}
\mathcal{J}_N &= \frac{\sum_{Y_u \neq Y_v}[2C(C-1)]^{-1}\|\boldsymbol{\mu}_{Y_u,:}-\boldsymbol{\mu}_{Y_u,:}\|^2}{C^{-1}|\boldsymbol{\sigma}^2|}, \\
\mathcal{J}_h^{\mathcal{G}} &= \frac{(\frac{h_L C-1}{C-1})^2}{(C(\frac{1-h_L}{C-1})^2+C\frac{(1-h_S)^2}{C-1}+(\frac{h_L C-1}{C-1})^2)}\mathcal{J}_h^{\neg\mathcal{G}}
\end{aligned} \tag{75}$$

## G.2   Proof of Theorem 3.1

**Theorem 3.1.** The partial derivative of $\mathcal{J}_h^{\neg\mathcal{G}}$ with respect to label homophily $h_L$ satisfies,

$$\frac{\partial \mathcal{J}_h^{\neg\mathcal{G}}}{\partial h_L} \begin{cases} < 0, & \text{if } h_F \in (-1,0) \\ \geq 0, & \text{if } h_F \in [0,1) \end{cases} \tag{76}$$

*Proof.* For the graph-agnostic models, we take the partial derivative of $\mathcal{J}_h^{\neg\mathcal{G}}$ with respect to $h_L$

$$\frac{\partial \mathcal{J}_h^{\neg\mathcal{G}}}{\partial h_L} = \frac{-h_F^2\Big(2C(1-h_L)(-1)+2C(h_L C-1)\Big)\mathcal{T}_{h,\text{DOWN}}^2-(-h_F C)2\mathcal{T}_{h,\text{DOWN}}\mathcal{T}_{h,\text{UP}}}{\mathcal{T}_{h,\text{DOWN}}^4} \tag{77}$$

Take the approximations of $\mathcal{T}_{h,\text{UP}}$ and $\mathcal{T}_{h,\text{DOWN}}$ are given in Eq. (87) back to $\frac{\partial \mathcal{J}_h^{\neg\mathcal{G}}}{\partial h_L}$, we have

$$\begin{aligned}
\frac{\partial \mathcal{J}_h^{\neg\mathcal{G}}}{\partial h_L} &\approx \frac{-2Ch_F^2[(1-h_L)(-1)+(h_L C-1)]+2Ch_F[\rho(\boldsymbol{A})(C-1)]}{[\rho(\boldsymbol{A})(C-1)]^2} \\
&\approx \frac{2C}{[\rho(\boldsymbol{A})(C-1)]}h_F
\end{aligned} \tag{78}$$

It is easy to see $\frac{2C}{[\rho(\boldsymbol{A})(C-1)]} > 0$, so the sign of $\frac{\partial \mathcal{J}_h^{\neg\mathcal{G}}}{\partial h_L}$ is solely depend on $h_F$. Therefore, we have

$$\frac{\partial \mathcal{J}_h^{\neg\mathcal{G}}}{\partial h_L} \begin{cases} < 0, & \text{if } h_F \in (-1,0) \\ \geq 0, & \text{if } h_F \in [0,1) \end{cases} \tag{79}$$

## G.3 Proof of Theorem 3.2

**Theorem 3.2.** The partial derivative of $\mathcal{J}_h^{\neg\mathcal{G}}$ with respect to structural homophily $h_S$ satisfies,

$$\frac{\partial \mathcal{J}_h^{\neg\mathcal{G}}}{\partial h_S} \geq 0 \tag{80}$$

*Proof.* We take the partial derivative of $\mathcal{J}_h^{\neg\mathcal{G}}$ with respect to $h_S$

$$\frac{\partial \mathcal{J}_h^{\neg\mathcal{G}}}{\partial h_S} = \frac{\frac{2C}{\rho^2(\boldsymbol{A})(C-1)}}{\left[1 - (\frac{h_F}{\rho(\boldsymbol{A})})(\frac{h_L C - 1}{C-1})\right]^2} h_F^2 (1 - h_S) \tag{81}$$

From Eq. (46), we have

$$\frac{\frac{2C}{\rho^2(\boldsymbol{A})(C-1)}}{\left[1 - (\frac{h_F}{\rho(\boldsymbol{A})})(\frac{h_L C - 1}{C-1})\right]^2} > 0, \ h_F^2 \geq 0, \ \text{and} \ (1 - h_S) \geq 0 \tag{82}$$

Therefore, we can approximate $\frac{\partial \mathcal{J}_h^{\neg\mathcal{G}}}{\partial h_S}$ that satisfies

$$\frac{\partial \mathcal{J}_h^{\neg\mathcal{G}}}{\partial h_S} \geq 0 \tag{83}$$

## G.4 Proof of Theorem 3.3

**Theorem 3.3.** The partial derivative of $\mathcal{J}_h^{\neg\mathcal{G}}$ with respect to feature homophily $h_F$ satisfies,

$$\frac{\partial \mathcal{J}_h^{\neg\mathcal{G}}}{\partial h_F} \begin{cases} < 0, & \text{if } h_L \in (0, h_L^-); h_L \in (h_L^-, h_L^+) \text{ and } h_F \in (\hat{h}_F, 1) \\ > 0, & \text{if } h_L \in (h_L^+, 1]; h_L \in (h_L^-, h_L^+) \text{ and } h_F \in (-1, \hat{h}_F) \\ = 0, & \text{if } h_L \in (h_L^-, h_L^+) \text{ and } h_F = \hat{h}_F \end{cases} \tag{84}$$

where $0 < h_L^- < h_L^+ < 1$ and $-1 < \hat{h}_F < 1$.

*Proof.* We take the partial derivative of $\mathcal{J}_h^{\neg\mathcal{G}}$ with respect to $h_F$

$$\frac{\partial \mathcal{J}_h^{\neg\mathcal{G}}}{\partial h_F} = \frac{-2h_F\left(C(1-h_L)^2 + C(C-1)(1-h_S)^2 + (h_L C - 1)^2\right)\mathcal{T}_{h,\text{DOWN}}^2 + 2(h_L C - 1)\mathcal{T}_{h,\text{DOWN}}\mathcal{T}_{h,\text{UP}}}{\mathcal{T}_{h,\text{DOWN}}^4} \tag{85}$$

where

$$\begin{aligned} \mathcal{T}_{h,\text{UP}} &= \left[\rho(\boldsymbol{A})^2(C-1)^2 - h_F^2\left(C(1-h_L)^2 + C(C-1)(1-h_S)^2 + (h_L C - 1)^2\right)\right] \\ \mathcal{T}_{h,\text{DOWN}} &= \left[\rho(\boldsymbol{A})(C-1) - h_F(h_L C - 1)\right] \end{aligned} \tag{86}$$

Next, we approximate $\mathcal{T}_{h,\text{UP}}$ and $\mathcal{T}_{h,\text{DOWN}}$ with Eq. (46)

$$\begin{aligned} \mathcal{T}_{h,\text{UP}} &\approx [\rho(\boldsymbol{A})(C-1)]^2 \\ \mathcal{T}_{h,\text{DOWN}} &\approx \rho(\boldsymbol{A})(C-1) \end{aligned} \tag{87}$$

Then take back to Eq. (86)

$$\begin{aligned} \frac{\partial \mathcal{J}_h^{\neg\mathcal{G}}}{\partial h_F} &\approx \frac{-2h_F\left(C(\frac{1-h_L}{C-1})^2 + \frac{C}{C-1}(1-h_S)^2 + (\frac{h_L C - 1}{C-1})^2\right) + 2\frac{h_L C - 1}{C-1}\rho(\boldsymbol{A})}{\rho^2(\boldsymbol{A})} \\ &\approx \frac{2(\hat{h}_F - h_F)}{\left(C(\frac{1-h_L}{C-1})^2 + \frac{C}{C-1}(1-h_S)^2 + (\frac{h_L C - 1}{C-1})^2\right)^{-1}\rho^2(\boldsymbol{A})} \end{aligned} \tag{88}$$

where

$$\hat{h}_F = \frac{\frac{h_L C - 1}{C - 1}\rho(\boldsymbol{A})}{C(\frac{1-h_L}{C-1})^2 + \frac{C}{C-1}(1-h_S)^2 + (\frac{h_L C - 1}{C - 1})^2} \tag{89}$$

As a result, the sign of $\frac{\partial \mathcal{J}_h^{\neg \mathcal{G}}}{\partial h_F}$ satisfies the following conditions:

$$\frac{\partial \mathcal{J}_h^{\neg \mathcal{G}}}{\partial h_F} \begin{cases} > 0, & \text{if } h_F < \hat{h}_F \\ < 0, & \text{if } h_F > \hat{h}_F \\ = 0, & \text{if } h_F = \hat{h}_F \end{cases} \tag{90}$$

Then we discuss the influence of $h_L$ to $\frac{\partial \mathcal{J}_h^{\neg \mathcal{G}}}{\partial h_F}$. If $\hat{h}_F \geq 1$, then $\frac{\partial \mathcal{J}_h^{\neg \mathcal{G}}}{\partial h_F} > 0$ holds $\forall h_F \in (-1, 1)$. In this case, we have

$$-\left(C(\frac{1-h_L}{C-1})^2 + \frac{C}{C-1}(1-h_S)^2 + (\frac{h_L C - 1}{C-1})^2\right) + \frac{h_L C - 1}{C-1}\rho(\boldsymbol{A}) > 0 \tag{91}$$

The solution of this condition for $h_L \in [0, 1]$ is $h_L > h_L^+$, where

$$h_L^+ = \frac{4C + C(C-1)\rho(\boldsymbol{A}) - \sqrt{[4C + C(C-1)\rho(\boldsymbol{A})]^2 - 4C(C+1)[C+1+(C-1)\rho(\boldsymbol{A})+C(C-1)(1-h_S)^2]}}{2C(C+1)} \tag{92}$$

Conversely, If $\hat{h}_F \leq -1$, then $\frac{\partial \mathcal{J}_h^{\neg \mathcal{G}}}{\partial h_F} < 0$ holds $\forall h_F \in (-1, 1)$. In this case, we have

$$\left(C(\frac{1-h_L}{C-1})^2 + \frac{C}{C-1}(1-h_S)^2 + (\frac{h_L C - 1}{C-1})^2\right) + \frac{h_L C - 1}{C-1}\rho(\boldsymbol{A}) > 0 \tag{93}$$

The solution of this condition for $h_L \in [0, 1]$ is $h_L < h_L^-$, where

$$h_L^- = \frac{4C - C(C-1)\rho(\boldsymbol{A}) + \sqrt{[4C - C(C-1)\rho(\boldsymbol{A})]^2 - 4C(C+1)[C+1-(C-1)\rho(\boldsymbol{A})+C(C-1)(1-h_S)^2]}}{2C(C+1)} \tag{94}$$

It is easy to show that $0 < h_L^- < h_L^+ < 1$. So far, we already know that $\frac{\partial \mathcal{J}_h^{\neg \mathcal{G}}}{\partial h_F} > 0$ when $h_L > h_L^+$ and $\frac{\partial \mathcal{J}_h^{\neg \mathcal{G}}}{\partial h_F} < 0$ when $h_L < h_L^-$ for all $h_F \in (-1, 1)$. Then, for $h_L^- < h_L < h_L^+$, the sign of $\frac{\partial \mathcal{J}_h^{\neg \mathcal{G}}}{\partial h_F}$ is dependent on $h_F$ as shown in Eq. (88). In conclusion, we have

$$\frac{\partial \mathcal{J}_h^{\neg \mathcal{G}}}{\partial h_F} \begin{cases} < 0, & \text{if } h_L \in (0, h_L^-); h_L \in (h_L^-, h_L^+) \text{ and } h_F \in (\hat{h}_F, 1) \\ > 0, & \text{if } h_L \in (h_L^+, 1]; h_L \in (h_L^-, h_L^+) \text{ and } h_F \in (-1, \hat{h}_F) \\ = 0, & \text{if } h_L \in (h_L^-, h_L^+) \text{ and } h_F = \hat{h}_F \end{cases} \tag{95}$$

### G.5   Proof of Theorem 2.1

**Theorem 2.1.** The partial derivative of $\mathcal{J}_h^{\mathcal{G}}$ with respect to label homophily $h_L$ satisfies,

$$\frac{\partial \mathcal{J}_h^{\mathcal{G}}}{\partial h_L} \begin{cases} < 0, & \text{if } h_L \in [0, \frac{1}{C}) \\ \geq 0, & \text{if } h_L \in [\frac{1}{C}, 1] \end{cases} \tag{96}$$

*Proof.* For the graph-aware models, we approximate the $\mathcal{J}_h^{\mathcal{G}}$ with Eq. (46)

$$\begin{aligned} \mathcal{J}_h^{\mathcal{G}} &= \frac{(\frac{h_L C - 1}{C-1})^2 \left[1 - (\frac{h_F}{\rho(\boldsymbol{A})})^2 (C(\frac{1-h_L}{C-1})^2 + C\frac{(1-h_S)^2}{C-1} + (\frac{h_L C - 1}{C-1})^2)\right]}{(C(\frac{1-h_L}{C-1})^2 + C\frac{(1-h_S)^2}{C-1} + (\frac{h_L C - 1}{C-1})^2)\left[1 - (\frac{h_F}{\rho(\boldsymbol{A})})(\frac{h_L C - 1}{C-1})\right]^2} \\ &\approx \frac{(\frac{h_L C - 1}{C-1})^2}{(C(\frac{1-h_L}{C-1})^2 + C\frac{(1-h_S)^2}{C-1} + (\frac{h_L C - 1}{C-1})^2)} \end{aligned} \tag{97}$$

Then we take the partial derivative with respect to $h_L$

$$
\begin{aligned}
\frac{\partial \mathcal{J}_h^{\mathcal{G}}}{\partial h_L} = (h_L C - 1)\Big[ & \frac{2C^2(C-1)(C+1)h_L^2}{(C-1)^4} + \frac{8C^2(C-1) + 2C(3C+1)}{(C-1)^4} h_L \\
& + \frac{2C(C+1)(C-1) - 4C + 2C^2(C-1)^2(1-h_S)^2}{(C-1)^4} \Big]
\end{aligned}
\tag{98}
$$

We can see that the first term monotonically increases with respect to $h_L$ and the second term is always positive for $h_L \in [0, 1]$. So we have

$$
\frac{\partial \mathcal{J}_h^{\mathcal{G}}}{\partial h_L} \begin{cases} < 0, & \text{if } h_L \in [0, \frac{1}{C}) \\ \geq 0, & \text{if } h_L \in [\frac{1}{C}, 1] \end{cases}
\tag{99}
$$

## G.6   Proof of Theorem 2.2

**Theorem 2.2.** The partial derivative of $\mathcal{J}_h^{\mathcal{G}}$ with respect to structural homophily $h_S$ satisfies,

$$
\frac{\partial \mathcal{J}_h^{\mathcal{G}}}{\partial h_S} \geq 0
\tag{100}
$$

*Proof.* We take the partial derivative of $\mathcal{J}_h^{\mathcal{G}}$ with respect to $h_F$

$$
\begin{aligned}
\frac{\partial \mathcal{J}_h^{\mathcal{G}}}{\partial h_S} = & \frac{(\frac{h_L C - 1}{C-1})^2 \frac{2C}{C-1}(1 - h_S)}{(C(\frac{1-h_L}{C-1})^2 + C\frac{(1-h_S)^2}{C-1} + (\frac{h_L C - 1}{C-1})^2)^2} \mathcal{J}_h^{\neg\mathcal{G}} \\
& + \frac{(\frac{h_L C - 1}{C-1})^2}{(C(\frac{1-h_L}{C-1})^2 + C\frac{(1-h_S)^2}{C-1} + (\frac{h_L C - 1}{C-1})^2)} \frac{\partial \mathcal{J}_h^{\neg\mathcal{G}}}{\partial h_S}
\end{aligned}
\tag{101}
$$

From Eq. (46), we have

$$
\frac{(\frac{h_L C - 1}{C-1})^2 \frac{2C}{C-1}(1 - h_S)}{(C(\frac{1-h_L}{C-1})^2 + C\frac{(1-h_S)^2}{C-1} + (\frac{h_L C - 1}{C-1})^2)^2} \geq 0 \text{ and } \frac{(\frac{h_L C - 1}{C-1})^2}{(C(\frac{1-h_L}{C-1})^2 + C\frac{(1-h_S)^2}{C-1} + (\frac{h_L C - 1}{C-1})^2)} \geq 0
\tag{102}
$$

Together with $\mathcal{J}_h^{\neg\mathcal{G}} > 0$ and $\frac{\partial \mathcal{J}_h^{\neg\mathcal{G}}}{\partial h_S} \geq 0$, we can approximate $\frac{\partial \mathcal{J}_h^{\mathcal{G}}}{\partial h_S}$ as

$$
\frac{\partial \mathcal{J}_h^{\mathcal{G}}}{\partial h_S} \geq 0
\tag{103}
$$

## G.7   Proof of Theorem 2.3

**Theorem 2.3.** The partial derivative of $\mathcal{J}_h^{\mathcal{G}}$ with respect to feature homophily $h_F$ satisfies,

$$
\begin{cases}
\frac{\partial \mathcal{J}_h^{\mathcal{G}}}{\partial h_F} < 0, & \text{if } h_L \in (0, h_L^-); h_L \in (h_L^-, h_L^+) \text{ and } h_F \in (\hat{h}_F, 1) \\
\frac{\partial \mathcal{J}_h^{\mathcal{G}}}{\partial h_F} > 0, & \text{if } h_L \in (h_L^+, 1]; h_L \in (h_L^-, h_L^+) \text{ and } h_F \in (-1, \hat{h}_F) \\
\frac{\partial \mathcal{J}_h^{\mathcal{G}}}{\partial h_F} = 0, & \text{if } h_L = \frac{1}{C}; h_L \in (h_L^-, h_L^+) \text{ and } h_F = \hat{h}_F
\end{cases}
\tag{104}
$$

where $0 < h_L^- < h_L^+ < 1$ and $-1 < \hat{h}_F < 1$. The expressions and detailed calculation of $h_L^-, h_L^+$, and $\hat{h}_F$ are shown in Appendix G.4.

*Proof.* For the graph-aware models, we take the partial derivative of $\mathcal{J}_h^{\mathcal{G}}$ with respect to $h_F$

$$
\frac{\partial \mathcal{J}_h^{\mathcal{G}}}{\partial h_F} = \frac{(\frac{h_L C - 1}{C-1})^2}{(C(\frac{1-h_L}{C-1})^2 + C\frac{(1-h_S)^2}{C-1} + (\frac{h_L C - 1}{C-1})^2)} \frac{\partial \mathcal{J}_h^{\neg\mathcal{G}}}{\partial h_F}
\tag{105}
$$

Since the term $\frac{(\frac{h_L C - 1}{C - 1})^2}{(C(\frac{1 - h_L}{C - 1})^2 + C\frac{(1 - h_S)^2}{C - 1} + (\frac{h_L C - 1}{C - 1})^2)} \geq 0$, the sign of $\frac{\partial \mathcal{J}_h^{\mathcal{G}}}{\partial h_F}$ is the determined by $\frac{\partial \mathcal{J}_h^{\neg \mathcal{G}}}{\partial h_F}$ except for $h_L = \frac{1}{C}$, which leads to the similar results as graph-aware models

$$
\begin{cases}
\frac{\partial \mathcal{J}_h^{\mathcal{G}}}{\partial h_F} < 0, & \text{if } h_L \in (0, h_L^-); h_L \in (h_L^-, h_L^+) \text{ and } h_F \in (\hat{h}_F, 1) \\
\frac{\partial \mathcal{J}_h^{\mathcal{G}}}{\partial h_F} > 0, & \text{if } h_L \in (h_L^+, 1]; h_L \in (h_L^-, h_L^+) \text{ and } h_F \in (-1, \hat{h}_F) \\
\frac{\partial \mathcal{J}_h^{\mathcal{G}}}{\partial h_F} = 0, & \text{if } h_L = \frac{1}{C}; h_L \in (h_L^-, h_L^+) \text{ and } h_F = \hat{h}_F
\end{cases}
\tag{106}
$$

