# OpenReview forum: "What Is Missing For Graph Homophily? Disentangling Graph Homophily For Graph Neural Networks"
_NeurIPS.cc/2024/Conference — NeurIPS 2024 poster_

### Official Review · Reviewer_9nLU · 2024-06-30

**Soundness:** 3
**Presentation:** 2
**Contribution:** 2
**Rating:** 4
**Confidence:** 3

**Summary:**

This paper analyzes graph homophily and distinguishes three types of homophily: label, structural, and feature homophily. Theoretical analysis is based on CSBM-3H (contextual stochastic block model with three types of homophily).  Based on this analysis, a new combined measure Tri-Hom is proposed. The relation between GNN performance and three types of homophily is empirically analyzed on CSBM-3H. Then, the agreement of Tri-Hom with GNN performance is analyzed using several real datasets. It is shown that the proposed homophily measure better agrees with the GNN performance than other existing measures.

**Strengths:**

1. The proposed model CSBM-3H is more flexible and models more complicated relations between graph structure, node features, and node labels than other typically used models. Such a model can be used as a synthetic benchmark in research papers.

2. Theoretical analysis shows how different homophily aspects relate to the distinguishability of aggregated and non-aggregated node features.

**Weaknesses:**

1. There seems to be a terminological inconsistency in the text. In the abstract, it is written that "graph homophily refers to the phenomenon that connected nodes tend to share similar characteristics." However, structural homophily introduced in Section 3.2 is about a different phenomenon: nodes from the same class tend to have similar neighbors (but they can be not connected). In some previous papers, e.g. in [31], these concepts are distinguished, and LI is not called a homophily measure.

2. The particular form of structural homophily (2) is not motivated. In Section 3.2, it is written that there are several existing structural homophily measures, but they are not chosen. Why the proposed expression (2) is better?

3. Similarly, there is not much motivation for the feature homophily proposed in (3). More motivation on why this diffusion process is supposed to reflect homophily would be helpful. In particular, (3) seems to assume a particular form of feature propagation. It is unclear how limiting this assumption is.

4. Throughout the paper, the class sizes are assumed to be balanced. In line 192, it is written that this is done without loss of generality, but it is not clear why so. In particular, several papers previously discussed that node/edge homophily is not a suitable measure for class-imbalanced datasets. In other words, class balance plays a crucial role for homophily measures and thus considering only balanced classes is a significant limitation. One of the examples where class balance seems to be critical is Theorem 2.2: here the threshold 1/C holds only for balanced classes.

Some typos:
- Line 12: "element" -> "elements"
- Line 61: "i.i.d.assumption" — missing space
- Line 62: "Compared previous" -> "Compared to the previous"
- Line 69: "In addition, Our" -> "In addition, our"
- Line 82: It is written that $I_E$ is a matrix, in this case, it should be from $R^{E\times E}$

**Questions:**

1. In Section 3.1, many label homophily measures are introduced. Which one is chosen by the authors?

2. In equation (2), how exactly $\sigma_{max}$ is computed?

3. Could you give more details/motivation about the difference between label and structural homophily (in general)? Since both are expected to describe the relation between structure and labels.

4. In equation (5), it would be useful to discuss whether the obtained values $h$ are comparable across features of different natures. In particular, how these values behave when we do certain feature transformations: shifts, scaling, or changing variance. These would give more intuition about the proposed form of feature homophily.

5. How performance is measured in Table 1?

**Limitations:**

Some of the limitations are discussed in Section 6. I agree with the authors that societal impact is irrelevant for this type of work.

---

> ### Author Rebuttal · Authors · 2024-08-07
>
> Thanks for your careful reviews. We have fixed the typos and revised our manuscripts accordingly. Here are our responses to your concerns.
>
> ## Part 1/4
>
> ## W1: Terminological inconsistency of graph homophily and structural homophily.
>
> ## RW1:
>
> Graph homophily is a general concept that describes the phenomenon of connected nodes tending to have similar characteristics. However, in the current graph community, the concept of homophily can sometimes be generalized to describe relations beyond connected nodes, such as Aggregation homophily [9], Coleman's  homophily [10], and Preference-aware homophily [11]. The proposed structural homophily measures the consistency of structural information within intra-class nodes, which is crucial for the disentanglement of graph homophily. There indeed exists potential confusion and we will clarify their relations in the revised version.
>
>
> ## W2: 1) The particular form of structural homophily (2) is not motivated. 2) Why not choose other existing structural homophily measures and 3) why the proposed expression (2) is better?
>
> ## RW2:
>
> 1) As mentioned in Line 109, Section 3.1, label homophily focuses solely on the consistency of label information for connected nodes, while neglecting structural information. This limitation leads to a partial understanding of graph homophily and results in the misalignment of label homophily with GNNs’ performance [1]. To address this, we propose structural homophily $h_S$ in Section 3.2.
>
> 2) The proposed $h_S$ in Eq. (2) is based on the sampling process, unlike the existing structural-based metric  [7,8]. This enables $h_S$ to be incorporated into CSBM-3H to analyze the impact on graph-aware models. Furthermore, $h_S$ provides a general form that can be applied to broad settings. For example, to extend $h_S$ to multi-hop neighbors, we can revise $\mathcal{S}(\cdot)$ in Eq. (2), a flexibility not available with other metrics.
>
> 3) In our experiments, the effectiveness of the proposed Eq. (2) is demonstrated in Table 5 in Appendix D.9, where we measure the correlation of metrics with the performance gap between GNNs and MLPs. $h_S$ shows the best correlation compared with other structural-based homophily metrics. Additionally, we explore how $h_S$ influences accuracy on graphs with respect to each class in Figure 6. The findings indicate a general tendency of increased accuracy with higher $h_S$.
>
>
> ## W3: 1) The motivation for the feature homophily proposed in (3). 2) Why this diffusion process is supposed to reflect homophily would be helpful. 3) In particular, Eq. (3) seems to assume a particular form of feature propagation. It is unclear how limiting this assumption is.
>
> ## RW3:
>
> 1) We would like to clarify the motivation for feature homophily $h_F$ in the paper. In Section 3.3, we list all the current feature-based homophily metrics, which cannot disentangle themselves from label homophily, leading to redundancy and a decrease in useful information within the feature aspect. Based on this motivation, we propose $h_F$ to fully disentangle the feature aspect. The $h_F$ is invariant with respect to both label and structural homophily, thereby dissociating it from these two types of homophily. This is part of our main contributions, which disentangles graph homophily from three aspects, better aligns with GNN performance, and helps explain other interesting but under-explored phenomena of graph homophily in previous studies [1, 2, 3].
>
> 2) In Line 152, we explain the meaning of $h_F$ through the sign of $\omega$. A positive, negative, or zero $\omega$ corresponds to an attractive relation, a repulsive relation, or independence of the nodes with their neighbors, respectively. Specifically, consider a case in social networks where people interact with each other. A positive $h_F$ refers to users influencing their neighbors to adopt the same opinions as theirs. A negative $h_F$ refers to users arguing with each other while more strongly holding their opinions to oppose others. A zero $h_F$ indicates that interactions among users do not change their opinions. This process in social networks is generally referred to as a diffusion process [4], which could reflect the $h_F$ defined in Eq. (3).
>
> 3) In Appendix B, we discuss why the assumption of a diffusion process makes sense. In Eq. (28), we show that the structural-aware features are updated by $\frac{\partial\boldsymbol{X}(t)}{\partial t}$, which contains $\boldsymbol{X}(0)$ (ego-node features) and $(\mathcal{F}(\boldsymbol{A}) - \boldsymbol{I})\boldsymbol{X}(t-1)$ (influences of the graph topology). We use linear dependency to model this feature dependency function $\mathcal{F}(\boldsymbol{X}) = \omega\boldsymbol{A}$ following [5] to analyze graph homophily with GNN performance. To the best of our knowledge, we are the first to consider feature dependencies to analyze GNN performance under CSBMs. In contrast, previous studies [1, 2, 3, 6], which assume no feature dependencies during graph modeling, oversimplify by completely ignoring feature dependency.
>
> ## W4: Throughout the paper, the class sizes are assumed to be balanced. In line 192, it is written that this is done without loss of generality, but it is not clear why so.
>
> ## RW4:
>
> This paper aims to disentangle graph homophily from three aspects and investigate its overall impact at the graph level. While in-balanced classes do influence the graph, this influence does not affect the impact of graph homophily, as shown in previous studies [2, 6]. Consequently, the class-balanced assumption has little effect on our findings and doesn't hurt our main contributions. Our observations on CSBM-3H also show the setting of in-balanced class sizes doesn't give us extra information.

---

> ### Author Response · Authors · 2024-08-07
> **Part 2/4**
>
> ## Q1: In Section 3.1, many label homophily measures are introduced. Which one is chosen by the authors?
>
> ## RQ1:
>
> As described in Footnote 6 on Page 8, we use node homophily as label homophily in our experiments.
>
>
> ## Q2: In equation (2), how exactly is $\sigma_{max}$ computed?
>
> ## RQ2:
>
> We measure the standard deviation of structural information in Eq. (2), the value of $\sigma_{max}$ depends on which function of structural information $\mathcal{S}(\cdot)$ is select. As mentioned in Line 127, we use the class distribution of local neighbors $\boldsymbol{D}^{\mathcal{N}}$ as the $\mathcal{S}(\cdot)$. The maximum variance is achieved when $\boldsymbol{D}^{\mathcal{N}}$ is the most diverse for nodes within the same class, which is equal to $\sqrt{\frac{C-1}{C^2}}$ as we used in our experiments. Let me briefly introduce how exactly $\sigma_{max}$ is calculated in this scenario. This question is equal to:
>
> Given $P\in\mathbb{R}^{N\times C}$ is a neighbor sampling probability matrix, where $p_{i,k}\in[0,1]$ and $\sum_{k=1}^C p_{i,k}=1$ for each i, we need to maximize: $\sigma_{max}=\frac{1}{C}\sum_{k=1}^C\sigma(p_{:,k})$.
>
> We first rewrite the $\sigma(p_{:,k})$ as
> \begin{equation}
>     \sigma(p_{:,k}) = \sqrt{\frac{1}{N}\sum_{i=1}^N p_{i,k}^2 - \mu_k^2}
> \end{equation}
> where $\mu_k=\frac{1}{N}\sum_{i=1}^N p_{i,k}$
>
> To maximize $\frac{1}{N}\sum_{i=1}^N p_{i,k}^2 - \mu_k^2$, we need the values to be as unequal as possible given the constraint $\sum_{k=1}^C p_{i,k}=1$. To get maximally unequal distribution, we have $p_{:,k}$ has one entry that is $1$ and the rest $0$ for each column $k$.
>
> Then, let $a_k$ be the number of $1$ in column k, we have $\frac{1}{N}\sum_{i=1}^N p_{i,k}^2=\frac{a_k}{N}$ and $\mu_k = \frac{a_k}{N}$. Next, we can express the original expression as
> \begin{equation}
>     \frac{1}{C}\sum_{k=1}^C\sigma(p_{:,k}) = \frac{1}{C}\sum_{k=1}^C \sqrt{ \frac{a_k}{N} - (\frac{a_k}{N})^2 }
> \end{equation}
>
> We observe that this is a non-convex function. According to Jensen’s inequality, we have
>
> \begin{equation}
>     \frac{1}{C}\sum_{k=1}^C \sqrt{ \frac{a_k}{N} - (\frac{a_k}{N})^2 } \le \sqrt{ \frac{\frac{1}{C}\sum_{k=1}^Ca_k}{N} - (\frac{\frac{1}{C}\sum_{k=1}^C a_k}{N})^2 } = \sqrt{\frac{1}{C}-\left(\frac{1}{C}\right)^2}= \sqrt{\frac{C-1}{C^2}}
> \end{equation}
>
> where we get $\sigma_{max} = \sqrt{\frac{C-1}{C^2}}$ proved.
>
> Note that this $\sigma_{max}$ only applies when using the class distribution of local neighbors, $\boldsymbol{D}^{\mathcal{N}}$, as the function of structural information $\mathcal{S}(\cdot)$. Different functions may yield different values for $\sigma_{max}$.
>
>
> ## Q3: Could you give more details/motivation about the difference between label and structural homophily (in general)? Since both are expected to describe the relation between structure and labels.
>
> ## RQ3:
>
> We highlight two key differences between label homophily ($h_L$) and structural homophily ($h_S$). First, the "atom" information, as described in line 116, for $h_L$ is the label information $Y$, while for $h_S$ it is the structural information $D^\mathcal{N}$ (where we use neighbor distribution in this paper). Second, the measurement function for $h_L$ is an indicator function of two nodes connected by topology, $\mathbb{1}(Y_u=Y_v)$, where $e_{uv}\in\mathcal{E}$. In contrast, for $h_S$, it is the consistency function between two nodes connected by their classes, $(D^\mathcal{N}_u - D^\mathcal{N}_v)^2$, where $Y_u=Y_v$ (by rewriting Eq. (2)). Each of these metrics reflects a unique aspect that the other cannot capture.
>
> To further illustrate these differences, we provide diagrams and visualizations of three types of homophily in the author rebuttal PDF. Figures 2(a), (b), and (c) show that an increase in $h_L$ enhances the connectivity of intra-class nodes. However, even when $h_L$ is low, sometimes the performance of the GNN can still be satisfactory. This is because $h_L$ does not account for the consistency of structural information among intra-class nodes. To address this, we propose $h_S$, which captures this aspect. Figures 2(d), (e), and (f) demonstrate that an increase in $h_S$ improves the informativeness of the neighbors of intra-class nodes, making the graph resemble planar and periodic graphs.
>
> Thus, $h_L$ and $h_S$ represent graph homophily from label and structural perspectives, respectively. Along with feature homophily ($h_F$), they provide a comprehensive understanding of graph homophily. Additional explanations of $h_L$ and $h_S$ are provided in our responses in RW2 and author rebuttal.

---

> ### Author Response · Authors · 2024-08-07
> **Part 3/4**
>
> ## Q4: In equation (5), how these values behave when we do certain feature transformations: shifts, scaling, or changing variance?
>
> ## RQ4:
>
> Thank you for raising the question about the feature transformations on feature homophily in Eq. (5). We need to clarify that the Eq. (4)(5) are the definition of feature homophily and the measurement are introduced in Eq. (6). Then we will prove how this estimation of feature homophily in Eq. (6) is invariant to feature transformation including shifts, scaling, or changing variance as follows.
>
> We first define the problem: In a graph $\mathcal{G}=\{\mathcal{V},\mathcal{E}\}$ with $N$ nodes, there are node labels $\boldsymbol{Y}$ and node features $\boldsymbol{X_{:,m}}$ in dimension $m$. We can estimate the feature homophily $h^*_{F,m}$ for feature in dimension $m$ as:
> \begin{equation}
> h_{F,m}^{*}(\mathcal{G},\boldsymbol{X_{:,m}},\boldsymbol{Y}) = \text{arg} \min_{h_{F,m}} \sum_{\substack{u,v\in\mathcal{V},\\ Y_u=Y_v}} \left[X_{u,m}(0)-X_{v,m}(0)\right]^2, \; \text{ where } \boldsymbol{X_{:,m}}(0) = \left(\boldsymbol{I}-\frac{h_{F,m}}{\rho(\boldsymbol{A})}\boldsymbol{A}\right)\boldsymbol{X_{:,m}}
> \end{equation}
> We need to prove the estimation of feature homophily \textbf{is invariant to the operations of feature shifts, scaling, or changing variance.}
>
> **A. Shifts**
>
> Let's consider a shift of node features $\boldsymbol{X_{:,m}}$ by a constant vector $\boldsymbol{C}$:
> \begin{equation}
>     \boldsymbol{X_{:,m}'} = \boldsymbol{X_{:,m}}+\boldsymbol{C}
> \end{equation}
>
> Then we have structural-agnostic features as
>
> $\boldsymbol{X_{:,m}'}(0)
> = \left(\boldsymbol{I}-\frac{h_{F,m}}{\rho(\boldsymbol{A})}\boldsymbol{A}\right)\boldsymbol{X_{:,m}'}$
>
> $= \left(\boldsymbol{I}-\frac{h_{F,m}}{\rho(\boldsymbol{A})}\boldsymbol{A}\right)(\boldsymbol{X_{:,m}}+\boldsymbol{C})$
>
> $=\boldsymbol{X_{:,m}}(0)+\left(\boldsymbol{I}-\frac{h_{F,m}}{\rho(\boldsymbol{A})}\boldsymbol{A}\right)\boldsymbol{C}$
>
> Then a new estimation of $h_{F,m}'$ under this feature shift can be expressed as
>
> $h_{F,m}'(\mathcal{G},\boldsymbol{X_{:,m}'},\boldsymbol{Y}) = \text{arg} \min_{h_{F,m}} \sum_{\substack{u,v\in\mathcal{V},\\ Y_u=Y_v}} \left[X_{u,m}'(0)-X_{v,m}'(0)\right]^2$
>
> $= \text{arg} \min_{h_{F,m}} \sum_{\substack{u,v\in\mathcal{V},\\ Y_u=Y_v}} \left[\left(X_{u,m}(0)+\left(\boldsymbol{I}-\frac{h_{F,m}}{\rho(\boldsymbol{A})}\boldsymbol{A}\right)C_u\right)-\left(X_{v,m}(0)+\left(\boldsymbol{I}-\frac{h_{F,m}}{\rho(\boldsymbol{A})}\boldsymbol{A}\right)C_v\right)\right]^2$
>
> $= \text{arg} \min_{h_{F,m}} \sum_{\substack{u,v\in\mathcal{V},\\ Y_u=Y_v}} \left[\left(X_{u,m}(0)-X_{v,m}(0)\right)+\left(\boldsymbol{I}-\frac{h_{F,m}}{\rho(\boldsymbol{A})}\boldsymbol{A}\right)(C_u-C_v)\right]^2$
>
> Since $\boldsymbol{C}$ is a constant vector, we have $C_u-C_v=0$. Next, we have
>
> $h_{F,m}'(\mathcal{G},\boldsymbol{X_{:,m}'},\boldsymbol{Y}) = \text{arg} \min_{h_{F,m}} \sum_{\substack{u,v\in\mathcal{V},\\ Y_u=Y_v}} \left[\left(X_{u,m}(0)-X_{v,m}(0)\right)\right]^2 = h_{F,m}^{*}(\mathcal{G},\boldsymbol{X_{:,m}},\boldsymbol{Y})$
>
> Therefore, we proved the estimation of feature homophily is invariant to the operation of feature shifts.
>
> **B. Scaling**
>
> Let's consider scaling node features $\boldsymbol{X_{:,m}}$ by a constant $\alpha$:
> \begin{equation}
>     \boldsymbol{X_{:,m}'} = \alpha\boldsymbol{X_{:,m}}
> \end{equation}
>
> Then we have structural-agnostic features as
>
> $\boldsymbol{X_{:,m}'}(0)
> = \left(\boldsymbol{I}-\frac{h_{F,m}}{\rho(\boldsymbol{A})}\boldsymbol{A}\right)(\alpha\boldsymbol{X_{:,m}'})$
>
> $= \alpha\left(\boldsymbol{I}-\frac{h_{F,m}}{\rho(\boldsymbol{A})}\boldsymbol{A}\right)\boldsymbol{X_{:,m}'}$
>
> $= \alpha\boldsymbol{X_{:,m}}(0)$
>
> Then a new estimation of $h_{F,m}'$ under this feature shift can be expressed as
>
> $h_{F,m}'(\mathcal{G},\boldsymbol{X_{:,m}'},\boldsymbol{Y}) = \text{arg} \min_{h_{F,m}} \sum_{\substack{u,v\in\mathcal{V},\\ Y_u=Y_v}} \left[X_{u,m}'(0)-X_{v,m}'(0)\right]^2$
>
> $= \text{arg} \min_{h_{F,m}} \sum_{\substack{u,v\in\mathcal{V},\\ Y_u=Y_v}} \alpha^2\left[X_{u,m}(0)-X_{v,m}(0)\right]^2$
>
> $= \text{arg} \min_{h_{F,m}} \alpha^2 \sum_{\substack{u,v\in\mathcal{V},\\ Y_u=Y_v}} \left[X_{u,m}(0)-X_{v,m}(0)\right]^2$
>
> Since $\text{arg} \min_{x} (\cdot)$ is invariant to the scaling, e.g. $\text{arg} \min_{x}(cf(x))=\text{arg} \min_{x}(f(x))$, we have
>
> $h_{F,m}'(\mathcal{G},\boldsymbol{X_{:,m}'},\boldsymbol{Y}) = \text{arg} \min_{h_{F,m}} \sum_{\substack{u,v\in\mathcal{V},\\ Y_u=Y_v}} \left[\left(X_{u,m}(0)-X_{v,m}(0)\right)\right]^2 = h_{F,m}^{*}(\mathcal{G},\boldsymbol{X_{:,m}},\boldsymbol{Y})$
>
> Therefore, we proved the estimation of feature homophily is invariant to the operation of feature scaling.

---

> ### Author Response · Authors · 2024-08-07
> **Part 4/4**
>
> **C. Variance Changing**
>
> Changing the variance of $\boldsymbol{X_{:,m}}$ can be seen as the combination of scaling and shifts. Assume node features follow a Gaussian distribution $N(\mu,\sigma^2)$, after the operation of changing the variance from $\sigma^2$ to $\beta\sigma^2$, we have new node features as
>
> \begin{equation}
>     \boldsymbol{X_{:,m}'} = \sqrt{\beta}(\boldsymbol{X_{:,m}}-\mu)+\mu
> \end{equation}
>
> where $\boldsymbol{X_{:,m}'}$ is calculated by deducing $\mu$, multiplying $\sqrt{\beta}$, and adding $\mu$. We already show the estimation of feature homophily is invariant to the operations of feature shifts and scaling. Since the operation of variance changing is a combination of scaling and shifts, we can conclude that the estimation of feature homophily is invariant to the variance changing. We will add these proofs to the Appendix later.
>
> ## Q5: How performance is measured in Table 1?
>
> ## RQ5:
>
> The measurement of the performance in Table 1 is introduced in Line 330, where we show the Pearson correlation between all the metrics and model performance on the 31 real-world datasets. Specifically, each cell in Table 1 denotes a correlation value, which is calculated using the corresponding homophily metrics and model performance on 31 datasets. For example, for $h_{edge}$ on GCN, we measure the $h_{edge}$ (in Table 3) and the performance of GCN (in Table 4) on 31 datasets. Then, we have 31 values for both the homophily metric and model performance. Finally, we calculate the correlation between these two sets of data, which reflects how well the homophily metric aligns with model performance. We hope this clarification helps in understanding the performance shown in Table 1.
>
> **References**
>
> [1] Ma, Y., Liu, X., Shah, N., Tang, J. Is Homophily a Necessity for Graph Neural Networks? In ICLR, 2022.
>
> [2] Wang, J., Guo, Y., Yang, L., Wang, Y. Understanding Heterophily for Graph Neural Networks. CoRR abs/2401.09125, 2024.
>
> [3] Lee, S. Y., Kim, S., Bu, F., Yoo, J., Tang, J., Shin, K. Feature Distribution on Graph Topology Mediates the Effect of Graph Convolution: Homophily Perspective. CoRR abs/2402.04621, 2024.
>
> [4] Chang, B., Xu, T., Liu, Q., et al. Study on Information Diffusion Analysis in Social Networks and Its Applications. Int. J. Autom. Comput. 15, 377–401, 2018.
>
> [5] Shi, D., Han, A., Lin, L., Guo, Y., Wang, Z., Gao, J. Design Your Own Universe: A Physics-Informed Agnostic Method for Enhancing Graph Neural Networks. arXiv:2401.14580, 2024.
>
> [6] Luan, S., Hua, C., Xu, M., Lu, Q., Zhu, J., Chang, X.-W., Fu, J., Leskovec, J., Precup, D. When Do Graph Neural Networks Help with Node Classification? Investigating the Homophily Principle on Node Distinguishability. In NeurIPS, 36, 2024.
>
> [7] S. Luan, C. Hua, Q. Lu, J. Zhu, M. Zhao, S. Zhang, X.-W. Chang, and D. Precup. Revisiting heterophily for graph neural networks. In NeurIPS, 2022.
>
> [8] O. Platonov, D. Kuznedelev, A. Babenko, and L. Prokhorenkova. Characterizing graph datasets for node classification: Homophily-heterophily dichotomy and beyond. In NeurIPS, 2024.
>
> [9] Luan S, Hua C, Lu Q, et al. Revisiting heterophily for graph neural networks[J]. In NeurIPS, 2022.
>
> [10] Coleman J S. Relational analysis: The study of social organizations with survey methods[J]. Human organization, 1958.
>
> [11] Jiang W, Gao X, Xu G, et al. Challenging Low Homophily in Social Recommendation[C], Proceedings of the ACM on Web Conference 2024.

---

> > ### Comment · Reviewer_9nLU · 2024-08-11
> >
> > Thank you for your detailed response! I have several questions to clarify.
> >
> > W4. Could you please provide more details on why class-balanced assumption does not affect the results and conclusions? In particular, could you please clarify whether the following is true: “One of the examples where class balance seems to be critical is Theorem 2.2: here the threshold 1/C holds only for balanced classes”?
> >
> > Q4. Thanks for the detailed response! Just to clarify – is it true that this invariance holds for the estimate (6) but may not hold for the original definition (or it is hard to prove it there)?
> >
> > Q5. Here my question was about “model performance” – which measure is used here?

---

> > > ### Author Response · Authors · 2024-08-14
> > > **Reply 2 nd rebuttal**
> > >
> > > Please let me know if my responses have addressed your concerns. Thank you!

---

> ### Author Response · Authors · 2024-08-12
> **Rebuttal - 2nd Response**
>
> ## W4
> Could you please provide more details on why class-balanced assumption does not affect the results and conclusions? In particular, could you please clarify whether the following is true: “One of the examples where class balance seems to be critical is Theorem 2.2: here the threshold 1/C holds only for balanced classes”?
>
> ## RW4 - 2nd
>
> Thank you for your additional review comments. You are correct that the $\frac{1}{C}$ factor in Theorem 2.2 is derived under the assumption of balanced datasets, as many current studies do not adequately address the challenges posed by unbalanced data [1, 2, 3, 7, 8]. Given that handling unbalanced data is not the primary focus of our current work and the absence of a standard benchmark for imbalanced data in heterophilous graphs, we conducted a preliminary experiment using synthetic datasets with imbalanced data. The experimental results, presented in the table below, suggest that the conclusions of Theorem 2.2 remain valid and are not significantly impacted by data imbalance. However, we agree that addressing unbalanced data will be crucial in future work, particularly for model design.
>
> The table below illustrates how varying $h_L$ impacts GCN performance on a node classification task. We examine a scenario with 3 classes and 1,000 nodes, reporting the accuracy and standard deviation across 5 random runs. The cases are: 1) balanced classes (33\%/33\%/33\%), 2) imbalanced classes case 1 (60\%/30\%/10\%), and 3) imbalanced classes  case 2 (80\%/10\%/10\%).
>
> | $h_L$ | 0.0 | 0.1 | 0.2 | 0.3 | 0.4 | 0.5 | 0.6 | 0.7 | 0.8 | 0.9 | 1.0 |
> | --- | --- | --- | --- | --- | --- | --- | --- | --- | --- | --- | --- |
> | Balanced | 87.44±4.14 |74.32±4.83 |60.72±6.77 |**53.44±6.90** |58.56±4.53 |72.56±3.69 |81.12±2.36 |90.08±4.13 |96.80±1.47 |98.00±1.36 |99.36±0.61 |
> | In-balanced 1 | 90.16±3.18 |83.12±3.50 |71.60±7.81 |**63.28±4.06** |68.24±5.21 |76.88±4.11 |83.04±2.91 |88.80±3.31 |92.56±2.63 |94.64±2.27 |97.84±1.69 |
> | In-balanced 2 | 80.92±3.98 |80.00±1.62 |**77.12±0.52** |78.96±2.52 |79.36±2.66 |80.36±2.94 |82.68±4.54 |85.76±5.37 |92.80±2.67 |94.12±2.45 |96.40±1.90 |
>
> The results demonstrate that as $h_L$ increases, GCN performance initially declines and then improves across both balanced and imbalanced datasets. This trend aligns with the conclusions of Theorem 2.2, which describe that GNN performance reaches its lowest point at mid-level values of $h_L$. Therefore, under the in-balanced classes, the conclusions from Theorem 2.2 still hold.
>
> ## Q4
> Thanks for the detailed response! Just to clarify – is it true that this invariance holds for the estimate (6) but may not hold for the original definition (or it is hard to prove it there)?
>
> ## RQ4 - 2nd
>
> Yes, it is hard to prove the original definition. As indicated in line 167, both $h_{F,m}$ and $\boldsymbol{X_{:,m}(0)}$ are unknown, making it impossible to measure homophily directly from Eq. (4). To overcome this challenge, we minimize Eq. (6) to estimate $h_{F,m}$, leveraging the fact that the intra-class distances of $\boldsymbol{X_{:,m}(0)}$ are small.
>
> To address the concern about the accuracy of the estimation in Eq. (6), an experiment has been conducted to evaluate the accuracy of this estimation on synthetic datasets. For a given original $h_F$, the estimated $h_F^*$ using Eq. (6) and report the differences $|h_F - h_F^*|$ are tabulated below.
>
> | Original $h_F$ | -1.00 | -0.80 | -0.60 | -0.40 | -0.20 | 0.00 | 0.20 | 0.40 | 0.60 | 0.80 | 1.00 |
> | --- | --- | --- | --- | --- | --- | --- | --- | --- | --- | --- | --- |
> | Estimated $h_F*$ | -0.95 | -0.80 | -0.62 | -0.42 | -0.21 | -0.01 | 0.20 | 0.41 | 0.61 | 0.86 | 1.00 |
> | Differences $\|h_F-h_F*\|$ | 0.05 | 0.00 | 0.02 | 0.02 | 0.01 | 0.01 | 0.00 | 0.01 | 0.01 | 0.06 | 0.00
>
> The results show that the average difference is less than 2\%. In our view, this is an acceptable estimation.
>
> ## Q5
> Here my question was about “model performance” – which measure is used here?
>
> ## RQ5 - 2nd
>
> Thank you for clarifying this with us. We use the **accuracy** for datasets with more than two classes and the **AUC-ROC** for binary-class datasets to evaluate the model performance on the node classification. This measurement has been widely used for assessing GNN performance [1, 2, 3, 6, 7, 8].
>
> Thank you for your valuable suggestions! We will revise our paper accordingly.

---

### Official Review · Reviewer_JQBx · 2024-07-10

**Soundness:** 3
**Presentation:** 3
**Contribution:** 3
**Rating:** 7
**Confidence:** 4

**Summary:**

The paper combines three different metrics (label, structural, and feature) for homophily proposing a Contextual Stochastic Block Model (CSBM-3H) describing these three types of homophily. By doing so it can control the topology and feature generation based on these three homophily metrics. Furthermore, there exists an extensive theoretical analysis of CSBM-3H, also defining new composite metric, named Tri-Hom, that considers all three homophily aspects and overcomes the limitations of the previously proposed and conventional homophily metrics. Importantly, they validate the correlation of model performance (under node classification) based on experimental set-up including 31 real-world benchmark datasets, outperforming various baseline metrics.

**Strengths:**

1) The paper is interesting, and combining all three homophily metrics into one unified metric is novel.

2) The theoretical analysis of the proposed method is robust. The paper introduces three theorems that establish connections to important aspects verified in the literature, thereby unifying multiple works under a single framework.

3) The proposed Contextual Stochastic Block Model, which incorporates three types of homophily, is innovative and could be highly useful for benchmarking general models under varying types and levels of homophily.

4) The experimental section is extensive, including 31 datasets, and yields promising results.

**Weaknesses:**

1) The paper validates performance only under the node classification task. It would be very interesting to also validate Tri-Hom for the task of link prediction.

2) A minor weakness is that some results in Table 1 for certain baselines are quite close, with only marginal improvements. Could the authors comment on this phenomenon? Why might this be the case, and could it indicate that for some models or networks, having only one homophily metric would be sufficient?

**Questions:**

Would it be possible to visualize the generated SBM networks under different levels of homophily and observe the resulting block structures with respect to all different types of homophily?

**Limitations:**

The authors should discuss limitations of their paper and proposed method.

---

> ### Author Rebuttal · Authors · 2024-08-05
>
> ## Part 1/2
> ## W1: The paper validates performance only under the node classification task. It would be very interesting to also validate Tri-Hom for the task of link prediction.
>
> ## RW1
>
> Thanks for your valuable suggestions. It is a good topic to evaluate Tri-Hom and other homophily metrics for link prediction. We only include the results on node classifications in the paper because it is the most classical task on GNNs and most previous studies focus on node classification. Further more, other than space limitation, to validate Tri-Hom for link prediction, there are several challenges that we need to tackle:
>
> **Missing of node labels.** Not all the graphs for the task of link prediction are associated with node labels. However, most of the homophily metrics, as well as our proposed Tri-Hom, are measured based on node labels. It is infeasible to directly apply these metrics on these graphs without labels and more task-oriented adaptations are needed.
>
> **Task Goal.** The goal of node classification is to learn distinguishable node representations, while the goal of link prediction is to predict the likelihood of future or missing links by leveraging reliable structural representations of the graph. This requires an understanding of existing connections and patterns within the graph. Therefore, it is hard to directly apply Tri-Hom on link prediction regarding the different goals in the task.
>
> Apart from that, the main contribution of this paper is disentangling the graph homophily from 3 aspects. Previous theoretical studies [1, 2 ,3 ,4] of graph homophily analysis only consider a single aspect of node classification, while we measure the synergy of 3 types of homophily and successfully validate the theoretical results on both the synthetic and real-world datasets. Considering the challenges in link prediction, here we only measure Tri-Hom with node classification since it is the most widely explored task for GNN performance. In the future, we will explore the Tri-Hom with adaptions to deal with challenges in other types of graph tasks such as link prediction and graph clustering.
>
>
> ## W2: A minor weakness is that some results in Table 1 for certain baselines are quite close, with only marginal improvements. Could the authors comment on this phenomenon? Why might this be the case, and could it indicate that for some models or networks, having only one homophily metric would be sufficient?
>
> ## RW2
>
> As shown in Table 1, the strongest baseline is class homophily ($h_{class}$), which underperforms our proposed Tri-Hom for graph-aware models $\mathcal{J}_h^{\mathcal{G}}$ on GNNs by an average gap of 4\%. It is not surprising that these label-based homophily metrics show strong performance because node labels contain the most important information for the task of node classification on graphs. Label-based homophily metrics outperform all other structural-based or feature-based metrics. The improvement of our metrics comes from the full consideration of all three aspects of graph homophily.
>
> Although we measure performance on 31 real-world datasets, this number is insufficient to fully demonstrate the importance of feature homophily or structural homophily. As shown in Table 5, the distribution of these datasets across the three types of homophily does not vary significantly. Additionally, the Twitch datasets collected by [4] and some heterophilic graphs (Wisconsin, Cornell, Texas, Squirrel, Actor, Chameleon) collected by [5] exhibit high similarity in homophily levels. Therefore, more real-world datasets with varying levels of homophily, particularly for structural and feature homophily, are needed to test the effectiveness of homophily metrics comprehensively. We have attempted to mitigate the impact of limited dataset availability by testing on 31 different datasets.
>
> Due to the scarcity of real-world datasets, we validate the performance of Tri-Hom on synthetic datasets, which offer a diverse range of homophily levels. As shown in Figure 1, the performance surface of Tri-Hom closely mirrors that of GNNs. It is evident that other types of homophily, based on a single factor, underperform compared to Tri-Hom in synthetic datasets, as they respond to only one type of homophily..
>
> Furthermore, other studies [1, 2] that focus on homophily analysis only report results on 9 datasets, evaluating 'good' or 'bad' homophily on a case-by-case basis. Our work represents a significant advancement by fairly comparing homophily metrics across 31 real-world datasets with statistical significance.

---

> > ### Comment · Reviewer_JQBx · 2024-08-07
> > **Comment for the response**
> >
> > I would like to thank the authors for their effort in addressing the concerns and questions raised by me and my fellow reviewers. All of my issues/questions have been adequately addressed. I will therefore increase my score to accept.

---

> > > ### Author Response · Authors · 2024-08-07
> > > **Thank you**
> > >
> > > We are grateful for your recognition and solid support of our paper. Your insightful suggestions have greatly improved it.

---

> ### Author Response · Authors · 2024-08-05
> **Part 2/2**
>
> ## Q1: Would it be possible to visualize the generated SBM networks under different levels of homophily and observe the resulting block structures with respect to all different types of homophily?
>
> ## RQ1:
>
> Thank you for your constructive suggestions. We added the visualization of the generated graphs by CSBM-3H in Figure 2 of the submitted author rebuttal PDF. Based on this visualization, we draw the following conclusions:
>
> **Label homophily.** As label homophily ($h_L$) increases, as shown in Figures 2(a), (b), and (c), nodes are more likely to connect with others that share the same label. Particularly, a high $h_L$ (Figure 2(c)) results in effective structural information, making it easier to delineate class boundaries. In comparison, a medium $h_L$ (Figure 2(b)) is less informative than an extremely low $h_L$ (Figure 2(a)). For instance, the absence of a red node among a red node's neighbors can help infer its class. This observation aligns with Theorem 2.1 in our paper.
>
> **Structural homophily.** As structural homophily ($h_S$) increases, as shown in Figures 2(d), (e), and (f), neighbor distributions become more consistent. Consequently, a high $h_S$ allows us to capture effective structural information, as suggested by Theorem 2.2. Interestingly, we also find that a higher $h_S$ makes a graph resemble planar graphs [7] and periodic graphs [6]. We hypothesize this phenomenon as stable structural information leads to more regular and meaningful structural patterns. In future work, it would be interesting to explore the connection between $h_S$ and these geometric properties of graphs.
>
> **Feature homophily.** Figure 2 (g), (h), and (i) illustrate different levels of feature homophily ($h_F$) within the same graph topology, where colors represent the distance of node features to various classes. Figure 2 (h) demonstrates that a medium positive $h_F$ causes the features of some boundary nodes to exhibit characteristics of neighboring classes. In contrast, a higher positive $h_F$ (Figure 2 (i)) increases feature dependencies, particularly affecting nodes closer to class boundaries. In real-world scenarios, a positive $h_F$ leads entities in a graph to show dependencies with their surrounding neighbors. For instance, people's opinions are influenced by their friends, resulting in similar characteristics. Conversely, a negative $h_F$ causes nodes to become more dissimilar from their neighbors. As shown in Figure 2 (g), a negative $h_F$ creates a distinct boundary between classes. Additionally, within the same classes in Figure 2 (g), nodes colors are different shades with their neighbors. This is because node features become more dissimilar due to the "repulsive force" rather than the "attractive force" induced by a negative $h_F$. In online media, people are likely to argue with those holding different opinions. After such interactions, individuals may reinforce their original opinions, a phenomenon resulting from the "repulsive force" associated with a negative $h_F$.
>
> Finally, we have revised our manuscript according to your valuable reviews. Thank you.
>
>
> ## References
>
> [1] Ma, Y., Liu, X., Shah, N., Tang, J. Is Homophily a Necessity for Graph Neural Networks? In ICLR, 2022.
>
> [2] Luan, S., Hua, C., Xu, M., Lu, Q., Zhu, J., Chang, X.-W., Fu, J., Leskovec, J., Precup, D. When Do Graph Neural Networks Help with Node Classification? Investigating the Homophily Principle on Node Distinguishability. In NeurIPS, 2023.
>
> [3] Wang, J., Guo, Y., Yang, L., Wang, Y. Understanding Heterophily for Graph Neural Networks. CoRR abs/2401.09125, 2024.
>
> [4] Lee, S. Y., Kim, S., Bu, F., Yoo, J., Tang, J., Shin, K. Feature Distribution on Graph Topology Mediates the Effect of Graph Convolution: Homophily Perspective. CoRR abs/2402.04621, 2024.
>
> [5] Pei, H., Wei, B., Chang, K. C.-C., Lei, Y., Yang, B. Geom-GCN: Geometric Graph Convolutional Networks. In ICLR, 2020.
>
> [6] Cohen, E., Megiddo, N. Recognizing properties of periodic graphs. In Applied Geometry and Discrete Mathematics, pp. 135-146, 1990.
>
> [7] Barthelemy, M. Morphogenesis of Spatial Networks. Cham, Switzerland: Springer International Publishing, 2018.

---

### Official Review · Reviewer_bySE · 2024-07-11

**Soundness:** 4
**Presentation:** 3
**Contribution:** 3
**Rating:** 7
**Confidence:** 3

**Summary:**

The paper proposes a novel approach to understanding graph homophily by disentangling it into label, structural, and feature homophily. The introduction of the Tri-Hom metric combines these aspects to provide a more comprehensive measure of GNN performance. CSBM-3H is used to study the impact of these types of homophily. Extensive experimental results on synthetic and real-world datasets show the effectiveness of the findings.

**Strengths:**

1. The paper takes a comprehensive thought on graph homophily and provides with a more detailed understanding of homophily.
2. The introduction of Tri-Hom metric as well as and the CSBM-3H model which is based on the metric is novel.
3. The paper includes both theoretical analysis and empirical validation through synthetic and real-world datasets.

**Weaknesses:**

The introduction of multiple new concepts and metrics might be overwhelming for readers who are not well-versed in graph theory and GNNs. Simplifying explanations or providing more intuitive examples could help.

**Questions:**

1. The definitions of structural and feature homophily are somewhat abstract. Could you provide more concrete examples or case studies to illustrate these concepts?
2. How to translate the finding to real-world applications?

**Limitations:**

Lack concrete examples for readers’ understanding.

---

> ### Author Rebuttal · Authors · 2024-08-05
>
> ## Q1 The definitions of structural and feature homophily are somewhat abstract. Could you provide more concrete examples or case studies to illustrate these concepts?
>
> ## RQ1
>
> Thanks for your valuable suggestions. In line 116 and 146, we show the differences of structural and feature homophily respectively. To help better understand these definitions, in author rebuttal PDF, we add diagrams of 3 types of homophily and visualize how these homophily metrics influences graphs generated by CSBM-3H. As shown in Figure 1, label homophily measures the label consistency along the graph topology, structural homophily measures the consistency of neighbor distribution of intra-class nodes, and feature homophily measures the feature dependencies along the graph topology. Each of the homophily shows a unique aspect. For example, in social networks, label homophily measures how likely the same types (age, hobby, and so on) of people are connected, structural homophily measures the consistency of the neighbors of certain type of people, and feature homophily measures how much the features of people are affected by their neighbors. For more concrete examples or explanations, please refer to the RQ2 of your second question.
>
> ## Q2 How to translate the finding to real-world applications?
>
> ## RQ2
>
> Current real-world applications of graph homophily only focus on label homophily, which cannot align well with GNNs performance as shown in this paper and other studies [1, 2]. To address this weakness, our findings provide a comprehensive view from 3 types of homophily, which could be applied in many real-world applications, such as social networks, recommendation, and urban computing.
>
> ### A. Social Networks
>
> In social networks, homophily is defined as the tendency for people to seek out or be drawn to others who are similar to themselves [3]. This definition primarily explains the consistency of certain characteristics of people within the network topology. Our proposed concepts of structural homophily and feature homophily offer additional insights into social networks.
>
> Structural homophily refers to the similarity of the local neighbors of individuals of the same type, which can be used to analyze the friend circles of specific user groups. For instance, in fraud detection on social media, fraudsters often target older individuals who are more vulnerable to scams, resulting in a high level of structural homophily. Therefore, we can identify potential fraudsters based on their structural connections. However, structural information can vary and may not always be informative. If fraudsters randomly select users to contact, identifying them through their neighbors becomes challenging, leading to a low level of structural homophily. Future research could focus on measuring the level of structural homophily in social networks to better understand user behaviors.
>
> Feature homophily, on the other hand, describes how individuals are influenced by their neighbors. Different types of networks exhibit varying levels of feature homophily. When people discuss similar events online, their opinions may be influenced by those they follow, leading to a higher similarity in features with their neighbors, indicating a high level of feature homophily. Conversely, during online arguments, individuals connect with those holding different opinions. After such interactions, they may reinforce their original viewpoints, resulting in greater dissimilarity with their neighbors, indicating a low level of feature homophily. Investigating user behavior through feature homophily can reveal underlying intentions and improve model performance. Furthermore, feature homophily provides valuable insights into the extent to which users are influenced by their neighbors.
>
> ### B. Recommendation
>
> Previous studies [4, 5] on recommendation systems using graph homophily have primarily focused on label homophily, which may result in misalignment with model performance, similar to what occurs in purely homogeneous graphs. To address this issue, it would be beneficial to define structural homophily within recommendation systems. For instance, we can measure the structural information of users within the same community by assessing the consistency of the items they have purchased. This approach allows us to determine whether this topological information can effectively predict links between users and specific items.
>
> ### C. Urban Computing
>
> A recent study [7] proposes a method to measure spatial graph homophily in urban computing using a spatial diversity score with direction-aware and distance-aware partitions. However, this metric focuses solely on label homophily, leaving a significant opportunity to explore structural homophily. Structural homophily involves measuring the consistency of geographic information among similar types of locations. For example, bookstores are often surrounded by coffee shops, where customers can enjoy coffee while reading books [6], indicating a high level of structural homophily. Conversely, convenience stores might be located near a high-end fashion boutique, a fast-food restaurant, or an office building. Since the geographic information does not reliably indicate the presence of a convenience store, it exhibits a low level of structural homophily. Future research could explore structural homophily in urban graphs to analyze the behaviors of various urban objects, aiding in better city planning.
>
> Thank you for your constructive suggestions. We have revised the paper.

---

> > ### Comment · Reviewer_bySE · 2024-08-12
> >
> > I appreciate the author for addressing my questions. The "Diagrams of 3 types of homophily" you provide in the rebuttal pdf helps me to understand your model better. I will maintain my positive score.

---

> > > ### Author Response · Authors · 2024-08-12
> > > **Thanks**
> > >
> > > Thank you for your positive rating and valuable suggestions.

---

> ### Author Response · Authors · 2024-08-05
> **References**
>
> [1] Ma, Y., Liu, X., Shah, N., Tang, J. Is Homophily a Necessity for Graph Neural Networks? In ICLR, 2022.
>
> [2] Luan, S., Hua, C., Xu, M., Lu, Q., Zhu, J., Chang, X.-W., Fu, J., Leskovec, J., Precup, D. When Do Graph Neural Networks Help with Node Classification? Investigating the Homophily Principle on Node Distinguishability. In NeurIPS, 2023.
>
> [3] Khanam, K. Z., Srivastava, G., Mago, V. The homophily principle in social network analysis: A survey. Multimedia Tools and Applications, 82(6): 8811-8854, 2023.
>
> [4] Jiang, W., Gao, X., Xu, G., et al. Challenging Low Homophily in Social Recommendation. In Proceedings of the ACM on Web Conference, pp. 3476-3484, 2024.
>
> [5] Gholinejad, N., Chehreghani, M. H. Heterophily-Aware Fair Recommendation using Graph Convolutional Networks. arXiv preprint arXiv:2402.03365, 2024.
>
> [6] Niche to Discount: 12 Major Types of Retail Stores \& Retailers, FounderJar, https://www.founderjar.com/types-of-retail-stores/, 2023
>
> [7] Xiao, C., Zhou, J., Huang, J., Xu, T., Xiong, H. Spatial Heterophily Aware Graph Neural Networks. In KDD, pp. 2752-2763, 2023.

---

### Official Review · Reviewer_SC6h · 2024-07-11

**Soundness:** 3
**Presentation:** 3
**Contribution:** 3
**Rating:** 6
**Confidence:** 4

**Summary:**

This paper evaluates graph homophily from the perspectives of label, structure, and feature, which disentangle the dependencies of these three aspects. The theoretical analysis and experimental evaluations demonstrate the effectiveness of Tri-Hom.

**Strengths:**

1. This paper is innovative and significant in evaluating graph properties from their central components, namely, label, feature, and structure.
2. This paper highlight the missing component on evaluating the graph homophily, marking it as novel compared to existing methods for designing graph learning methods and new evaluation metrics.
3. There have been extensive experiments conducted on synthetic and real-world datasets.

**Weaknesses:**

1. This paper lacks suggestions for designing models. New metrics are provided for evaluating graph homophily properties. However, as new models are continually proposed, it would be beneficial if the authors could provide some guidelines for designing models when tackling new datasets.

2. The connections and distinctions between the proposed metrics and the existing metrics from the label, feature, and structure perspectives need further clarification.

**Questions:**

None

**Limitations:**

please check the weaknesses.

---

> ### Author Rebuttal · Authors · 2024-08-05
>
> ### Part 1/2
>
> ## Q1: Suggestions for designing models.
> ## RQ1
>
> Thanks for your positive rating and constructive suggestions. It is interesting to see how our conclusions can guide the model design. Due to the page limitation, we did not share suggestions in model design. Here we show some guidelines for model designs and future directions from 3 perspectives: label homophily, structural homophily, and feature homophily as follows:
>
> ### A. Label homophily
>
> We discussed how label homophily influences GCN and MLP, providing both theoretical proof (Theorem 2.1 in Line 232) and empirical experiments (the three sub-figures in the first row of Figure 4 in Appendix D.6). Our results show that GCN performs better than MLP in conditions of extremely low homophily (good heterophily [1]), but significantly worse than MLP in medium levels of homophily (mid-homophily pitfall [2]). This suggests that GCNs sometimes fail to extract effective topological information. To mitigate this weakness, it is preferable to add a residual connection to GNNs or introduce a learnable parameter that allows the model to balance graph-aware and graph-agnostic information. The necessity of residual connections has been verified in previous studies [3, 4, 5].
>
> In addition to model design at the graph level, we can also consider a fine-grained approach at the node level. Our results indicate that GCN does not always outperform MLP, suggesting that different models could be applied to nodes with varying levels of label homophily. To our knowledge, this type of personalized design has rarely been explored in current GNN research on heterophilic graphs, yet it holds significant potential for improving overall model performance.
>
> ### B. Structural homophily
>
> As mentioned in Theorem 2.2, the performance of graph-aware models improves with an increase in structural homophily. This leads to a crucial question: how can we deal with graphs exhibiting varying levels of structural homophily (i.e., the consistency of structural information among intra-class nodes)? To address this issue, we can enhance current GNNs using two approaches: message-passing calibration and graph rewriting.
>
> For the message passing calibration, several methods, such as GPRGNN [13], FB-GNNs [14], and ACM-GNNs [15], propose adding an additional high-pass filter to capture local variations and details in the graph structure. When structural homophily is low, the high-pass filter captures the diversification of individual nodes. Along with a low-pass filter, these methods perform well on graphs with varying levels of homophily. However, the high-pass filters used in these methods cannot capture more complicated structural information. Since $\mathcal{S}(\cdot)$ in structural homophily can be any measurement of structural homophily, this metric can evaluate more complex graph structures. In the future, it is promising to design novel filters based on structural homophily to capture more intricate structural information and improve model performance.
>
> For the graph structure rewriting, many methods (MVGCN [6], GloGNN [7], WRGNN [8]) propose deriving a new graph topology based on node features or embeddings. This operation improves the connectivity of nodes with similar semantic contexts, thereby enhancing model performance. However, this rewriting could connect nodes from different classes, which impedes GNN performance. To resolve this, we can measure class-wise structural homophily as shown in Eq. (2) and design adaptations for different classes, which will be particularly beneficial for class-imbalanced graphs, such as in bot detection and fraud detection. Furthermore, we can adapt the structural measurement function $\mathcal{S}(\cdot)$. Since most current structural rewriting methods do not evaluate the informativeness of their rewriting basis (node embeddings with structural information), the proposed structural homophily can serve as a metric to evaluate which types of rewriting basis to select. For example, Geom-GNN [9] uses Isomap [10], Poincare embedding [11], and struc2vec [12] to construct new neighbors of nodes and empirically determines the best approach. Our structural homophily could identify the most effective embedding approach before training GNNs. Therefore, structural homophily provides a guideline for graph rewriting methods.
>
> ### C. Feature homophily
>
> In Appendix B, we thoroughly discuss the motivation behind feature homophily, where feature dependencies measure how node features are influenced by their neighbors. To our knowledge, only a few GNNs [16] consider these feature dependencies in their design. There is significant potential to explore how feature dependencies function in graphs. For instance, in social networks, people's opinions are affected by those around them. Identifying different user types while filtering out the noise introduced by their neighbors remains an open question. Both our theoretical results (Theorem 2.3) and empirical results (Figure 4) demonstrate the synergy between feature homophily and label homophily in enhancing model performance. Based on these findings, future work could focus on designing various graph filters to optimize the objective in Eq. (8) by considering both label and feature homophily. Furthermore, feature homophily can explain the presence of node features, making it worthwhile to investigate how much features are influenced by their neighbors, particularly in temporal graphs.
>
> In conclusion, our findings offer valuable insights for designing models. Compared with previous studies that only analyze the single factor of label homophily [1, 2], structural homophily [17], or feature homophily [18] on GNNs, our work consider the synergy of all 3 types of homophily. We validated our theoretical results using both synthetic and real-world datasets. We believe this work represents a significant advancement on graph homophily and provides numerous intriguing directions for future research.

---

> ### Author Response · Authors · 2024-08-05
> **Part 2/2**
>
> ## Q2: The connections and distinctions between the proposed metrics and the existing metrics from the label, feature, and structure perspectives need further clarification.
>
> ## RQ2
>  In lines 116 and 146, we highlight the differences between the proposed structural homophily and feature homophily. These definitions represent the basic elements of a graph: label, structural, and feature information, thereby providing a comprehensive understanding of graph homophily. Label homophily ($h_L$) describes the label consistency along the topology, which has been widely used in previous studies [2, 5, 7]. However, $h_L$ cannot capture the consistency of structural information among intra-class nodes, which also influences GNN performance. To address this limitation, we propose structural homophily ($h_S$) to describe the consistency of structural information among intra-class nodes. Unlike existing structural-based metrics [1, 14], our metric allows for any kind of structural measurement function and can be easily incorporated into CSBM-3H for analysis. Feature homophily ($h_F$) measures the feature consistency of nodes with their neighbors and can be fully disentangled from $h_L$ and $h_S$, something other feature-based metrics [18] cannot achieve (as shown in line 138). Furthermore, we have included diagrams and visualizations of the three types of homophily in the author rebuttal PDF to aid in understanding these definitions. According to your suggestions, we have revised our manuscript. Thank you.
>
> ### References
>
> [1] Ma, Y., Liu, X., Shah, N., Tang, J. Is Homophily a Necessity for Graph Neural Networks? In ICLR, 2022.
>
> [2] Luan, S., Hua, C., Xu, M., Lu, Q., Zhu, J., Chang, X.-W., Fu, J., Leskovec, J., Precup, D. When Do Graph Neural Networks Help with Node Classification? Investigating the Homophily Principle on Node Distinguishability. In NeurIPS, 2023.
>
> [3] Luo, Y., Shi, L., Wu, X.-M. Classic GNNs are Strong Baselines: Reassessing GNNs for Node Classification. CoRR abs/2406.08993, 2024.
>
> [4] Xu, K., Hu, W., Leskovec, J., Jegelka, S. How Powerful are Graph Neural Networks? In ICLR, 2019.
>
> [5] Platonov, O., Kuznedelev, D., Diskin, M., Babenko, A., Prokhorenkova, L. A critical look at the evaluation of GNNs under heterophily: Are we really making progress? In ICLR, 2023.
>
> [6] Wang, Y., Xiang, S., Pan, C. Improving the homophily of heterophilic graphs for semi-supervised node classification. In ICME, 2023.
>
> [7] Li, X., Zhu, R., Cheng, Y., Shan, C., Luo, S., Li, D., Qian, W. Finding Global Homophily in Graph Neural Networks When Meeting Heterophily. In ICML, pp. 13242-13256, 2022.
>
> [8] Suresh, S., Budde, V., Neville, J., Li, P., Ma, J. Breaking the limit of graph neural networks by improving the assortativity of graphs with local mixing patterns. In SIGKDD, 2021.
>
> [9] Pei, H., Wei, B., Chang, K. C.-C., Lei, Y., Yang, B. Geom-GCN: Geometric Graph Convolutional Networks. In ICLR, 2020.
>
> [10] Tenenbaum, J. B., De Silva, V., Langford, J. C. A global geometric framework for nonlinear dimensionality reduction. Science, 290(5500):2319–2323, 2000.
>
> [11] Nickel, M., Kiela, D. Poincare embeddings for learning hierarchical representations. In NeurIPS, 2017.
>
> [12] Ribeiro, L. F. R., Saverese, P. H. P., Figueiredo, D. R. struc2vec: Learning node representations from structural identity. In SIGKDD, 2017.
>
> [13] Chien, E., Peng, J., Li, P., Milenkovic, O. Adaptive universal generalized pagerank graph neural network. In ICLR, 2021.
>
> [14] Luan, S., Zhao, M., Hua, C., Chang, X.-W., Precup, D. Complete the missing half: Augmenting aggregation filtering with diversification for graph convolutional networks. In NeurIPS 2022 Workshop: New Frontiers in Graph Learning, 2022.
>
> [15] Luan, S., Hua, C., Lu, Q., Zhu, J., Zhao, M., Zhang, S., Chang, X.-W., Precup, D. Revisiting heterophily for graph neural networks. In NeurIPS, 2022.
>
> [16] Shi, D., Han, A., Lin, L., Guo, Y., Wang, Z., Gao, J. Design Your Own Universe: A Physics-Informed Agnostic Method for Enhancing Graph Neural Networks. CoRR abs/2401.14580, 2024.
>
> [17] Wang, J., Guo, Y., Yang, L., Wang, Y. Understanding Heterophily for Graph Neural Networks. CoRR abs/2401.09125, 2024.
>
> [18] Lee, S. Y., Kim, S., Bu, F., Yoo, J., Tang, J., Shin, K. Feature Distribution on Graph Topology Mediates the Effect of Graph Convolution: Homophily Perspective. CoRR abs/2402.04621, 2024.

---

> ### Author Response · Authors · 2024-08-13
> **reply to your concerns**
>
> Please let me know if our responses have addressed your concerns. Thank you!

---

### Author Rebuttal · Authors · 2024-08-07

We would like to thank all the reviewers for their valuable feedback. In this author rebuttal PDF, we provide diagrams and visualizations to help better understand our proposed definitions.

Figure 1 shows the definition of three types of homophily. Label homophily $h_L$ measures the label consistency along the graph topology, structural homophily $h_S$ measures the consistency of structural information within intra-class nodes, and feature homophily $h_F$ represents the feature dependencies along the graph topology. Each type of homophily represents a unique perspective towards the concept of graph homophily. In our paper, we disentangle the concept of graph homophily and investigate the synergy of these three types of homophily in Graph Neural Networks (GNNs) through theorems, simulations, and real-world experiments, providing a better understanding of graph homophily.

Figure 2 visualizes the impact of the three types of homophily in CSBM-3H: **1. Label homophily.** As label homophily ($h_L$) increases, nodes are more likely to connect with others sharing the same label, with high $h_L$ (Figure 2(c)) providing clearer class boundaries. Medium $h_L$ (Figure 2(b)) is less informative than very low $h_L$ (Figure 2(a)). **2. Structural homophily.** As shown in Figure 2(d), (e), and (f), when structural homophily ($h_S$) increases, neighbor distributions become more consistent, and high $h_S$ captures effective structural information. Besides, a higher $h_S$ makes a graph resemble planar and periodic graphs, suggesting stable structural information leads to regular patterns. **3. Feature homophily.** Figures 2(g), (h), and (i) show different levels of feature homophily ($h_F$) in the same graph topology. Medium positive $h_F$ (Figure 2(h)) causes boundary node features to resemble neighboring classes. Higher positive $h_F$ (Figure 2(i)) increases feature dependencies, especially near class boundaries. In real-world scenarios, positive $h_F$ causes entities to show dependencies with neighbors, like people influenced by friends. Negative $h_F$ makes nodes dissimilar from neighbors, creating distinct class boundaries and varied shades within the same class (Figure 2(g)). This "repulsive force" leads to reinforced original opinions after interactions, akin to online arguments.

---

### Decision · Program_Chairs · 2024-09-25

**Decision:**

Accept (poster)

**Comment:**

The paper proposes a method to understanding graph homophily by disentangling it into label, structural, and feature homophily. It introduces Tri-Hom metric that combines these aspects to provide a more comprehensive measure of GNN performance. Extensive experimental results on synthetic and real-world datasets show the effectiveness of the findings.